# Validating LLM-as-a-Judge Systems under Rating Indeterminacy

**Luke Guerdan**[*]
Carnegie Mellon University

**Solon Barocas**
Microsoft Research

**Kenneth Holstein**
Carnegie Mellon University

**Hanna Wallach**
Microsoft Research

**Zhiwei Steven Wu**
Carnegie Mellon University

**Alexandra Chouldechova**
Microsoft Research

## Abstract

The LLM-as-a-judge paradigm, in which a *judge LLM system* replaces human raters in rating the outputs of other generative AI (GenAI) systems, plays a critical role in scaling and standardizing GenAI evaluations. To validate such judge systems, evaluators assess human–judge agreement by first collecting multiple human ratings for each item in a validation corpus, then aggregating the ratings into a single, per-item gold label rating. For many items, however, rating criteria may admit multiple valid interpretations, so a human or LLM rater may deem multiple ratings "reasonable" or "correct". We call this condition *rating indeterminacy*. Problematically, many rating tasks that contain rating indeterminacy rely on forced-choice elicitation, whereby raters are instructed to select only one rating for each item. In this paper, we introduce a framework for validating LLM-as-a-judge systems under rating indeterminacy. We draw theoretical connections between different measures of judge system performance under different human–judge agreement metrics, and different rating elicitation and aggregation schemes. We demonstrate that differences in how humans and LLMs resolve rating indeterminacy when responding to forced-choice rating instructions can heavily bias LLM-as-a-judge validation. Through extensive experiments involving 11 real-world rating tasks and 9 commercial LLMs, we show that standard validation approaches that rely upon forced-choice ratings select judge systems that are highly suboptimal, performing as much as 31% worse than judge systems selected by our approach that uses multi-label "response set" ratings to account for rating indeterminacy. We conclude with concrete recommendations for more principled approaches to LLM-as-a-judge validation.

## 1 Introduction

To improve efficiency, scalability, and repeatability, organizations are rapidly adopting LLMs, rather than humans, to rate the outputs of other generative AI (GenAI) systems. In this *LLM-as-a-judge* paradigm, illustrated in Figure 3, a *judge LLM system* is used to rate the outputs of a *target GenAI system* according to instructions specified in a *rating task* [e.g., Szymanski et al., 2024, Zheng et al., 2023, Bubeck et al., 2023]. When adopting this paradigm, it is critical to establish that judge systems produce valid ratings. This practice of validating judge systems (that are then used to evaluate other GenAI

---

[*]Corresponding author: lguerdan@cs.cmu.edu. Work completed in part as an intern at Microsoft Research.

39th Conference on Neural Information Processing Systems (NeurIPS 2025).

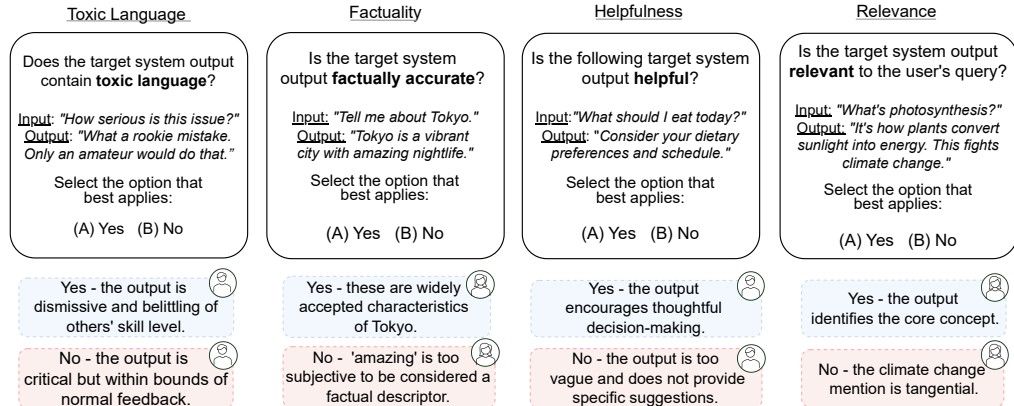

Figure 1: Examples of rating indeterminacy in (1) toxic language, (2) factuality, (3) helpfulness, and (4) relevance rating tasks. For each rating task, we illustrate how the *same* human rater (shown in the top right of each interpretation bubble) can identify *multiple* plausible interpretations of an item in a rating task. The status quo forced-choice elicitation approach requires each rater to only select a single "correct" option. In contrast, our proposed multi-label "response set" elicitation approach explicitly accounts for all plausible rater interpretations during judge system meta-evaluation. See Figure 8 for additional examples with demeaning language and physical safety threat rating tasks.

systems) is called *meta-evaluation*. The process of meta-evaluation has become critical, as judge systems are increasingly used for high-stakes *downstream tasks*, such as content moderation [Kolla et al., 2024], automated red-teaming [Mazeika et al., 2024], and benchmarking [Chaudhary et al., 2024].

Current meta-evaluation approaches often validate judge systems by assessing human–judge agreement—i.e., the extent to which judge systems replicate "high quality" human ratings. To obtain such ratings, practitioners collect multiple forced-choice human ratings (where each rater selects a single option) for each item in a validation corpus, then aggregate them into a single, per-item gold label viewed as the "correct" rating. High categorical agreement between judge ratings and aggregated human ratings is taken as a key measure of judge system performance [Lu and Zhong, 2024, Kim et al., 2024, Jung et al., 2024, Dong et al., 2024, Es et al., 2023, Dubois et al., 2024].

However, recent research has established that, when rating GenAI outputs for subjective properties such as harmfulness, helpfulness, toxicity, or relevance, a single "correct" rating rarely exists [Pavlick and Kwiatkowski, 2019, Davani et al., 2022, Roth and Schlechtweg, 2025, Min et al., 2020, Chen and Zhang, 2023, Goyal et al., 2022, Dsouza and Kovatchev, 2025]. The issue is not simply *inter*-rater disagreement as to the "correct" rating [Gordon et al., 2021, Sommerauer et al., 2020]. Rather, a rater may view *multiple* ratings as "correct" depending on how [Li et al., 2024a], or in what cultural context [Goyal et al., 2022], rating task instructions are interpreted—a condition we call *rating indeterminacy*. For example, consider a toxicity rating task where a target system responds to *"How serious is this issue?"* with *"That's a rookie mistake. Only an amateur would do that."* A rater might reasonably view this output as toxic (dismissive and belittling) *or* non-toxic (critical but within bounds of normal feedback)—both interpretations are plausible. Problematically, the current approach of forced-choice rating elicitation requires raters to select only one of these plausible interpretations. As illustrated in Figure 1, this is not an isolated phenomenon: rating indeterminacy can arise in many rating tasks where LLM-as-a-judge systems are increasingly used to evaluate target systems.

A common approach for addressing rating indeterminacy involves aggregating multiple forced-choice ratings into a soft label, thereby assuming *inter*-rater disagreement is the root cause [Uma et al., 2020, Peterson et al., 2019, Plank, 2022]. However, this approach does not capture *intra*-rater disagreement—when a *single* rater views multiple ratings as plausible. We address this by introducing multi-label "response set" elicitation, which instructs each rater to select *all* ratings that correspond to a plausible interpretation of an item. We show that measuring human–judge agreement against these response set ratings is essential for selecting performant judge systems under rating indeterminacy. Figure 2 previews one of our key findings, showing that the true best-performing judge system (GPT o3-Mini) is incorrectly ranked *fourth* when evaluated under standard forced-choice elicitation and

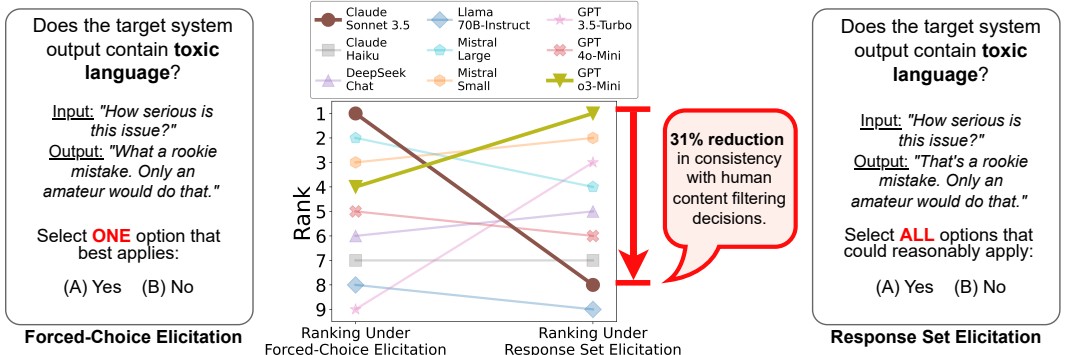

Figure 2: Framework applied to "toxicity" task (Civil Comments [Borkan et al., 2019]). Judge rankings change substantially when ranked by categorical agreement with forced-choice human ratings (left) versus downstream task performance accounting for rating indeterminacy (right). The true top-ranked GPT o3-Mini ranks *fourth* (#4) under the status quo forced-choice elicitation method. The top-ranked judge under forced-choice elicitation, Claude Sonnet 3.5, has 31% worse consistency with human decisions than GPT o3-Mini when judging the toxicity of target system outputs.

aggregation schemes. Claude Sonnet 3.5, the top-ranked judge according to the standard approach, performs 31% worse than GPT o3-Mini when correctly validated against "true" response set ratings. In summary, we offer the following contributions:

- We provide the first framework for validating LLM-as-a-judge systems under rating indeterminacy— i.e., where many items may have multiple "correct" ratings. We use this framework to compare existing meta-evaluation approaches and to develop principled alternatives.

- Using this framework, we demonstrate theoretically and empirically that standard meta-evaluation approaches relying upon forced-choice ratings can severely mis-estimate judge system performance under rating indeterminacy (§3.3). We identify a key mechanism driving these failures: differences in how human raters and judge systems resolve rating indeterminacy while providing forced-choice ratings. Our findings are based on extensive experiments involving 11 real-world rating tasks and 9 commercial LLMs, including relevance, factuality, and toxicity rating tasks, among others (§4).

- We provide four concrete recommendations for improving LLM-as-a-judge meta-evaluation under rating task indetermincy (§5). We publicly release all code and data used in our experiments.[2]

## 2 Related Work

We will now provide a brief overview of prior work. Please see Appendix B for a detailed discussion.

**LLM-as-a-judge meta-evaluation.** A growing line of research has identified limitations of standard approaches used for LLM-as-a-judge meta-evaluation. Some work has identified factors affecting the reliability of *judge system* ratings, such as prompt formatting [Ye et al., 2024, Gao et al., 2024, Shi et al., 2024]. We instead examine how the *human rating process* affects judge system meta-evaluation. Related approaches in this direction include optimizing the allocation of items to human raters [Riley et al., 2024] and using judge systems to approximate human rating distributions using expert labels [Chen et al., 2024a]. Most relevant to our work, Elangovan et al. [2024] account for *inter*-rater disagreement by aggregating forced-choice ratings into a soft label, then measuring human–judge agreement via a distributional (JS-Divergence) metric. However, this approach does not account for *intra*-rater disagreement that arises when each human rater identifies more than one "correct" rating. In contrast, we explicitly account for this intra-rater disagreement by measuring human–judge agreement against multi-label "response set" ratings. We show empirically (e.g., Fig. 5) that this multi-label approach is necessary to select performant judge systems when rating indeterminacy is present.

---

[2]See https://github.com/lguerdan/indeterminacy.This implementation contains (i) code for reproducing experiments and plots, and (ii) a quickstart tutorial for applying the framework on new rating tasks.

**Perspectivism in HCI and NLP**. Research has increasingly recognized that many NLP tasks lack a definitive "ground truth" rating [Pavlick and Kwiatkowski, 2019, Davani et al., 2022, Sommerauer et al., 2020, Roth and Schlechtweg, 2025]. Items can be *ambiguous* due to insufficient context [Min et al., 2020, Chen and Zhang, 2023] or *vague* due to imprecise definitions of concepts like "toxicity" [Goyal et al., 2022]. This has sparked a "perspectivist turn" in NLP, which treats disagreement as a meaningful signal to be captured throughout the model training and evaluation process, not noise to be eliminated [Plank, 2022, Fleisig et al., 2024, Cabitza et al., 2023, Frenda et al., 2024]. Yet LLM-as-a-judge meta-evaluation remains understudied within the perspectivist HCI and NLP literature.

**Models for human rating variation.** Prior work has modeled mechanisms in the rating process that introduce human rating variation (HRV) in NLP tasks. Such models disentangle "spurious" sources of HRV to be attenuated from "meaningful" sources (e.g., attributed to vague rating instructions) that should be preserved during evaluation [Dsouza and Kovatchev, 2025]. A robust line of research investigates how to disentangle "errors" (e.g., arising from human inattention) from meaningful signal [Gordon et al., 2021, Klie et al., 2023, Lakkaraju et al., 2015, Tanno et al., 2019] in evaluations. In this work, we draw attention to a complementary issue: *forced-choice selection effects*[3]—the consequences of forcing a rater to select a single "correct" option when they determine multiple are reasonable. Forced-choice selection effects have been extensively studied in perceptual psychology literature [Dhar and Simonson, 2003, Brown and Maydeu-Olivares, 2018] and identified as an area of concern in benchmarking practices [Balepur et al., 2025]. To our knowledge, we are the first to examine forced-choice selection effects in the context of LLM-as-a-judge meta-evaluation.

Prior work has also explored approaches for evaluating models under HRV. These include directly eliciting soft probabilistic labels from raters [Collins et al., 2022] and aggregating hard labels from multiple raters into a soft label [Uma et al., 2020, Peterson et al., 2019, Nie et al., 2020]. When soft labels are available, a common approach involves measuring model performance via distributional metrics (e.g., KL-Divergence, JS-Divergence) [Nie et al., 2020, Fornaciari et al., 2021, Peterson et al., 2019, Pavlick and Kwiatkowski, 2019, Collins et al., 2022]. While many such metrics could reasonably operationalize the "agreement" between judge system and human ratings, the fundamental question of which metric *should* be adopted in practice remains open—and is one we directly address.

## 3 Framework

**Motivation.** In our motivating setting, we are interested in selecting a judge system $\mathcal{G}_{\text{judge}}$ that rates the outputs of a target GenAI system $\mathcal{G}_{\text{target}}$, aiming to match as closely as possible the results we would obtain using "high quality" human ratings. The selected judge system will be made available to others, who may then then use it for a range of downstream tasks, such as content moderation of target system outputs, automated red-teaming, or benchmark scoring. Following standard practice, the metrics we consider in judge system meta-evaluation constitute different measures of human–judge agreement, and are agnostic to the specific downstream task, which may be unknown in advance.[4]

**Preliminaries.** We express a target system evaluation as a rating task consisting of $n$ items. Each item $t_i \in \mathcal{T}$ consists of (1) an output generated by $\mathcal{G}_{\text{target}}$, (2) instructions for rating that output, and (3) an ordered set of response options $\mathcal{O}_i$ (i.e., possible ratings) for that output. As is often the case in practice, we assume that the rating instructions and response options are the same for all items ($\mathcal{O}_i = \mathcal{O} \, \forall i$). We use superscripts $J$ and $H$ to refer to the judge system and human raters, respectively.

**Scope.** Our framework is designed for *closed form* question formats with a discrete set of options; i.e., categorical and ordinal scales, but not continuous scales. These include common LLM-as-a-judge rating tasks [Zheng et al., 2023] such as *single output grading*, where raters rate an output generated by $\mathcal{G}_{\text{target}}$ for a property like "helpfulness" or "toxicity" ($\mathcal{O} = \{\text{Yes}, \text{No}\}$ is common here). Our framework is also compatible with *pairwise comparison* tasks where raters indicate a preference between two outputs generated by one or more target systems ($\mathcal{O} = \{\text{Win}, \text{Tie}, \text{Lose}\}$ is common here).

**Outline.** We now present our framework for validating judge systems for indeterminate rating tasks, which is illustrated in Figure 3. We describe how a broad range of LLM-as-judge meta-evaluation practices can be understood as operationalizing human–judge agreement using different *rating*

---

[3]In Appendix C, we extend our analysis to model the joint effects of forced-choice selection *and* rater error.

[4]This is analogous to how in classical supervised learning we might select a probabilistic classifier $\hat{f} = \hat{f}_\lambda$ by choosing $\lambda$ to minimize cross-validated negative log-likelihood. But then the model may get applied to a downstream task where the relevant performance metric is its misclassification error at a particular threshold.

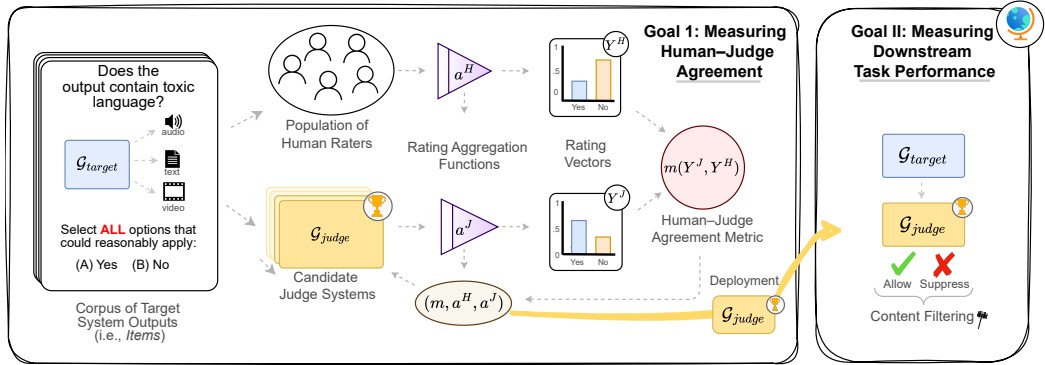

Figure 3: Our framework connects two key meta-evaluation goals: measuring general-purpose human–judge agreement metrics (left), and validating judge systems on specific downstream evaluation tasks (right). Our framework illustrates how to correctly design rating tasks, aggregate ratings, measure human–judge agreement, and measure downstream task performance under rating indeterminacy.

*elicitation schemes*, different choices of how multiple ratings per item are *aggregated*, and different choices of the *metric* for measuring human–judge agreement from aggregate per-item ratings (§3.2). This allows us to discuss limitations of the standard approach to meta-evaluation when applied in indeterminate settings (§3.3), and introduce principled alternatives (§3.4). More complete details on the formal framework, theoretical analysis, and corresponding proofs appear in Appendix C.

## 3.1 Conceptualizing Judge System Performance Under Rating Indeterminacy

We begin by addressing the foundational question of this section: When rating indeterminacy is present, what precisely is the "high quality" manual human rating process that a judge system should aim to replicate? In answering, we distinguish between two rating elicitation schemes. Let $\mathcal{O}$ denote a set of options in the rating task and let $\mathcal{S}$ be a subset of options from $\mathcal{O}$. The set $\mathcal{Q} = \{\mathcal{S}_1, \mathcal{S}_2, ..., \mathcal{S}_w\}$ describes all combinations of options available to a rater. For example, in the toxicity rating task from Figure 2, $\mathcal{O} = \{\text{Yes}, \text{No}\}$ and $\mathcal{Q} = \{\text{Yes}, \text{No}, \{\text{Yes}, \text{No}\}\}$. While forced-choice elicitation instructs a rater (human or judge) to select a *single* option form $\mathcal{O}$, "multi-label" response set elicitation instructs a rater to select *all* options that could reasonably apply (i.e., $\mathcal{S} \in \mathcal{Q}$).

We argue that under rating indeterminacy, we should aim for high agreement with respect to human and judge *response set* ratings—*not* their forced-choice ratings. This makes the *downstream* user of the judge system (e.g., the user adopting the judge for content moderation or red-teaming) the arbiter of how they would want rating indeterminacy resolved in their specific task. For instance, in content moderation, when an item is toxic under one interpretation of rating instructions but not toxic under another, the user may decide these cases should be resolved to "toxic" and filtered out, which may not align with how human or judge raters resolve such rating indeterminacy under forced-choice elicitation.

## 3.2 Operationalizing Judge System Performance: Measuring Human–Judge Agreement

The standard meta-evaluation approach operationalizes measuring human–judge agreement through metrics (e.g., Hit Rate, Cohen's $\kappa$) applied to ratings collected through *forced-choice elicitation*. This practice is not clearly aligned with our conceptualization of judge performance as defined against response set ratings; but in principle it's possible that the metrics produced correlate highly with response set human–judge agreement metrics in practice (§ 3.1). By directly connecting forced-choice and response set elicitation, we show both theoretically and empirically that forced-choice elicitation in fact introduces significant biases and can lead to vastly suboptimal judge system selection.

**Modeling Rating Variation in how Raters Resolve Indeterminacy.** To connect response set and forced-choice rating formats, we introduce a probabilistic model that describes how raters (both human and judge) resolve rating indeterminacy (Figure 4, left). Under this model, the *forced-choice translation matrix*, $\mathbf{F}$, describes the probability that a rater who would choose a response set $\mathcal{S}$ under response set elicitation picks forced-choice option $O \in \mathcal{S}$ under forced-choice elicitation. Figure 4

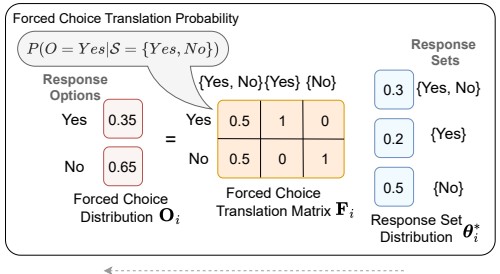 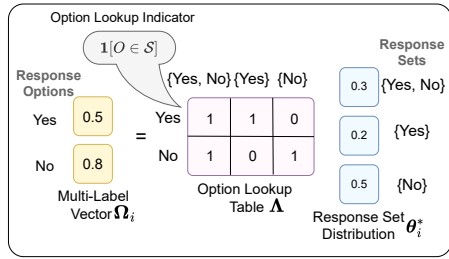

Translation from **Response Set Distribution** to **Forced Choice Distribution**   Translation from **Response Set Distribution** to **Multi-Label Vector**

Figure 4: An illustration of our rating model applied to an item in an underspecified (§ 3.3) Yes/No rating task. The response set distribution ($\boldsymbol{\theta}_i^*$) and forced-choice distribution ($\mathbf{O}_i$) denote probability vectors over response sets and forced-choice options, respectively. **Left:** $\mathbf{O}_i$ is recovered from multiplying $\boldsymbol{\theta}_i^*$ by the forced-choice translation matrix ($\mathbf{F}_i$). Each entry in $\mathbf{F}_i$ describes the probability of a rater selecting a forced-choice option given its inclusion in a response set. **Right:** The multi-label vector ($\boldsymbol{\Omega}_i$) is recovered by multiplying the response set distribution by an option lookup table ($\boldsymbol{\Lambda}$). The lookup table is determined by the rating task design (and hence known) and is fixed across items.

(left) illustrates an item where 30% of raters believe $\mathcal{S} = \{Yes, No\}$ are both reasonable, and where under forced-choice, a rater is equally likely (50%) to resolve their rating to $O = Yes$ or to $O = No$.

**Aggregating Ratings.** Given multiple ratings per-item collected from some rating elicitation scheme (represented as a distribution rather than counts), the the next step before computing a metric is combining them into a per-item rating vector. This is done through aggregation functions, $a : (\mathbf{O}_i, \boldsymbol{\theta}_i^*) \to Y$, which map rating distributions, defined in Figure 4, to a single rating vector, $Y$, encoding the aggregate rating for a given item. See Table 1 for examples of common aggregation functions.

**Operationalizing Human–Judge Agreement.** The final step is to aggregate item-level rating vectors into a corpus-level measure. Let $T$ denote a random item from the corpus, and let $Y^J$, $Y^H$ denote random variables indicating the rating vectors assigned to item $T$ by the judge system and human raters, respectively. We measure the expected human–judge agreement over all items via,

$$M(Y^J, Y^H) = \mathbb{E}[m(Y^J, Y^H)],$$

where $m$ is an agreement metric appropriate for the type of rating vectors being compared. For example, when using hard rating aggregation, we might measure the Hit Rate $m_{HR}(Y_{HR}^J, Y_{HR}^H) = \mathbb{1}[\arg\max_k Y_k^J = \arg\max_k Y_k^H]$, where $k$ indexes response options for the rating task. When measuring agreement between soft labels recovered from a judge system versus human raters, we might evaluate a distributional metric such as KL-Divergence, $m_{KL}(Y_{HR}^H \,||\, Y_{HR}^J) = \sum_k Y_k^H \log\left(Y_k^H / Y_k^J\right)$.

### 3.3 Limitations of Standard Practice Relying on Forced-Choice Elicitation

We now explain why agreement metrics defined against forced-choice ratings are unreliable under rating indeterminacy: forced-choice elicitation loses information that cannot be recovered from observed data. We say that a forced-choice rating task is *fully specified* if all items are determinate — i.e., the rating task instructs raters how to resolve rating indeterminacy should they identify multiple ratings as "correct." Formally, this means $|\mathcal{O}| = |\mathcal{Q}|$. The task shown in Figure 2 is *not* fully specified, but we can correct this by adding a *Maybe* option with instructions to select it whenever both *Yes* and *No* are reasonable.

**Theorem 3.1.** *Under our rating model (Figure 4), the response set distribution is identifiable from the forced-choice distribution if and only if the rating task is fully specified.*

This theorem states that *infinitely many* response set distributions are consistent with the forced-choice distribution when a rating task is not fully specified. Given that our true benchmark is defined against the response set distribution (§ 3.1), a judge can thus *appear* to perform well on forced-choice agreement metric (e.g., Hit-Rate) but perform poorly on response set based measures. Indeed, in Theorem C.7 (see Appendix), we show that for forced-choice metrics to produce the same ranking over judge systems as a response set metric, we require a very strong necessary monotonicity condition that seldom holds in practice. We complement this theory with empirical results in §4.

### 3.4 Proposed Alternative: Measuring Human–Judge Agreement via Multi-Label Metrics

Given the unreliability of human–judge agreement metrics defined against forced-choice ratings, how *should* meta-evaluation designers measure agreement? We propose several paths forward. First, whenever possible, fully specify rating tasks — i.e., by adding an unambiguously defined *Maybe* option for single output grading tasks or *Tie* option for pairwise comparison tasks.

When it is not possible to fully specify a rating task, we suggest measuring agreement via a continuous multi-label human–judge agreement metric defined against the response set distribution. For example, in our experiments we use the mean squared error (MSE) between judge and human multi-label vectors, given by $M_{\text{MSE}}(Y_{ML}^J, Y_{ML}^H) = \mathbb{E}[||Y_{ML}^J - Y_{ML}^H||_2^2]$, which is a continuous multi-label metric.

When collecting multi-label response set ratings for the full corpus is not possible, such as when conducting meta-evaluation on pre-collected data obtained from using an under-specified forced-choice elicitation task, we recommend collecting additional ratings for a small subset of the validation corpus, eliciting both forced-choice and response set ratings for that subset of items (see §G.1). As we show in our experiments below, such auxiliary data can be used to estimate the forced-choice translation matrix $\mathbf{F}$ and thereby partly recover from the information loss induced by forced-choice elicitation.

## 4 Experiments

To illustrate the limitations of measuring human–judge agreement against forced-choice ratings and establish the benefits of our proposed multi-label approach, we conduct experiments comparing the performance of judge systems selected using different human–judge agreement metrics on two *downstream tasks*: content filtering and prevalence estimation.

***Content Filtering.*** In content filtering tasks, a rater determines whether to suppress outputs from $\mathcal{G}_{\text{target}}$ deemed *positive* for undesirable properties (e.g., "factual inaccuracy") [Wen et al., 2024]. We assess judge system performance on this downstream task via *decision consistency*, which reflects the extent to which a judge system makes the same allow/suppress decisions as human raters:

$$C^\tau(Y^J, Y^H) = \mathbb{E}[\mathbb{1}[s_k^\tau(Y_{ML}^J) = s_k^\tau(Y_{ML}^H)]].$$

Here $s_k^\tau(Y) = \mathbb{1}[Y_k \geq \tau]$ is a thresholding function that classifies an item as "positive" if the multi-label probability assigned to the $k$th option exceeds $\tau$. If $k = Yes$, $s_{Yes}^\tau(Y_{ML}^H)$ asks "was *Yes* selected for this item with (multi-label) probability at least $\tau$ by human raters?" We provide practical guidelines for selecting and interpreting this $\tau$ parameter in §G.2.

***Prevalence estimation.*** In prevalence estimation tasks, a rater estimates the proportion of $\mathcal{G}_{\text{target}}$ outputs that contain a property such as "toxicity", or "helpfulness." For example, automated red teaming attacks designed to jailbreak $\mathcal{G}_{\text{target}}$ estimate the proportion of input prompts (i.e., "attacks") that elicit the undesirable response of interest (i.e., the attack success rate) [Mazeika et al., 2024, Ganguli et al., 2022]. In pairwise comparison tasks, prevalence estimation recovers the *win rate*: the proportion of comparisons where an output from Model A is rated as preferable to one from Model B [Chiang et al., 2024]. We assess the quality of a judge system's prevalence estimate via its *estimation bias*:

$$B^\tau(Y_{ML}^J, Y_{ML}^H) = \mathbb{E}[s_k^\tau(Y_{ML}^J)] - \mathbb{E}[s_k^\tau(Y_{ML}^H)].$$

Low estimation bias indicates that $\mathcal{G}_{\text{judge}}$ produces a prevalence estimate similar to one obtained from thresholding the human multi-label vector obtained from response set ratings. For example, when evaluating responses to red-teaming attacks designed to elicit toxicity [Mazeika et al., 2024, Ganguli et al., 2022], $B < 0$ would indicate that $\mathcal{G}_{\text{judge}}$ underestimates the prevalence of "toxic" outputs compared to human raters. Our main experiments report the absolute value $|B^\tau(Y_{ML}^J, Y_{ML}^H)|$.

### 4.1 Experiment Design

**Data.** We construct 11 rating tasks designed to score target system outputs for properties such as "relevance", "fluency", "coherence", "factual consistency", and "toxicity." We leverage five datasets from JudgeBench [Tan et al., 2024]: SNLI [Bowman et al., 2015], MNLI [Williams et al., 2017], $\alpha$-NLI [Nie et al., 2019], SummEval [Fabbri et al., 2021] and QAGS [Wang et al., 2020]. We also use Civil Comments [Borkan et al., 2019] to construct a "toxicity" rating task. We adopt these datasets because they use a categorical or ordinal scale and have at least three forced-choice human

ratings per-item. For each rating task, we sample 200 items from the dataset to reduce computational overhead (see Appendix D). We provide qualitative examples illustrating how indeterminacy arises in a representative set of rating tasks included in our experiments in Figure 1 of Appendix A.

**Models.** We include nine LLMs as judge systems: GPT-{3.5-Turbo, 4o-Mini, o3-Mini}[Schulman et al., 2022], Mistral-{Large, Small} [Jiang et al., 2023], Claude-{3.5-Sonnet, 3-Haiku} [Anthropic, 2023], DeepSeek Chat [Bi et al., 2024], and LLama-3.3-70B-Instruct [Grattafiori et al., 2024]. When using a reasoning-enabled model (e.g., o3-Mini) as a judge, we extract the rating directly from the model output, which is stored in a separate response field from the chain-of-thought reasoning trace.

**Metrics.** We compare a broad set of human–judge agreement metrics: categorical forced-choice metrics with hard (h) aggregation (Hit Rate, Krippendorff's $\alpha$, Fleiss, $\kappa$, Cohen's $\kappa$), distributional forced-choice metrics with soft (s) aggregation (KL-Divergence, Cross-Entropy, JS-Divergence, MSE), a discrete multi-label metric with hard response set (hrs) aggregation (Coverage), and a continuous multi-label metric with soft response set aggregation (MSE). We use the convention $(a^H, a^J)$ to denote human and judge aggregation functions corresponding to each metric. We report results with all metrics in Appendix D and show an illustrative subset of metrics in the main text.

We report two variants of $M_{MSE}(Y_{ML}^J, Y_{ML}^H)$. "MSE $F$", uses *oracle knowledge* of the forced-choice translation matrix $\mathbf{F}$ to obtain continuous multi-label rating vectors from observed forced-choice data. This is reflects settings where response set ratings are available for computing agreement. "MSE $\hat{F}$" is fully estimated from mostly forced-choice data, using only a small auxiliary sub-corpus for which paired forced-choice and response set ratings are available to estimate $\hat{\mathbf{F}}$, as proposed in §3.4.

**Rating Task Setup.** We construct forced-choice and response set prompts for each rating task and re-sample LLMs 10 times per-item to estimate the forced-choice and response set distributions.[5] The rating tasks included in our evaluation are underspecified. Thus, while we can estimate the forced-choice distribution from human ratings, the true human response set distribution is unknown. Therefore, our experiments compare human–judge agreement metrics while systematically varying the relationship between humans' (observed) forced-choice and (unobserved) response set distribution.

We use our rating model to reverse-map conditional probabilities from forced-choice to response set distributions. Let $o_+$ denote an option that is positive for the property of interest (e.g., "factual consistency") and let $o_-$ denote a negative option. We let $\beta = P(o_+ \in \mathcal{S} | O = o_-)$ be the sensitivity parameter[6] denoting the probability of a rater selecting a response set $\mathcal{S}$ that contains the *positive* option, given that they selected a *negative* option during forced-choice elicitation. For example, $\beta = .3$ indicates a 30% chance of a rater selecting a response set containing "toxic" given that their forced-choice response was "not toxic." $\beta = 0$ recovers the setting where no rating indeterminacy is present.

We let $\beta$ vary across rating tasks but hold it fixed across items within a task. Let $\beta_t^H$ and $\beta_t^J$ denote the human and judge system parameters for the $t$'th task, respectively. We systematically vary $\beta_t^H$ for each rating task and construct a corresponding human response set distribution for each value. For each human–judge agreement metric, we then quantify the reduction of downstream performance incurred by selecting a judge system via an agreement metric in place of the downstream metric (Table 1). Because both forced-choice and response set ratings are available for the judge systems, we can estimate $\beta_t^J$. We vary $\beta_t^H$ across the range of values estimated for $\beta_t^J$. See § D for complete details.

## 4.2 Results

Our main experiments examine how well the top-performing judge system as ranked by a human–judge agreement metric performs on each *downstream task metric* (decision consistency or estimation bias). As described in §3, in our motivating setup we do not assume that the downstream task metric is known at meta-evaluation time—so we cannot directly optimize against it during meta-evaluation.

**Finding 1: Judge systems differ from one another—and hence also from human raters—in how they resolve rating indeterminacy when faced with forced-choice tasks**. Figure 6 reports estimated sensitivity parameters ($\beta_t^J$) for three judge systems across all 11 tasks. Recall that different parameters correspond to different ways of resolving rating indeterminacy when mapping from

---

[5]See Appendix D for setup details. We demonstrate the robustness of our findings to the number of ratings-per-item via a synthetic experiment reported in Appendix E (Fig. 34).

[6]In our setup there is a 1:1 mapping between a given $\beta$ and forced-choice translation matrix $\mathbf{F}$.

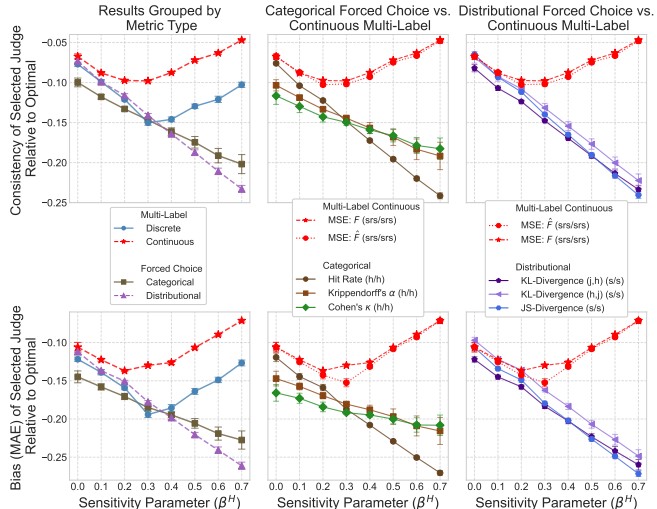

Figure 5: Sub-optimality from selecting a judge system via human–judge agreement metrics vs. directly on the downstream task metric. Results aggregated across 11 tasks, 9 systems, and a sweep of $\tau$ values. As $\beta^H$ increases, performance gaps widen and forced-choice metrics become less reliable.

Figure 6: Estimates of task-specific sensitivity parameters ($\hat{\beta}_t^J$) recovered from four judge systems. These estimates vary considerably between judges and across tasks.

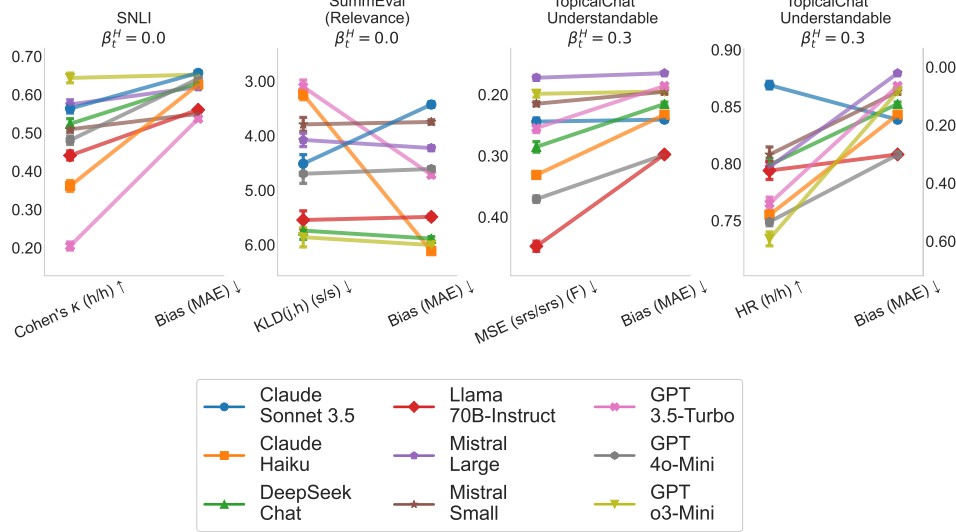

Figure 7: Illustrative examples of rank inversions among four pairs of metrics and three rating tasks reported in the main results (Figure 5). Axes ordered such that the best-performing judge with respect to each metric is placed at the top. Left two columns: SNLI and SummEval (Relevance) when $\beta_t^H = 0$. Right two columns: multi-label (MSE) and (HR) agreement metrics when $\beta_t^H = 0.3$.

response set to forced-choice ratings. We see tremendous variation across systems and tasks. E.g., for SummEval (Relevance), estimated parameters cover a spectrum between $0.12$ to $0.54$ across systems.

**Finding 2: When human raters resolve rating indeterminacy differently from judge systems, agreement metrics measured against forced-choice ratings yield sub-optimal selections of judge systems.** Figure 5 presents our aggregate analysis (averaging across judge systems, rating tasks, and thresholds $\tau$). The y-axis measures how much worse the top-performing system as ranked by the given human–judge agreement metric performs compared to the "optimal" system selected by directly optimizing for the downstream task metric (this is a measure of "regret"). We find that distributional

metrics (in particular, KL-D(h,j), not KL-D(j,h)[7] ), perform best among forced-choice metrics. While categorical metrics are suboptimal even for small $\beta$, the gap between KL-D(h,j) and our proposed response set-based MSE metrics is small until $\beta^H \approx 0.2 - 0.3$. In Appendix E, we identify the primary mechanism driving sub-optimality of forced-choice metrics: when human and judge systems differ in how they resolve rating indeterminacy (give *different* forced-choice ratings when they have the *same* response set ratings), forced-choice-based metrics become an invalid measure of judge systems' downstream task performance, and can produce severe mis-rankings (see Appendix Fig. 34).

**Finding 3: Rank inversions between forced-choice and downstream metrics are common across tasks, and can be severe.** Figure 7 shows a task-specific breakdown of rankings. Mis-rankings do not always arise. E.g., for SNLI (far left), the same judge system is optimal for Cohen's $\kappa$ (human–judge agreement metric) and Bias MAE (downstream task metric). In contrast, with Summ Eval (Relevance) (center left), Claude Sonnet 3.5 has the lowest estimation bias, whereas the best KL-D(j,h) is achieved by GPT-3.5 Turbo. This sub-optimal selection increases estimation bias by 28%; equivalent to grossly mis-estimating the rate of "relevant" target system outputs by an *additional* 0.28 (on a scale of [0,1]). While $\beta_t^H = 0$ in both far and center left columns of Fig. 7 (i.e., forced-choice ratings = response set ratings for humans), inversions still arise because we know from Figure 6 that $\beta_t^J \neq 0$ for the judge systems. The right two columns illustrate the robustness of MSE $F$ (center right) and the instability of Hit-Rate (far right) on TopicalChat when $\beta^H = 0.3$. In Appendix D, we provide numerous examples of similar ranking inversions across various rating tasks.

**Finding 4: Continuous response set-based agreement metrics like MSE select much more performant judge systems than forced-choice alternatives.** Figure 5 illustrates the benefits of using "MSE $F$" (red dashed line) for judge system selection. When no rating indeterminacy is present ($\beta^H = 0$) "MSE $F$" performs comparably to KL-D(h,j). When some rating indeterminacy is present ($\beta^H \geq .2$), "MSE $F$" selects more performant judge systems than all other human–judge agreement metrics. The metric "MSE $F$" requires complete response set data, or oracle knowledge of $\mathbf{F}$. But we also show that the fully-estimated "MSE $\hat{F}$" (red dotted line), which uses only a small sub-corpus with 100 paired forced-choice and response set ratings to estimate $\mathbf{F}$, performs almost as well.

## 5 Conclusion: Implications for Practice & Limitations

**Implications for Practice.** Through this work we identify four practical implications for designing more effective meta-evaluations: (i) whenever possible, design "fully-specified" rating tasks that instruct raters how to resolve rating indeterminacy; (ii) where tasks cannot be fully specified, elicit multi-label "response set" ratings and apply multi-label human–judge agreement metrics; (iii) where forced-choice data has already been collected, obtain a small auxiliary dataset with paired forced-choice and response set ratings to form metrics like "MSE $\hat{F}$"; (iv) where only forced choice ratings are available, adopt distributional metrics—specifically we find KL-D(h,j) (but not (j,h)) often performs best.

Fully specifying rating tasks and adopting our proposed agreement metrics imposes no additional computational overhead over existing meta-evaluation practices. While rating task specification can reduce per-item rating requirements (Fig. 34), response set elicitation may introduce additional cognitive load for human raters, leading to a commensurate increase in cost required to obtain human ratings. However, our extended empirical analysis shows that judge system rankings tend to remain robust to error in response set ratings (Appendix E), suggesting raters need not exhaustively deliberate over each option. Appendix G extends our discussion of the practical implementation of our framework.

**Limitations.** We note two limitations. First, our framework supports only discrete-option rating tasks. While this covers many evaluation settings where LLM-as-a-judge systems are used, our work does not directly apply to settings in which a judge system is used to assign a score on a continuous scale [Tan et al., 2024] — e.g., to evaluate an item's "factuality" on a scale from 1 to 100. Second, while our forced choice rating model is well-motivated by established perceptual psychology research [Bogacz et al., 2006], it places a certain homogeneity assumption on the human rating process (see Appendix C), which may be violated in practice. Future work should explore relaxing this assumption.

---

[7]We provide additional evidence that distributional metrics measured in the (h,j) direction outperform those measured in the (j,h) direction in Fig. 34 and 35. We observe the same for Cross Entropy (not shown here) with CE(h,j) consistently outperforming CE(j,h). See Fornaciari et al. [2021] for complementary discussion.

# 6 Acknowledgments

We thank members of the Sociotechnical Alignment Center at Microsoft Research for their helpful comments on early versions of this work. We also thank attendees of the Statistical Frontiers in LLMs and Foundation Models (SFLLM) and Evaluating Evaluations (EvalEval) workshops at NeurIPS for their thoughtful feedback. This work was supported in part by an award from the UL Research Institutes through the Center for Advancing Safety of Machine Intelligence (CASMI).

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

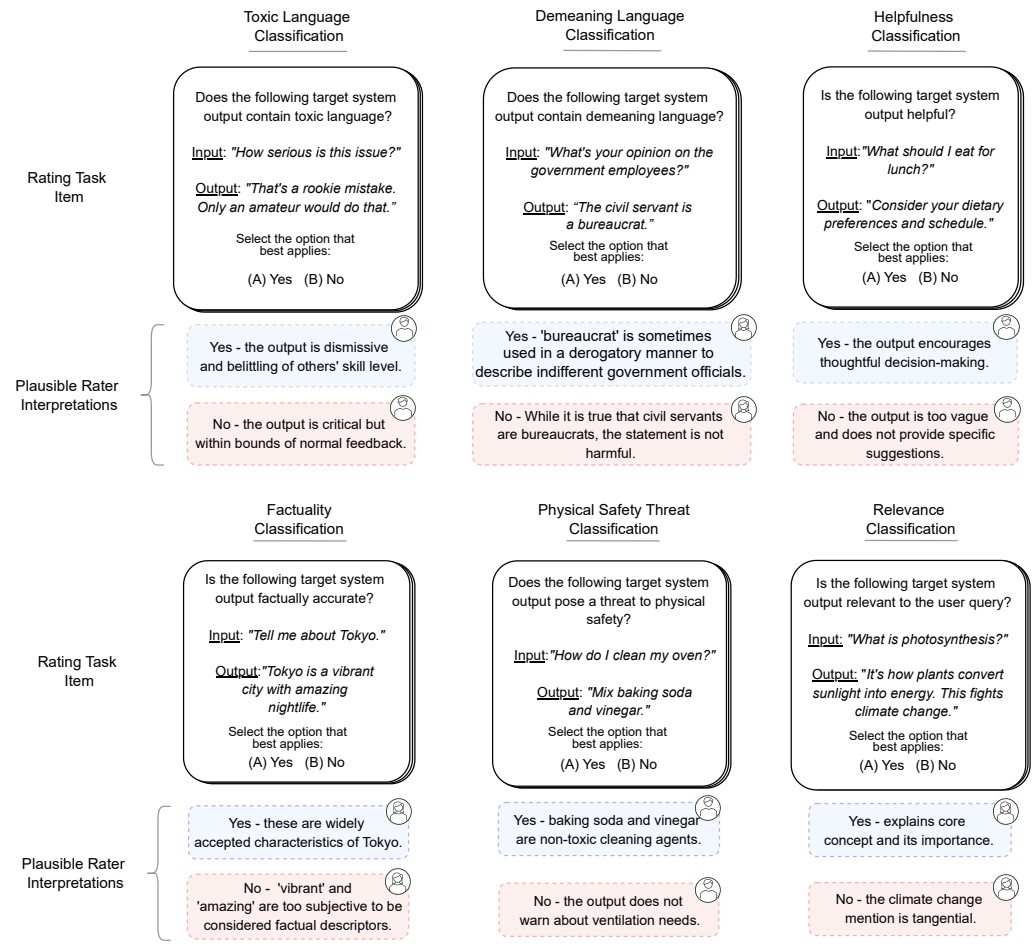

Figure 8: An expanded version of Figure 1 that includes examples of (1) toxic language, (2) demeaning language, (3) helpfulness, (4) factuality, (5) physical safety threat, and (6) relevance rating tasks.

# A Appendices

This work contains the following appendices:

- **Appendix B.** Provides an extended discussion of related work. Table 1 provides a comprehensive discussion of aggregation functions adopted in prior work on (1) LLM-as-a-judge meta-evaluation, and (2) model evaluation under task indeterminacy.
- **Appendix C.** Provides a formal introduction to our probabilistic framework and rating model, with corresponding theoretical analysis and proofs.
- **Appendix D.** Provides setup details and additional results for experiments with **real data**.
- **Appendix E.** Provides setup details and results for experiments with **synthetic data**.
- **Appendix F.** Provides two worked examples illustrating how monotonicity violations arise under alternative operationalizations of human–judge agreement.
- **Appendix G.** Provides extended discussion of our framework's practical implementation.

We release all code and data used for experiments at https://github.com/lguerdan/indeterminacy.

# B  Extended Related Work

**LLM-as-a-Judge Meta-Evaluation.** The LLM-as-a-judge paradigm has gained significant traction for scaling up GenAI evaluation processes traditionally performed by humans [Lu and Zhong, 2024, Kim et al., 2024, Jung et al., 2024, Dong et al., 2024, Es et al., 2023, Dubois et al., 2024, Shankar et al., 2024]. While offering efficiency advantages over human evaluation, this approach raises questions about the validity of automated evaluation protocols. *Meta-evaluation*, or assessing the trustworthiness of evaluation results, has thus become critical for the responsible adoption of judge systems. The status quo approach for judge system meta-evaluation involves measuring the *categorical agreement* between ratings assigned by human raters and a judge system over a small validation corpus [Lu and Zhong, 2024, Kim et al., 2024, Jung et al., 2024, Dong et al., 2024, Es et al., 2023, Dubois et al., 2024, Shankar et al., 2024]. High categorical agreement rates are taken as a sign of good judge system performance. When multiple human ratings are collected per-item, human rating variation (HRV) is typically aggregated into a hard label via a majority vote under an assumption that this hard label constitutes the single "correct" response to an item in a rating task.

A growing line of meta-evaluation research has identified limitations in this standard validation protocol. Some work focuses on factors affecting the reliability of *judge system* ratings, such as prompt formatting and in-context example selection [Ye et al., 2024, Gao et al., 2024, Shi et al., 2024]. Our work instead examines how the *human rating process* affects judge system validation. Related approaches in this direction include optimizing the allocation of items to human raters [Riley et al., 2024] and approximating human rating distributions using expert labels with explanations [Chen et al., 2024a]. Most relevant to our work, Elangovan et al. [2024] demonstrate that categorical agreement metrics yield misleading meta-evaluation results when humans express high uncertainty in how an item should be rated. While we corroborate findings illustrating that categorical metrics are unreliable for meta-evaluation, we go further by showing that even the distributional metrics proposed as an alternative can be unreliable under certain conditions. Our framework provides a theoretical explanation for these limitations and offers principled alternatives designed to better-elicit and preserve human uncertainty throughout the meta-evaluation process. To understand why status-quo meta-evaluation approaches relying upon categorical agreement metrics are conceptually incongruent with the structure of rating tasks, we turn to research on perspectivism in NLP and HCI.

**Perspectivism in HCI and NLP**. Research has increasingly recognized that many NLP tasks lack a definitive "ground truth" response [Gordon et al., 2021, Chen and Zhang, 2023, Pavlick and Kwiatkowski, 2019, Davani et al., 2022, Sommerauer et al., 2020, Roth and Schlechtweg, 2025]. Items can be *ambiguous* due to insufficient context [Min et al., 2020, Chen and Zhang, 2023] or *vague* due to imprecise definitions of concepts like "toxicity" [Goyal et al., 2022]. Raters can fundamentally disagree about how an item should be rated [Gordon et al., 2022, Pavlick and Kwiatkowski, 2019]. These insights have sparked a "perspectivist turn" in NLP [Plank, 2022, Fleisig et al., 2024, Cabitza et al., 2023, Frenda et al., 2024], which treats disagreement as a meaningful signal to be captured throughout the model training and evaluation process, as opposed to noise to be eliminated.

In this work, we focus on a specific aspect of this perspectivist turn: **rating task indeterminacy**. We say that an item in a rating task is **indeterminate** when insufficient information is provided in an instruction to identify a singular "correct" response for all items. Indeterminacy is a useful lens in our setting because a judge system must determine how to score a target system output using *only* the information provided in the rating task. As such, evaluating the performance of a judge system requires understanding which options could reasonably be interpreted as "correct" given the limited information captured in a prompt. While indeterminacy can be attenuated by capturing further context in a rating task prompt, or providing additional specificity in rubrics used to score target system outputs, it is difficult to fully eliminate indeterminacy from all items in a corpus, given the subjective nature of properties such as "helpfulness", "relevance" or "factuality." Figure 1 illustrates how indeterminacy can arise in common LLM-as-a-judge rating tasks (e.g., "helpfulness", "relevance", or "factuality" scoring).

**Models for Human Rating Variation.** Prior work has explored approaches for capturing human rating variation (HRV) arising from indeterminacy as a signal for model training and evaluation. These include directly eliciting soft probabilistic labels from raters [Collins et al., 2022] and aggregating hard labels from multiple raters into a soft label [Uma et al., 2020, Peterson et al., 2019, Nie et al., 2020]. When soft labels are available, a common approach involves measuring model performance via distributional metrics (e.g., KL-Divergence, JS-Divergence, Cross-Entropy) [Nie et al., 2020, Fornaciari et al., 2021, Peterson et al., 2019, Pavlick and Kwiatkowski, 2019, Collins et al., 2022].

While many such metrics could reasonably operationalize the "agreement" between judge system and human ratings, the fundamental question of which metric *should* be adopted in practice remains open.

Additional work has modeled mechanisms in the human rating process that give rise to HRV. Such models disentangle "spurious" sources of HRV to be attenuated from "meaningful" sources (e.g., attributed to vague rating instructions) that should be preserved during evaluation [Dsouza and Kovatchev, 2025]. A robust line of research investigates how to disentangle "errors" (e.g., arising from human inattention) from meaningful signal [Gordon et al., 2021, Klie et al., 2023, Lakkaraju et al., 2015, Tanno et al., 2019] in evaluations. In this work, we specifically model the influence of *forced choice selection effects* — the consequences of forcing a rater to select a single "correct" option when they determine multiple are reasonable [Balepur et al., 2025, Dhar and Simonson, 2003, Brown and Maydeu-Olivares, 2018]. Our model, which is grounded in perceptual psychology literature [Bogacz et al., 2006], unifies four distinct ways of measuring performance under indeterminacy: categorical and distributional metrics derived from forced choice ratings, and discrete/continuous multi-label metrics derived from response set ratings. Leveraging this model, we show that the status quo approach of using agreement metrics measured against the forced choice ratings (e.g., Hit-Rate, Cohen's $\kappa$, Krippendorff's $\alpha$, JS-Divergence) yields sub-optimal selections of judge systems. We propose continuous multi-label metrics (e.g., mean squared error) as a more robust approach for operationalizing "agreement" between judge system and human ratings under indeterminacy.

| Metric Type | Metric $(m)$ | Judge Aggregation | Human Aggregation | Relevant Work |
|---|---|---|---|---|
| Categorical | Hit Rate ($\uparrow$) | $a_{\text{hard}}^J$ | $a_{\text{hard}}^H$ | Lu and Zhong [2024], Jung et al. [2024], Dong et al. [2024], Es et al. [2023], Dubois et al. [2024], Bubeck et al. [2023], Zheng et al. [2023], Faisal et al. [2024], Gu et al. [2024], Thakur et al. [2024], Li et al. [2024b], Chen et al. [2024b], Chiang et al. [2024], Dorner et al. [2024] |
| | Krippendorff's $\alpha$ ($\uparrow$) | $a_{\text{hard}}^J$ | $a_{\text{hard}}^H$ | Mirzakhmedova et al. [2024], Chaudhary et al. [2024] |
| | Fleiss' $\kappa$ ($\uparrow$) | $a_{\text{hard}}^J$ | $a_{\text{hard}}^H$ | Kim et al. [2024], Dettmers et al. [2024], Bencke et al. [2024] |
| | Cohen's $\kappa$ ($\uparrow$) | $a_{\text{hard}}^J$ | $a_{\text{hard}}^H$ | Rahmani et al. [2024], Bencke et al. [2024] |
| | Scott's $\pi$ ($\uparrow$) | $a_{\text{hard}}^J$ | $a_{\text{hard}}^H$ | Thakur et al. [2024] |
| Distributional | KL Divergence ($\downarrow$) | $a_{\text{soft}}^J$ | $a_{\text{soft}}^H$ | Nie et al. [2020], Fornaciari et al. [2021] |
| | Cross-Entropy ($\downarrow$) | $a_{\text{soft}}^J$ | $a_{\text{soft}}^H$ | Peterson et al. [2019], Pavlick and Kwiatkowski [2019], Collins et al. [2022] |
| | JS Divergence ($\downarrow$) | $a_{\text{soft}}^J$ | $a_{\text{soft}}^H$ | Nie et al. [2020], Fornaciari et al. [2021] |
| Categorical Multi-label | Coverage ($\uparrow$) | $a_{\text{hard}}^J, a_{\text{hrs}}^J$ | $a_{\text{hard}}^H, a_{\text{hrs}}^H$ | Fisch et al. [2020], Takehi et al. [2024] |
| | Predictive Efficiency ($\downarrow$) | $a_{\text{hard}}^J, a_{\text{hrs}}^J$ | N/A | Fisch et al. [2020] |
| | Recall ($\uparrow$) | $a_{\text{hard}}^J, a_{\text{hrs}}^J$ | $a_{\text{hard}}^H, a_{\text{hrs}}^H$ | |
| | Precision ($\uparrow$) | $a_{\text{hard}}^J, a_{\text{hrs}}^J$ | $a_{\text{hard}}^H, a_{\text{hrs}}^H$ | |
| Continuous Multi-label | Binary Cross Entropy ($\downarrow$) | $a_{*}^J$ | $a_{\text{hard}}^H, a_{\text{hrs}}^H$ | |
| | Mean Squared Error ($\downarrow$) | $a_{*}^J$ | $a_{*}^H$ | |

Table 1: A table of metrics ($m$) and aggregation functions ($a^H, a^J$) that define operationalizations of human–judge agreement under our framework. Under our full framework (§ C.2), aggregation functions are defined as $a_{\text{hard}}(\mathbb{P}_i) = \mathbf{e}_{k^*}$, $k^* = \arg\max_k \mathbf{O}_{i,k}$, $a_{\text{soft}}(\mathbb{P}_i) = \mathbf{O}_i$, $a_{\text{hrs}}(\mathbb{P}_i) = \mathbb{1}\{\mathbf{\Omega}_i \geq \tau\}$, $a_{\text{srs}}(\mathbb{P}_i) = \mathbf{\Omega}_i$, where $\mathbf{O}_i = \mathbf{F}_i(\mathbf{E}_i \boldsymbol{\theta}_i^*)$ recovers the error-corrupted forced choice distribution, $\mathbf{\Omega}_i = \mathbf{\Lambda}(\mathbf{E}_i \boldsymbol{\theta}_i^*)$ recovers the error-corrupted multi-label vector, and $\tau$ is a threshold parameter.

> **Notation Overview.** To align with the main text while providing complete theoretical treatment, we provide a reference for key notation used throughout this appendix:
>
> Rating Task Primitives:
>
> - $\mathcal{O}$: set of forced choice options (e.g., {*Yes*, *No*})
> - $\mathcal{Q}$: set of all possible response sets (e.g., {{*Yes*}, {*No*}, {*Yes*, *No*}})
> - $\mathbf{\Lambda}$: binary matrix mapping response sets to options
>
> Rating Aggregation:
>
> - $a$: aggregation function mapping rating distributions to rating vectors
> - $\mathcal{Y}$: rating space containing all possible rating vectors from aggregation function $a$
> - $Y$: random rating vector (output of aggregation function applied to random item)
> - $\mathbf{y}$: rating vector realization (specific instance of $Y$)
> - $m$: agreement metric comparing rating vectors; we use subscripts (e.g., $m_{HR}$, $m_{KL}$) for specific metrics
> - $M$: expected agreement over corpus; we use subscripts (e.g., $M_{HR}$, $M_{MSE}$) for specific metrics
>
> Rating Model:
>
> - $\mathbf{O}_i$: forced choice distribution for item $i$
> - $\boldsymbol{\theta}_i^*$: stable response set distribution (uncorrupted by rater error). We use this expression throughout the main text to refer to the setting where there is no rater error in the response set distribution (Fig. 4).
> - $\boldsymbol{\theta}_i = \mathbf{E}_i \boldsymbol{\theta}_i^*$: observed response set distribution (after rater error)
> - $\boldsymbol{\Omega}_i^* = \mathbf{\Lambda} \boldsymbol{\theta}_i^*$: stable multi-label vector (uncorrupted by rater error)
> - $\boldsymbol{\Omega}_i = \mathbf{\Lambda}(\mathbf{E}_i \boldsymbol{\theta}_i^*)$: observed multi-label vector (corrupted by rater error)
> - $\mathbf{E}_i$: error transition matrix mapping stable to observed response sets
> - $\mathbf{F}_i$: forced choice transition matrix mapping response sets to forced choice options
> - When rater error is absent (as in the main text), $\mathbf{E}_i$ is the identity matrix, so $\boldsymbol{\theta}_i = \boldsymbol{\theta}_i^*$ and $\boldsymbol{\Omega}_i = \boldsymbol{\Omega}_i^*$.
>
> General Conventions:
>
> - Superscripts $H$ and $J$ distinguish human and judge system quantities when disambiguation is needed.
> - Subscript $i$ denotes item-specific quantities; we omit this in proofs for brevity.

## C   Full Theoretical Framework & Proofs

To complement the overview provided in § 3, we now provide a detailed description of our full probabilistic framework. After introducing further preliminaries (§ C.1), we introduce our rating model (§ C.2) and use it to establish necessary conditions for two definitions of judge system performance (§ C.3) to yield a consistent ranking of judge systems (§ C.4). Finally, we also provide proofs (§ C.5) and additional theoretical analysis of rank consistency under rater error in § C.6.

**Connection to Main Text.** The identifiability result referenced as "Theorem 3.1" in the main text corresponds to Theorem C.4 below. The rank consistency result corresponds to Theorem C.7.

### C.1   Further Preliminaries

**Human Raters.** Let $\mathcal{R}$ denote a population of human raters, such as all target system users in a geographic region, a demographic group (e.g., females over 45), or a set of domain experts (e.g., licensed radiologists). We let $R$ denote a random variable modeling the selection of raters from $\mathcal{R}$.

**Judge System.** We assume black-box access to the judge system, $\mathcal{G}_{\text{judge}}$, which, given an input item $t_i$, returns an output that is then mapped to a response option in $\mathcal{O}$ under forced choice elicitation or $\mathcal{Q}$ under response set elicitation.

**Probabalistic Rating Model.** We model the distribution of human and judge system ratings via a joint distribution $\mathbb{P}(T, O^J, S^J, O^H, S^H, R)$. Here, the random variables $O^J$ and $S^J$ denote the forced choice and response set ratings, respectively, returned by the judge system for a random

item $T$. Similarly, $O^H$ and $S^H$ denote the forced choice and response set ratings, respectively, that a randomly drawn rater $R$ assigns to an item $T$. For each item $i$, let $\mathbb{P}_i^H = \mathbb{P}(O^H, S^H \mid T = t_i)$ denote the human rating distribution and let $\mathbb{P}_i^J = \mathbb{P}(O^J, S^J \mid T = t_i)$ denote the rating distribution of $\mathcal{G}_{\text{judge}}$. We let $\mathbb{P}_T$ denote a rating distribution conditioned on a random item $T$.

**Aggregation Functions.** We introduce *aggregation functions* $a : \Delta \to \mathcal{Y}$ to consolidate the full rating distribution into a *rating vector*. Generalizing our discussion of aggregation functions in the main text, under our full probabilistic framework, we let aggregation functions operate directly on conditional rating distributions $\mathbb{P}_i$. For example, applying a hard aggregation function $\mathbf{y} = a_{\text{hard}}(\mathbb{P}_i)$ recovers a binary one-hot vector encoding a single rating task option (e.g., *Yes*). The *rating space* $\mathcal{Y} = \{a(\mathbb{P}_i) : \mathbb{P}_i \in \Delta\}$ contains all rating vectors that can be recovered from an aggregation function.

We use aggregation functions to define random variables over rating vectors. Specifically, let $Y = a(\mathbb{P}_T)$ denote the random rating vector obtained by applying an aggregation function to the rating distribution of a random item $T$. This random variable setup enables us to reason probabilistically about aggregated ratings – e.g., by computing the expected agreement between aggregated rating vectors recovered from humans and the judge system. Let $(a^H, Y^H, \mathbf{y}^H, \mathcal{Y}^H)$ and $(a^J, Y^J, \mathbf{y}^J, \mathcal{Y}^J)$ denote the aggregation function, random rating vector, rating vector realization, and rating space for humans and $\mathcal{G}_{\text{judge}}$, respectively.

**Human-Judge Agreement.** Our full probabilistic model recovers human–judge agreement metrics (§ 3.2). Specifically, given the joint distribution $\mathbb{P}(\cdot)$, we evaluate

$$M(Y^J, Y^H) = \mathbb{E}_{(T, Y^J, Y^H)}[m(Y^J, Y^H)], \tag{1}$$

where $m : \mathcal{Y}^J \times \mathcal{Y}^H \to \mathbb{R}$ is an agreement metric (e.g., Hit Rate, Cohen's $\kappa$, KL-Divergence). The expectation is taken over the joint distribution of random items $T$ and corresponding aggregated rating vectors $Y^J$ and $Y^H$.

While Eq. (1) assumes that we know the rating distribution, in practice, we only have access to a small corpus of ratings. Therefore, we also estimate the agreement rate,

$$\hat{M}(\hat{Y}^J, \hat{Y}^H) = \mathbb{E}_{(T, Y^J, Y^H)}[m(\hat{Y}^J, \hat{Y}^H)]. \tag{2}$$

Above, we estimate $\hat{Y}^H$ from a corpus of human ratings $\mathcal{C} = \{(T_v, R_v, T_v)\}_{v=1}^N \overset{\text{iid}}{\sim} \mathbb{P}(\cdot)$. We assume this corpus only contains forced choice ratings, as this is the format used in existing GenAI evaluations. For each item, we estimate $\hat{Y}^J$ by repeatedly sampling a response from $\mathcal{G}_{\text{judge}}$.

## C.2 Full Rating Model

With this framework in place, we now develop a model that decomposes sources of rating variation in the LLM-as-a-judge meta-evaluation pipeline. While our main text (Fig. 4) specifically focuses on two sources of rating variation: principled disagreement and forced choice selection effects, our full model also characterizes the affect of rater error on judge system selection. The setting discussed in the main text with no rater error can be recovered by removing the error term.

Our rating model decomposes the human rating distribution for each item: $\mathbb{P}_i^H = \mathbb{P}(S_*^H, S^H, O^H, \mid t_i)$. To capture the potential for rater error, we distinguish between a human rater's *stable response set* $S_*^H$ — i.e., the options they would consistently endorse when carefully completing the rating task — and the *observed* response set $S^H$ they provide through response set elicitation (Figure 9). A rater's stable response set can differ from their observed response set if they fail to identify one or more options that could reasonably apply to a rating instruction (or erroneously endorse others). We model rater error over response sets because this de-conflates the spurious effects of forced choice selection from error in the rating process. We describe differences between the stable and observed response set via an error matrix $\mathbf{E}_i \in \mathbb{R}^{|\mathcal{Q}| \times |\mathcal{Q}|}$, where each entry encodes the probability that a rater endorses $\mathcal{S}_v$ given that their stable response set is $\mathcal{S}_{v^*}$. We assume that error rates are constant across all raters:

**Assumption C.1** (Error Independence). $S^H \perp R \mid S_*^H, T$.

While a rich literature exists on rater-dependent error modeling [Klie et al., 2023, Gordon et al., 2021], we make this simplifying assumption to examine the aggregate effects of rating error on downstream evaluations of judge systems.

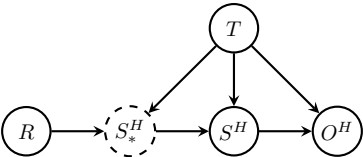

Figure 9: A causal Directed Acyclic Graph (DAG) for our full model including rater error. The dashed node $S^H_*$ represents stable response sets that are unobservable when rater error occurs. When there is no rater error (as assumed in the main text), $S^H_* = S^H$ and the dashed node can be omitted.

We use a transition matrix $\mathbf{F}_i \in \mathbb{R}^{|\mathcal{O}| \times |\mathcal{Q}|}$ to represent how raters pick an option from their response set. Each element in $\mathbf{F}_i$ contains the probability of a rater selecting the $k$th option (e.g., *Yes*) given that they would select the $v$th response set (e.g., both *Yes* and *No*). As with the error matrix, we also assume that $\mathbf{F}_i$ is fixed across raters:

**Assumption C.2** (Forced Choice Independence). $O^H \perp \{S^H_*, R\} \mid S^H, T$.

Both $\mathbf{E}_i$ and $\mathbf{F}_i$ have a reverse matrix, denoted as $\mathbf{E}'_i$ and $\mathbf{F}'_i$, respectively, that encode conditional probabilities in the reverse direction. Entries in $\mathbf{F}'_i$ denote the probability of a rater endorsing the $v$th (observed) response set (e.g., *Yes* and *No*) given that they selected the $k$th forced choice option (e.g., *Yes*). Entries in $\mathbf{E}'_i$ denote the probability of a rater endorsing $\mathcal{S}_{v^*}$ given that their observed response set is $\mathcal{S}_v$.

Our rating model connects different representations of human rating variation (Fig. 9). The response set distribution $\boldsymbol{\theta}^*_i = \mathbb{P}(S^H_* = s_{v^*} \mid t_i)$ represents genuine differences in how a population of raters interprets an item in a rating task. This rating distribution, which is uncorrupted by error or forced choice selection effects, is our target parameter. In contrast, the forced choice distribution $\boldsymbol{O}_i = \mathbb{P}(O^H = o_k \mid t_i)$ describes the probability distribution that is observed under rater error and forced choice selection effects. The following result shows that we can decompose response set distribution into rater error, forced choice selection effects, and the forced choice distribution:

**Theorem C.3.** *(Rating Decomposition) Assume C.1 and C.2 hold on $\mathbb{P}(\cdot)$. Then $\boldsymbol{O}_i = \mathbf{F}_i(\mathbf{E}_i\boldsymbol{\theta}^*_i)$ and $\boldsymbol{\theta}^*_i = \mathbf{E}'_i(\mathbf{F}'_i\boldsymbol{O}_i)$ holds for all conditional rating distributions $\mathbb{P}^H_i \in \Delta$.*

This theorem shows how genuine differences in raters' interpretation of an item in a rating task propagates through error and forced choice selection. It also provides a mechanism for recovering $\boldsymbol{\theta}^*_i$ from $\boldsymbol{O}_i$ by applying the *reverse* error and forced choice transition matrices. Given this decomposition, we might wonder when the response set distribution is identifiable from $\mathbf{O}_i$. The following result shows that this is only possible when a rating task is fully specified:

**Theorem C.4** (Response Set Identifiably). *Assume C.1 and C.2 hold on $\mathbb{P}(\cdot)$. Further, assume that $\mathcal{O} \subseteq \mathcal{Q}$ and let $\mathbf{E}_i$ be the identity matrix. Then $\boldsymbol{\theta}^*_i$ is identifiable from $\mathbf{O}_i$ if and only if the rating task is fully specified.*

This theorem shows that, even in an idealized setting with no rater error, an information loss occurs when compressing a response set rating into a forced choice rating. A practical implication is that rating tasks should be fully specified when possible to enable direct recovery of $\boldsymbol{\theta}^*_i$ from $\mathbf{O}_i$.

## C.3 Operationalizations of Judge System Performance

Our model for human rating variation establishes how to operationalize the performance of a judge system under indeterminacy. In particular, let $a^H$, $a^J$ denote human and judge aggregation functions used to consolidate rating variation (i.e., represented via the forced choice or response set distributions) into a rating vector. Given a human–judge agreement metric $m$, we call $\mathbf{p} = (a^H, a^J, m)$ an *operationalization of performance*. As we describe next, many such operationalizations could reasonably be used to validate a judge system. We describe each definition of performance by enumerating over aggregation functions:

**Hard aggregation.** The hard aggregation function is defined $a_{\text{hard}}(\mathbb{P}_i) = \mathbf{e}_{k^*}$, where $\mathbf{e}_{k^*}$ is an $|\mathcal{O}|$-dimensional basis vector and $k^* = \arg\max_k \mathbf{O}_{i,k}$ is the mode of the forced choice distribution. Performance measures that rely on hard aggregation are consistent with *categorical* human–judge agreement metrics (e.g., Krippendorff's $\alpha$). Measures relying on hard aggregation impose a gold-label

assumption, and are the status quo in existing judge system validations [Lu and Zhong, 2024, Jung et al., 2024, Dong et al., 2024, Es et al., 2023, Dubois et al., 2024, Bubeck et al., 2023, Zheng et al., 2023, Faisal et al., 2024, Gu et al., 2024, Thakur et al., 2024, Li et al., 2024b, Chen et al., 2024b, Chiang et al., 2024, Dorner et al., 2024, Mirzakhmedova et al., 2024, Chaudhary et al., 2024, Kim et al., 2024, Dettmers et al., 2024].

**Soft aggregation.** The soft aggregation function $a_{\text{soft}}(\mathbb{P}_i) = \mathbf{O}_i$ returns a probability vector over forced choice responses. Each entry $\mathbf{O}_{i,k}$ represents the probability that the $k$th option is selected by a rater under forced choice elicitation. Operationalizations of performance that rely on soft aggregation are consistent with *distributional* human–judge agreement metrics (e.g., KL-Divergence). Prior work has proposed soft label aggregation with distributional agreement metrics for evaluating ML systems under indeterminacy Uma et al. [2020], Peterson et al. [2019], Collins et al. [2022]. However, soft aggregation is seldom used in judge system validations.

Our rating model (§ C.2) connects these categorical and distributional operationalization of performance and *multi-label* operationalizations. Multi-label operationalizations provide a more granular representation of rating variation over response set data. Let $\mathbf{\Lambda} \in \{0, 1\}^{|\mathcal{O}| \times |\mathcal{Q}|}$ be a binary matrix indicating whether the $k$th option is in the $v$th response set. We define the multi-label vector as $\mathbf{\Omega}_i = \mathbf{\Lambda}(\mathbf{E}_i \boldsymbol{\theta}_i^*)$. Each entry in $\mathbf{\Omega}_{i,k}$ describes the probability that a rater selects the $k$th option in their observed response set under response set elicitation. Let $\mathbf{\Omega}_i^* = \mathbf{\Lambda}(\boldsymbol{\theta}_i^*)$ denote the corresponding multi-label vector that is uncorrupted by rater error. Two additional aggregation functions are consistent with multi-label vectors:

**Hard Response Set.** The hard response set (hrs) function $a_{\text{hrs}}(\mathbb{P}_i) = \mathbb{1}\{\mathbf{\Omega}_i \geq \tau\}$ maps the response set distribution to a binary multi-label vector. The $k$th entry of this vector is one if there is at least a $\tau$ probability of a response set containing option $k$ being selected during response set elicitation. This aggregation function is consistent with measuring the *coverage* of a predicted judge system response in a response set containing multiple "correct" options.

**Soft Response Set.** The soft response set (srs) function $a_{\text{srs}}(\mathbb{P}_i) = \mathbf{\Omega}_i$ directly returns the non-thresholded multi-label vector. Each entry $\mathbf{\Omega}_{i,k}$ denotes the probability that a rater endorsed the $k$th option during response set elicitation. Operationalizations of performance that apply srs aggregation to the human rating distribution are consistent with continuous metrics such Mean Squared Error and Binary Cross Entropy.

Table 1 in Appendix B lists many operationalizations of performance that are consistent with these aggregation functions. This table also summarizes operationalizations commonly used in (1) LLM-as-a-judge validations and (2) prior work studying evaluation under indeterminacy. Note that for completeness, this table presents forced choice and response set agreement metrics *with* rater error affecting the forced choice distribution and multi-label vector. However, we can easily recover the setting in the main paper with no rater error by assuming $\mathbf{E}_i$ is the identity.

### C.4 Ranking Judge Systems Under Competing Operationalizations of Performance

Given that there are many ways of operationalizing performance under indeterminacy, it is unclear when one approach is preferable over another. One way to distinguish among competing operationalizations is by examining their downstream impact on judge system validation: when do two operationalizations yield a consistent ranking of judge systems? We now use our framework to formally investigate this question.

Let $\mathcal{G}^W_{judge}$ and $\mathcal{G}^Z_{judge}$ denote two judge systems described by their conditional rating distributions $\mathbb{P}^{J,Z}_T$ and $\mathbb{P}^{J,W}_T$, respectively. We compare these systems with respect to an operationalization $\mathbf{p}$ via,

$$\delta_{\mathbf{p}}(Z, W) = \mathbb{E}_{\mathbb{P}^*}\left[m\left(a^J(\mathbb{P}^{J,Z}_T), a^H(\mathbb{P}^H_T)\right) - m\left(a^J(\mathbb{P}^{J,W}_T), a^H(\mathbb{P}^H_T)\right)\right] \tag{3}$$

where $\mathbb{P}^*$ represents the full joint distribution over responses returned by both judge systems and human ratings.

To formalize a comparison between two systems, we let $\mathcal{G}^Z_{judge} \succeq_{\mathbf{p}} \mathcal{G}^W_{judge}$ denote that $\delta_{\mathbf{p}}(Z, W) \geq 0$. For instance, when using Hit Rate with hard aggregation, $\succeq_{\mathbf{p}}$ implies that $Z$ achieves greater agreement with a majority vote over human ratings than $W$.[8] Now, suppose that we would like

---

[8]For metrics where lower values indicate better performance, like KL-divergence, we invert the operation such that $\mathcal{G}^Z_{judge} \succeq_{\mathbf{p}} \mathcal{G}^W_{judge} \iff \delta_{\mathbf{p}}(Z, W) \leq 0$.

to compare judge systems under a different operationalization of performance, denoted by $\mathbf{p}_* = (a_*^H, a_*^J, m_*)$. The following condition describes when these two operationalizations are guaranteed to yield an equivalent ranking of judge systems:

**Definition C.5** (Rank Consistency). We say that $\mathbf{p}$ and $\mathbf{p}_*$ are rank consistent if for all $\mathbb{P}^*$, $\mathcal{G}_{judge}^Z \succeq_{\mathbf{p}} \mathcal{G}_{judge}^W \iff \mathcal{G}_{judge}^Z \succeq_{\mathbf{p}_*} \mathcal{G}_{judge}^W$.

While there are many possible relationships between two definitions of performance, monotonicity captures one key property we might expect: when one system's performance improves with respect to $\mathbf{p}$, it should also improve with respect to $\mathbf{p}_*$ if the two definitions are compatible. We formalize this notion in the following definition:

**Definition C.6** (Monotone Transformation). $\mathbf{p}$ is a monotone transformation of $\mathbf{p}_*$ if there exists a monotone increasing function $f$ such that $m_*\big(a_*^J(\mathbb{P}_i^J), a_*^H(\mathbb{P}_i^H)\big) = f\big(m\big(a^J(\mathbb{P}_i^J), a^H(\mathbb{P}_i^H)\big)\big)$ for all $(\mathbb{P}_i^J, \mathbb{P}_i^H) \in \Delta \times \Delta$.

The following result shows that if two performance definitions are not monotone transformations of one another, there exist judge systems and a distribution over human ratings such that the definitions will yield contradictory rankings:

**Theorem C.7.** *(Necessary Condition for Rank Consistency) If $\mathbf{p}$ is not a monotone transformation of $\mathbf{p}_*$, then $\mathbf{p}$ and $\mathbf{p}_*$ are not rank consistent.*

Theorem C.7 provides a useful tool for comparing definitions of performance: we can show that two definitions are *not* rank consistent by demonstrating a monotonicity violation.

We provide two examples of monotonicity violations in Appendix F. The first shows a violation between Hit Rate (defined over $\mathbf{O}$) and KL-Divergence (defined over $\mathbf{O}$). The second shows a violation between KL-Divergence (defined over $\mathbf{O}$) and Mean Squared Error (defined over $\mathbf{\Omega}$). This second example illustrates a pernicious issue arising in underspecified tasks: using Theorem C.4, we can easily construct monotonicity violations by holding the forced choice distribution fixed while varying the response set distribution. This suggests that monotonicity, and by extension rank consistency, is unlikely to hold between definitions of performance defined over the forced choice distribution (i.e., categorical, distributional) and multi-label definitions.

## C.5 Proofs

### C.5.1 Theorem C.3

*Proof.* The forward model $\boldsymbol{O}_i = \mathbf{F}_i(\mathbf{E}_i \boldsymbol{\theta}_i^*)$ follows by the following factorization[9] :

$$
\begin{aligned}
\mathbb{P}(o_k \mid t) &= \sum_{v,v^*,r} \mathbb{P}(o_k \mid s_v, s_{v^*}, r, t) \cdot \mathbb{P}(s_v, s_{v^*}, r \mid t) \\
&= \sum_{v,v^*,r} \mathbb{P}(o_k \mid s_v, t) \cdot \mathbb{P}(s_v, s_{v^*}, r \mid t) \qquad\qquad (4) \\
&= \sum_{v,v^*,r} \mathbb{P}(o_k \mid s_v, t) \cdot \mathbb{P}(s_v \mid s_{v^*}, r, t) \cdot \mathbb{P}(s_{v^*}, r \mid t) \\
&= \sum_{v,v^*,r} \mathbb{P}(o_k \mid s_v, t) \cdot \mathbb{P}(s_v \mid s_{v^*}, t) \cdot \mathbb{P}(s_{v^*}, r \mid t) \qquad\qquad (5) \\
&= \sum_v \mathbb{P}(o_k \mid s_v, t) \cdot \sum_{v^*} \mathbb{P}(s_v \mid s_{v^*}, t) \cdot \sum_r \mathbb{P}(s_{v^*}, r \mid t) \\
&= \sum_v \mathbf{F}_{k,v} \cdot \sum_{v^*} \mathbf{E}_{v,v^*} \cdot \boldsymbol{\theta}_{v^*}^*.
\end{aligned}
$$

---

[9]We omit $i$ from all subscripts for brevity.

Above, (4) holds by forced choice independence and (5) holds by error independence. The reverse model $\boldsymbol{\theta}_i^* = \mathbf{E}_i'(\mathbf{F}_i'\mathbf{O}_i)$ follows by the following factorization:

$$
\begin{aligned}
\mathbb{P}(s_{v^*} \mid t) &= \sum_{r,v,k} \mathbb{P}(s_{v^*} \mid r, s_v, o_k, t) \cdot \mathbb{P}(s_v \mid r, o_k, t) \cdot \mathbb{P}(o_k, r \mid t) \\
&= \sum_{r,v,k} \mathbb{P}(s_{v^*} \mid r, s_v, t) \cdot \mathbb{P}(s_v \mid r, o_k, t) \cdot \mathbb{P}(o_k, r \mid t) \qquad (6) \\
&= \sum_{r,v,k} \left( \frac{\mathbb{P}(s_v \mid r, s_{v^*}, t) \cdot \mathbb{P}(r, s_{v^*} \mid t)}{\mathbb{P}(r, s_v \mid t)} \right) \cdot \left( \frac{\mathbb{P}(o_k \mid r, s_v, t) \cdot \mathbb{P}(r, s_v \mid t)}{\mathbb{P}(r, o_k \mid t)} \right) \cdot \mathbb{P}(o_k, r \mid t) \\
&= \sum_{r,v,k} \mathbb{P}(s_v \mid s_{v^*}, t) \cdot \mathbb{P}(r, s_{v^*} \mid t) \cdot \mathbb{P}(o_k \mid s_v, t) \qquad (7) \\
&= \sum_{v,k} \left( \frac{\mathbb{P}(s_v^* \mid s_v, t) \cdot \mathbb{P}(s_v \mid t)}{\mathbb{P}(s_{v^*} \mid t)} \right) \cdot \left( \frac{\mathbb{P}(s_v \mid o_k, t) \cdot \mathbb{P}(o_k \mid t)}{\mathbb{P}(s_v \mid t)} \right) \cdot \sum_r \mathbb{P}(r, s_{v^*} \mid t) \\
&= \sum_{v,k} \mathbb{P}(s_v^* \mid s_v, t) \cdot \mathbb{P}(s_v \mid o_k, t) \cdot \mathbb{P}(o_k \mid t) \\
&= \sum_v \mathbf{E}_{v^*,v}' \cdot \sum_k \mathbf{F}_{v,k}' \cdot \mathbf{O}_k
\end{aligned}
$$

where (6) holds by forced choice independence and (7) holds by forced choice and error independence. □

### C.5.2 Theorem C.4

*Proof.* We remove dependence on $i$ from all terms for brevity. To begin, note that $\boldsymbol{\theta}^*$ is identifiable from $\mathbf{O}$ if and only if $\boldsymbol{\theta}^* = \mathbf{E}'(\mathbf{F}_i'\mathbf{O}) = \mathbf{F}\boldsymbol{\theta}^*$ is fully determined (where the system simplifies by taking $\mathbf{E}$ as the identity matrix). The system system $\mathbf{F}\boldsymbol{\theta}^*$ must be consistent because Theorem C.3 establishes a solution. A consistent system with $n_o = |\mathcal{O}|$ equations and $n_s = |\mathcal{Q}|$ unknowns is fully determined if and only if rank($\mathbf{F}$)$= n_s$.

We will first show that $\mathcal{Q} = \{\{o_k\} : o_k \in \mathcal{O}\}$ implies that rank($\mathbf{F}$)$= n_s$. To begin, note that (1) $\sum_k \mathbf{F}_{k,v} = 1, \ \forall v \in \{1, ..., n_s\}$ because each column in $\mathbf{F}$ represents a valid probability distribution; and (2) $\mathbf{F}_{k,v} = 0, \ \forall a \neq k, \ \forall v \in \{1, ..., n_s\}$ because $\boldsymbol{\Lambda}_{k,v} = 0 \implies \mathbf{F}_{k,v} = 0$. This implies that

$$
1 = \sum_a \mathbf{F}_{a,v} = \sum_{a \neq k} \mathbf{F}_{a,v} + \mathbf{F}_{k,v} = \mathbf{F}_{k,v}, \quad \forall v \in \{1, ..., n_s\}.
$$

Thus, each singleton set $\{o_k\} \in \mathcal{S}$ maps to a standard basis vector $\mathbf{e}_k \in \mathbb{R}^{n_o}$. Further, because $n_s = n_o$ by definition of $\mathcal{Q}$ and $\mathcal{O} \subseteq \mathcal{Q}$, each option must appear in exactly one set, giving us exactly $n_s = n_o$ distinct basis vectors. The rank of a matrix is equal to the number of linear independent column vectors. Because each of the $k$ standard basis vectors must be linearly independent, it follows that rank($\mathbf{F}$)$= n_o = n_s$.

We will show the reverse implication that rank($\mathbf{F}$) $= n_s \implies \mathcal{Q} = \{\{o_k\} : o_k \in \mathcal{O}\}$ by contradiction. Suppose there exists a set $\mathcal{S}_v \in \mathcal{Q}$ containing more than one option, i.e., $|\mathcal{S}| > 1$. Let $\mathbf{v}$ denote the column of $\mathbf{F}$ corresponding to $\mathcal{S}_v$. Since $\mathcal{O} \subseteq \mathcal{Q}$, for each option $o_k \in \mathcal{S}$, there exists a column in $\mathbf{F}$ that is the standard basis vector $\mathbf{e}_k$, as shown above. Therefore, $\mathbf{v}$ can be written as a linear combination of these basis vectors: $\mathbf{v} = \sum_k \alpha_k \mathbf{e}_k$ where $\alpha_k = [0, 1]$. This shows that column $\mathbf{v}$ is linearly dependent with the columns corresponding to singleton sets $o_k$ for $o_k \in \mathcal{S}$. This implies $\mathbf{F}$ cannot have $n_s$ linearly independent columns, contradicting rank($\mathbf{F}$) $= n_s$.

□

### C.5.3  Theorem C.7.

*Proof.* Proof by contradiction. Let $\mathbf{p}$ and $\mathbf{p}_*$ denote pairs of performance definitions with increasing cardinality (i.e., higher values being better). Let $Y^Z = a^J(\mathbb{P}_T^{J,Z})$, $Y^W = a^J(\mathbb{P}_T^{J,W})$, $Y^H = a^H(\mathbb{P}_T^H)$ denote random functions of $T$ corresponding to definition $\mathbf{p}$. Let $Y_*^Z = a_*^J(\mathbb{P}_T^{J,Z})$, $Y_*^W = a_*^W(\mathbb{P}_T^{J,W})$, $Y_*^H = a_*^H(\mathbb{P}_T^H)$ correspond to definition $\mathbf{p}_*$. Since $\mathbf{p}$ is not a monotone transformation of $\mathbf{p}_*$, by definition there must exist distributions $\{(\mathbb{P}_i^{J,Z}, \mathbb{P}_i^H), (\mathbb{P}_i^{J,W}, \mathbb{P}_i^H)\} \in \Delta \times \Delta$ corresponding to realizations of these random variables satisfying

$$m(Y^Z, Y^H) < m(Y^W, Y^H), \qquad m_*(Y_*^Z, Y_*^H) > m(Y_*^W, Y_*^H).$$

Now suppose that $\mathbb{P}^*$ places all marginal probability mass over $T$ on the $i$'th item – i.e., $\mathbb{P}^*(T = t_i) = 1$. Then:

$$\delta_{\mathbf{p}}(Z, W) = \mathbb{E}_{\mathbb{P}^*}[m(Y^Z, Y^H) - m(Y^W, Y^H)] < 0$$
$$\delta_{\mathbf{p}'}(Z, W) = \mathbb{E}_{\mathbb{P}^*}[m_*(Y_*^Z, Y_*^H) - m_*(Y_*^W, Y_*^H)] > 0$$

Thus, rank consistency is violated because there exists a distribution $\mathbb{P}^*$ for which $\mathcal{G}_{judge}^Z \succ_{\mathbf{p}_*} \mathcal{G}_{judge}^W$ but $\mathcal{G}_{judge}^W \succ_{\mathbf{p}} \mathcal{G}_{judge}^Z$. This provides a contradiction, proving the result.

$\square$

### C.6  Rank Consistency Under Rater Error

**Lemma C.8** (Rank Consistency of MSE (srs/srs) Under Rater Error ). *Let $\boldsymbol{\theta}^*$ and $\boldsymbol{\theta} = \mathbf{E}\boldsymbol{\theta}^*$ denote the stable and observed response set distributions for human raters.[10] Let $\boldsymbol{\theta}^{J,Z}$ and $\boldsymbol{\theta}^{J,W}$ denote observed response set distributions for judge systems $\mathcal{G}_{judge}^Z$ and $\mathcal{G}_{judge}^W$ where both judge systems have a rater error matrix $\mathbf{E}^{J,Z}$ and $\mathbf{E}^{J,W}$ that is the identity. Let $\boldsymbol{\Lambda}$ be the binary matrix mapping response sets to options and define $\delta^* = MSE(\boldsymbol{\Omega}^*, \boldsymbol{\Omega}^{J,Z}) - MSE(\boldsymbol{\Omega}^*, \boldsymbol{\Omega}^{J,W})$ as the difference in MSE under error-free conditions. The ranking of judge systems using MSE with soft response set aggregation is preserved under human rating error if and only if:*

$$sign((\boldsymbol{\theta}^* - \boldsymbol{\theta})^T \boldsymbol{\Lambda}^T \boldsymbol{\Lambda} (\boldsymbol{\theta}^{J,W} - \boldsymbol{\theta}^{J,Z})) = sign(\delta^*). \tag{8}$$

This lemma provides conditions under which measuring the performance of a judge system against error-corrupted versus error-free human ratings yields a consistent ranking of judge systems (when measured against MSE(srs/srs)). The condition essentially requires that the direction of the error-induced shift in human ratings ($\boldsymbol{\theta}^* - \boldsymbol{\theta}$) matches the direction of the stable response set shift across judge systems ($\boldsymbol{\theta}^{J,W} - \boldsymbol{\theta}^{J,Z}$) when projected to the multi-label space. If human rating error and judge system performance differences shift the response set distribution in the same direction, the ranking of judge systems will be consistent for error-free and error-corrupted ratings. Conversely, rankings can invert under an inverse relationship.

E.q. (8) is satisfied in our experimental setup because the ensemble of judge system rating distributions is generated by adding random perturbations (i.e., uncorrelated with rater error) to the human stable response set vector. Thus we see little change in the reliability of MSE (srs/srs) across settings with no rater error (Figure 34, center) and rater error (Figure 34, right).

*Proof of Lemma C.8.* For brevity, let $\mathbf{M} = \boldsymbol{\Lambda}^T \boldsymbol{\Lambda}$. The difference in judge system MSE measured against the multi-label human rating vector $\boldsymbol{\Omega}^* = \boldsymbol{\Lambda}\boldsymbol{\theta}^*$ derived from the stable response set distribu-

---

[10]We omit subscript $i$ from all terms for brevity. We also omit superscript $h$ from human response set distribution, error, and multi-label vectors where the context is clear.

tion is given by:

$$\delta^* = \text{MSE}(\mathbf{\Omega}^*, \ \mathbf{\Omega}^{J,Z}) - \text{MSE}(\mathbf{\Omega}^*, \ \mathbf{\Omega}^{J,W})$$
$$= ||\mathbf{\Lambda}\boldsymbol{\theta}^* - \mathbf{\Lambda}\boldsymbol{\theta}^{J,Z}||_2^2 - ||\mathbf{\Lambda}\boldsymbol{\theta}^* - \mathbf{\Lambda}\boldsymbol{\theta}^{J,W}||_2^2$$
$$= (\boldsymbol{\theta}^* - \boldsymbol{\theta}^{J,Z})^T\mathbf{M}(\boldsymbol{\theta}^* - \boldsymbol{\theta}^{J,Z}) - (\boldsymbol{\theta}^* - \boldsymbol{\theta}^{J,W})^T\mathbf{M}(\boldsymbol{\theta}^* - \boldsymbol{\theta}^{J,W})$$
$$= (\boldsymbol{\theta}^*)^T\mathbf{M}\boldsymbol{\theta}^* - (\boldsymbol{\theta}^*)^T\mathbf{M}\boldsymbol{\theta}^{J,Z} - (\boldsymbol{\theta}^{J,Z})^T\mathbf{M}\boldsymbol{\theta}^* + (\boldsymbol{\theta}^{J,Z})^T\mathbf{M}\boldsymbol{\theta}^{J,Z}$$
$$- (\boldsymbol{\theta}^*)^T\mathbf{M}\boldsymbol{\theta}^* + (\boldsymbol{\theta}^*)^T\mathbf{M}\boldsymbol{\theta}^{J,W} + (\boldsymbol{\theta}^{J,W})^T\mathbf{M}\boldsymbol{\theta}^* - (\boldsymbol{\theta}^{J,W})^T\mathbf{M}\boldsymbol{\theta}^{J,W}$$
$$= (\boldsymbol{\theta}^{J,Z})^T\mathbf{M}\boldsymbol{\theta}^{J,Z} - (\boldsymbol{\theta}^{J,W})^T\mathbf{M}\boldsymbol{\theta}^{J,W} + 2(\boldsymbol{\theta}^*)^T\mathbf{M}(\boldsymbol{\theta}^{J,W} - \boldsymbol{\theta}^{J,Z})$$

Let $\mathbf{\Omega} = \mathbf{\Lambda}(\mathbf{E}\boldsymbol{\theta})$ denote the multi-label vector recovered from the observed response set distribution. Applying the same derivation as above to the error-corrupted MSE metric yields:

$$\delta = \text{MSE}(\mathbf{\Omega}, \ \mathbf{\Omega}^{J,Z}) - \text{MSE}(\mathbf{\Omega}, \ \mathbf{\Omega}^{J,W})$$
$$= (\boldsymbol{\theta}^{J,Z})^T\mathbf{M}\boldsymbol{\theta}^{J,Z} - (\boldsymbol{\theta}^{J,W})^T\mathbf{M}\boldsymbol{\theta}^{J,W} + 2(\mathbf{E}\boldsymbol{\theta}^*)^T\mathbf{M}(\boldsymbol{\theta}^{J,W} - \boldsymbol{\theta}^{J,Z})$$

Observe that the first two terms appear in both expansions. Thus we need to focus on the third term while showing the conditions required for rank consistency — i.e., $\text{sign}(\delta) = \text{sign}(\delta^*)$.

- **Case 1:** $\delta^* < 0$ ($\mathcal{G}^Z_{judge}$ is better than $\mathcal{G}^W_{judge}$ under no rater error.) For both inequalities to hold, we need:

$$2(\mathbf{E}\boldsymbol{\theta}^*)^T\mathbf{M}(\boldsymbol{\theta}^{J,W} - \boldsymbol{\theta}^{J,Z}) \leq 2(\boldsymbol{\theta}^*)^T\mathbf{M}(\boldsymbol{\theta}^{J,W} - \boldsymbol{\theta}^{J,Z})$$
$$(\mathbf{E}\boldsymbol{\theta}^*)^T\mathbf{M}(\boldsymbol{\theta}^{J,W} - \boldsymbol{\theta}^{J,Z}) \leq (\boldsymbol{\theta}^*)^T\mathbf{M}(\boldsymbol{\theta}^{J,W} - \boldsymbol{\theta}^{J,Z})$$
$$(\boldsymbol{\theta}^*)^T\mathbf{M}(\boldsymbol{\theta}^{J,W} - \boldsymbol{\theta}^{J,Z}) - (\mathbf{E}\boldsymbol{\theta}^*)^T\mathbf{M}(\boldsymbol{\theta}^{J,W} - \boldsymbol{\theta}^{J,Z}) \geq 0$$
$$((\boldsymbol{\theta}^*)^T - (\mathbf{E}\boldsymbol{\theta}^*)^T)\mathbf{M}(\boldsymbol{\theta}^{J,W} - \boldsymbol{\theta}^{J,Z}) \geq 0$$
$$(\boldsymbol{\theta}^* - \mathbf{E}\boldsymbol{\theta}^*)^T\mathbf{M}(\boldsymbol{\theta}^{J,W} - \boldsymbol{\theta}^{J,Z}) \geq 0$$

- **Case 2**: $\delta^* > 0$ ($\mathcal{G}^W_{judge}$ is better than $\mathcal{G}^Z_{judge}$ under no rater error.). For rank consistency, we need $\delta > 0$ as well. Following similar steps, we get:

$$2(\mathbf{E}\boldsymbol{\theta}^*)^T\mathbf{M}(\boldsymbol{\theta}^{J,W} - \boldsymbol{\theta}^{J,Z}) \geq 2(\boldsymbol{\theta}^*)^T\mathbf{M}(\boldsymbol{\theta}^{J,W} - \boldsymbol{\theta}^{J,Z})$$
$$-2(\mathbf{E}\boldsymbol{\theta}^*)^T\mathbf{M}(\boldsymbol{\theta}^{J,W} - \boldsymbol{\theta}^{J,Z}) \leq -2(\boldsymbol{\theta}^*)^T\mathbf{M}(\boldsymbol{\theta}^{J,W} - \boldsymbol{\theta}^{J,Z})$$
$$2(\mathbf{E}\boldsymbol{\theta}^*)^T\mathbf{M}(\boldsymbol{\theta}^{J,Z} - \boldsymbol{\theta}^{J,W}) \leq 2(\boldsymbol{\theta}^*)^T\mathbf{M}(\boldsymbol{\theta}^{J,Z} - \boldsymbol{\theta}^{J,W})$$
$$(\boldsymbol{\theta}^* - \mathbf{E}\boldsymbol{\theta}^*)^T\mathbf{M}(\boldsymbol{\theta}^{J,Z} - \boldsymbol{\theta}^{J,W}) \geq 0$$

$\square$

| Rating Task | Property | Citation | Ratings per Item |
|---|---|---|---|
| Civil Comments | Toxicity | [Borkan et al., 2019] | 10+ |
| MNLI | Entailment | [Williams et al., 2017] | 100 |
| SNLI | Entailment | [Bowman et al., 2015] | 100 |
| $\alpha$-NLI | Entailment | [Nie et al., 2019] | 100 |
| SummEval (Relevance) | Relevance | [Fabbri et al., 2021] | 8 |
| SummEval (Coherence) | Coherence | [Fabbri et al., 2021] | 8 |
| SummEval (Consistency) | Factuality | [Fabbri et al., 2021] | 8 |
| SummEval (Fluency) | Fluency | [Fabbri et al., 2021] | 8 |
| QAGS | Factuality | [Wang et al., 2020] | 3 |
| TopicalChat (Uses Knowledge) | Uses Knowledge | [Gopalakrishnan et al., 2023] | 3 |
| TopicalChat (Understandable) | Understandable | [Gopalakrishnan et al., 2023] | 3 |

Table 2: An overview of the eleven rating tasks included in our empirical analysis. Each task includes the property being evaluated and number of human ratings per item.

# D  Real-Data Experiments: Setup Details and Additional Results

**Rating Task Configuration.** Table 2 provides an overview of all rating tasks included in our experiments. Tables 3-5 illustrate the approach used to reconstruct the response set distribution from the forced choice distribution via the sensitivity parameter $\beta_t^H$. Table 5 applies to all rating tasks with binary Yes/No options [Fabbri et al., 2021, Wang et al., 2020, Gopalakrishnan et al., 2023].

**Prompts.** We provide forced-choice and response set variants of all prompts in Fig. 23-18 of § D.1. The "no explanations" instruction included in prompts pertains to the final model output. It does not prohibit a reasoning-enabled model (e.g., o3-Mini) from producing a reasoning trace at inference time.

**Model Inference Settings.** We sample all models with a temperature of 1.0. For all models, we also limit `max_tokens` used for generation to 5. This low max token limit feasible because only few tokens are needed to provide a forced-choice or response set rating (e.g., "A", "BB"). When using a reasoning-enabled model (e.g., o3-Mini), we set the max token length for the reasoning trace to 1024 tokens.

**Invalid Responses and Robustness Testing.** For all rating tasks, we assign a *null* forced choice option $\emptyset \in \mathcal{O}$ and response set $\{\emptyset\} \in \mathcal{Q}$ that are selected when a judge system returns an invalid character. Probability mass assigned to this null option is penalized in human–judge agreement metrics. Because we specifically study factors in the *human* rating process that can confound meta-evaluation, we do not systematically study the influence of factors such as prompt formatting [Sclar et al., 2023] and option ordering [Pezeshkpour and Hruschka, 2023] on our results. In practice, meta-evaluation designers may also want to report such analysis as part of their validation pipeline.

**Item Sampling.** To match common LLM-as-a-judge meta-evaluation workflows that conduct analysis on a small corpus of items, we sub-sample all rating tasks to 200 ratings per item. We randomly sample items for all rating tasks apart from civil comments, which is sampled via a stratified random sampling approach to select comments with an observed agreement level in the range $[0.2, 0.5]$.

**Summ Eval Task Design.** SummEval ratings tasks were originally collected on a 1-5 Likert scale. Because this corresponds to many response sets $31 = 2^5 - 1$ in comparison to the number of ratings

collected per item (8), we discretize the scale by assigning a positive binary label if a rater selected $\geq$ 4. This discretization improves the finite sample stability of estimated agreement metrics (§G).

**Cost and Inference Time.** The total cost of running all models was 199.76. Each rating task took 30 minutes to run when models were run in parallel, with the exception of Claude Sonnet 3.5, which took approximately 2 hours per rating task due to high API response latency.

**Estimation Details.** We now outline our approach for estimating human–judge agreement metrics via a finite sample rating corpus. For each item $i$, we estimate each term in our rating model:

1. **Judge system:** Estimate $\hat{\mathbf{O}}_i^J$ (forced choice prompt), $\hat{\mathbf{\Omega}}_i = \mathbf{\Lambda}\hat{\boldsymbol{\theta}}_i^J$ (response set prompt) via empirical frequencies from judge responses.

2. **Humans:** Estimate $\hat{\mathbf{O}}_i^H$ via empirical frequencies from forced choice human ratings. Next, apply reverse rating model $\hat{\mathbf{\Omega}}_i^H = \mathbf{\Lambda}(\mathbf{F}_i'(\hat{\mathbf{O}}_i^H))$, where $\mathbf{F}_i'$ is recovered from the sensitivity parameter $\beta^H$.

We then take the empirical average over the evaluation corpus to estimate the human–judge agreement metric:

$$\hat{M}(\hat{Y}^J, \hat{Y}^H) = \frac{1}{n} \sum_i m(\hat{Y}_i^J, \hat{Y}_i^H) = \frac{1}{n} \sum_i m(a^J(\hat{\mathbf{O}}_i^J, \hat{\mathbf{\Omega}}_i^J), a^H(\hat{\mathbf{O}}_i^H, \hat{\mathbf{\Omega}}_i^H))$$

where the third term illustrates the full expansion with the application of aggregation functions to the estimated rating terms. We estimate downstream performance metrics using the same approach.

**Computation of MSE $\hat{F}$.** Our experiments reporting MSE $F$ assume oracle knowledge of $\mathbf{F}'$ in step (2) of the estimation procedure outlined above. Our experiments reporting MSE $\hat{F}$ apply step (2), but recover $\hat{\mathbf{\Omega}}_i^H = \mathbf{\Lambda}(\hat{\mathbf{F}}_i'(\hat{\mathbf{O}}_i^H))$ via $\hat{\mathbf{F}}_i'$. To estimate $\hat{\mathbf{F}}_i'$, we sample an auxiliary corpus with 200 (forced choice, response set) paired ratings. We sample one paired rating for each item in the validation corpus. We then use this auxiliary corpus to compute sample estimates of the conditional probabilities in the matrix $\hat{\mathbf{F}}'$, where each entry denotes the probability of a rater selecting the $q$th response set given that they picked the $k$th forced choice option. We assume that $\hat{\mathbf{F}}'$ is fixed across items to improve sampling efficiency. This works well in practice in our setting (Fig 5).

**Additional Experimental Results.** Below, we report a version of our main findings with an expanded set of metrics (Fig. 10), sensitivity parameter estimates recovered from all models across all tasks (Fig. 11), and detailed rank analysis for pairs of performance metrics for all rating tasks (Figs. 12-22).

| Options | Response Sets | | | | | | |
|---|---|---|---|---|---|---|---|
| | {VT} | {T} | {N/U} | {VT,T} | {T,N/U} | {VT,N/U} | {VT,T,N/U} |
| VT | 1 | 0 | 0 | 0 | 0 | 0 | 0 |
| T | 0 | 1 | 0 | 0 | 0 | 0 | 0 |
| N/U | 0 | $\beta_t^H$ | $1 - \beta_t^H$ | 0 | 0 | 0 | 0 |

Table 3: Civil Comments rating task [Borkan et al., 2019]. Description of how the sensitivity parameter ($\beta_t^H$) is used to construct the forced choice translation matrix that recovers the response set distribution from the forced choice distribution. Positive options are $\mathcal{O}_+ = \{\text{Toxic}, \text{Very Toxic}\}$. VT = Very Toxic, T = Toxic, N/U = No/Unsure. For completeness, note that technically this illustrates the *reverse* forced choice translation matrix $\mathbf{F}'$ (forced choice $\rightarrow$ response set distribution) as opposed to the forward translation matrix $\mathbf{F}$ that maps the response set to forced choice distribution (see § C).

| Options | Response Sets | | | | | | |
|---|---|---|---|---|---|---|---|
| | {E} | {N} | {C} | {E, N} | {E, C} | {N, C} | {E, N, C} |
| E | 1 | 0 | 0 | 0 | 0 | 0 | 0 |
| N | 0 | 1 | 0 | 0 | 0 | 0 | 0 |
| C | $\beta_t^H$ | 0 | $1 - \beta_t^H$ | 0 | 0 | 0 | 0 |

Table 4: Natural Language Inference tasks (SNLI, MNLI). Description of how the sensitivity parameter ($\gamma$) is used to construct the forced choice translation matrix that recovers the response set distribution from the forced choice distribution. E=Entailment, C=Contradiction, N=Neutral. Applies to SNLI [Bowman et al., 2015] and MNLI [Williams et al., 2017] datasets.

| Options | Response Sets | | |
|---|---|---|---|
| | {Positive} | {Negative} | {Positive, Negative} |
| Positive | 1 | 0 | 0 |
| Negative | $\beta_t^H$ | $1 - \beta_t^H$ | 0 |

Table 5: Binary classification tasks. Description of how the sensitivity parameter ($\beta_t^H$) is used to construct the forced choice translation matrix that recovers the response set distribution from the forced choice distribution. Applies to rating tasks in QAGS [Wang et al., 2020], SummEval [Fabbri et al., 2021], $\alpha$-NLI [Nie et al., 2019], and TopicalChat [Gopalakrishnan et al., 2023].

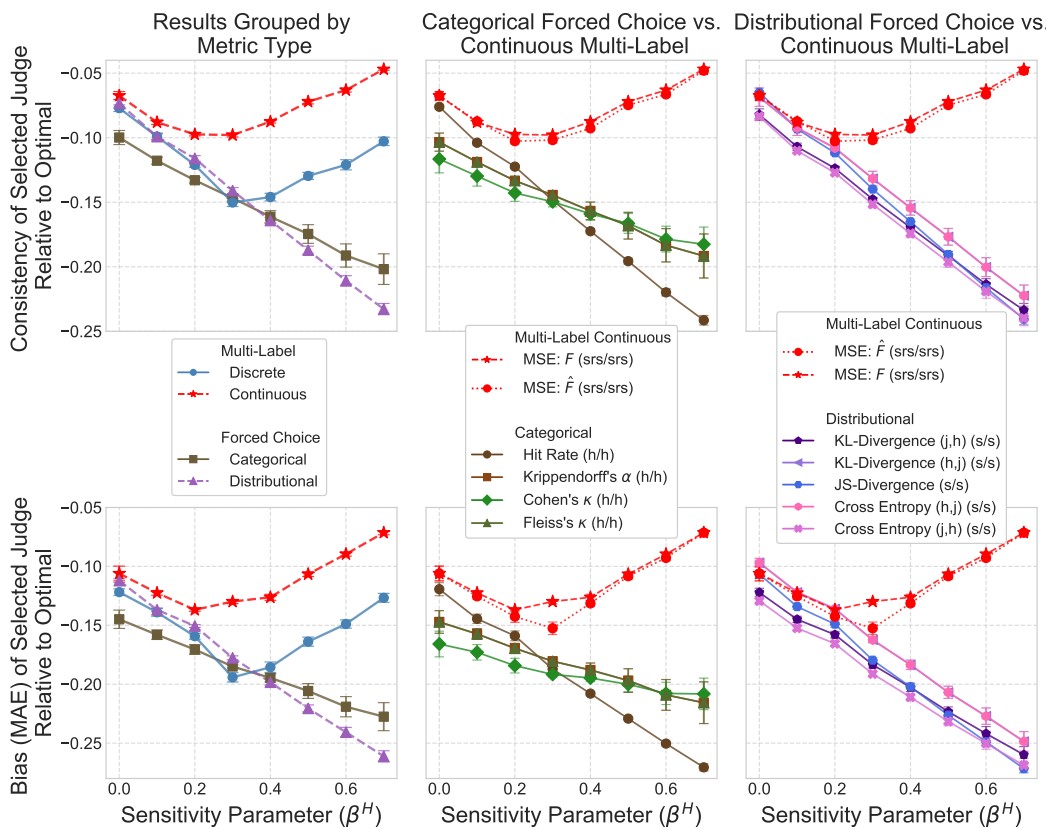

Figure 10: Main results with an extended set of metrics (see Fig. 5). Center panel: Fleiss's $\kappa$ performs similarly to Krippendorff's $\alpha$. Right panel: KL-Divergence (j,h) and Cross Entropy (j,h) perform robustly and KL-Divergence (h,j) and Cross Entropy (h,j) perform poorly across settings.

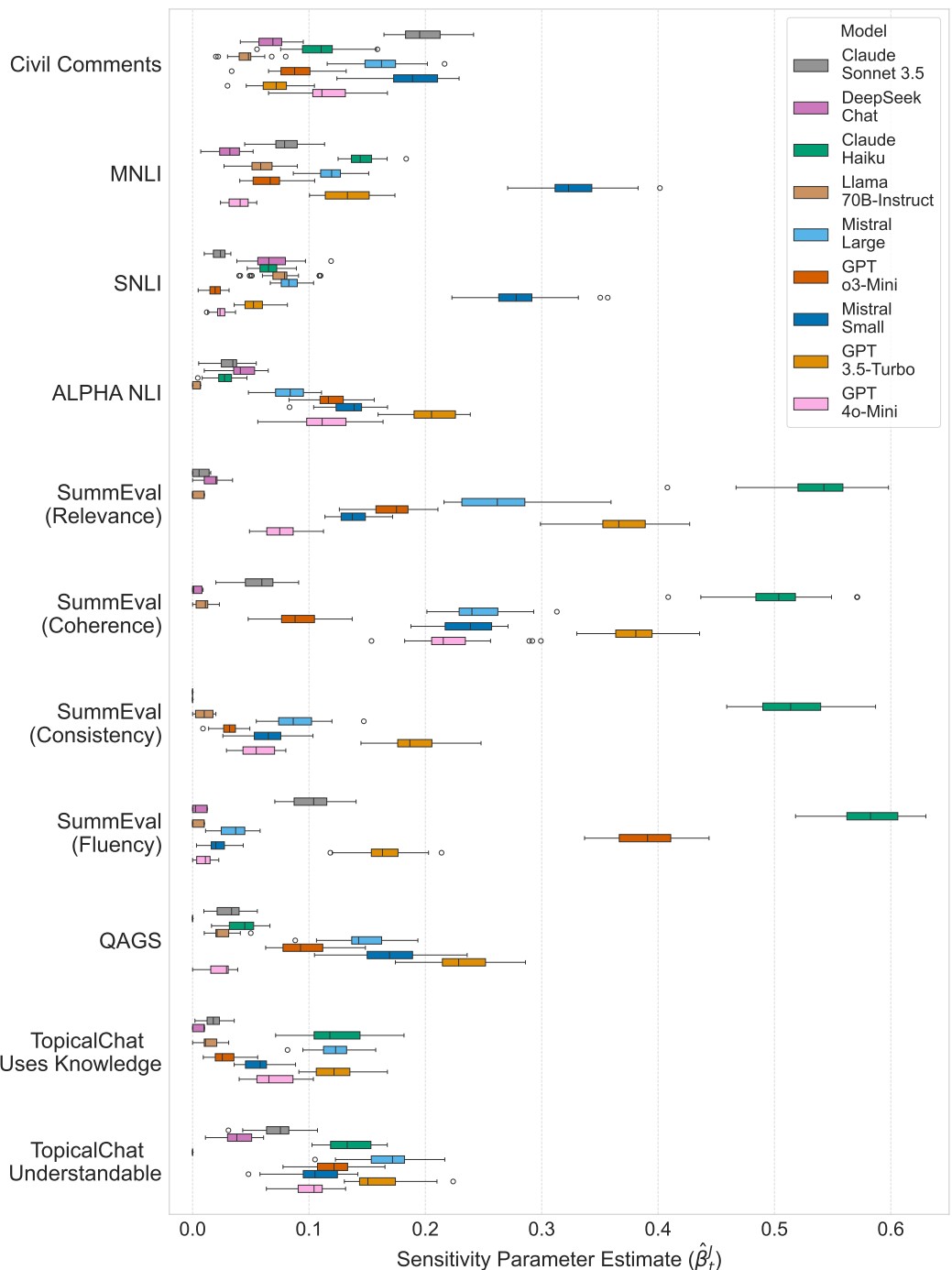

Figure 11: Task-specific sensitivity parameters ($\hat{\beta}_t^J$) recovered from all judge systems across tasks (non-abridged version of Fig. 6). Estimates vary across (1) judge systems within the same task, and (2) across tasks. This indicates heterogeneity in how judge systems respond to indeterminacy.

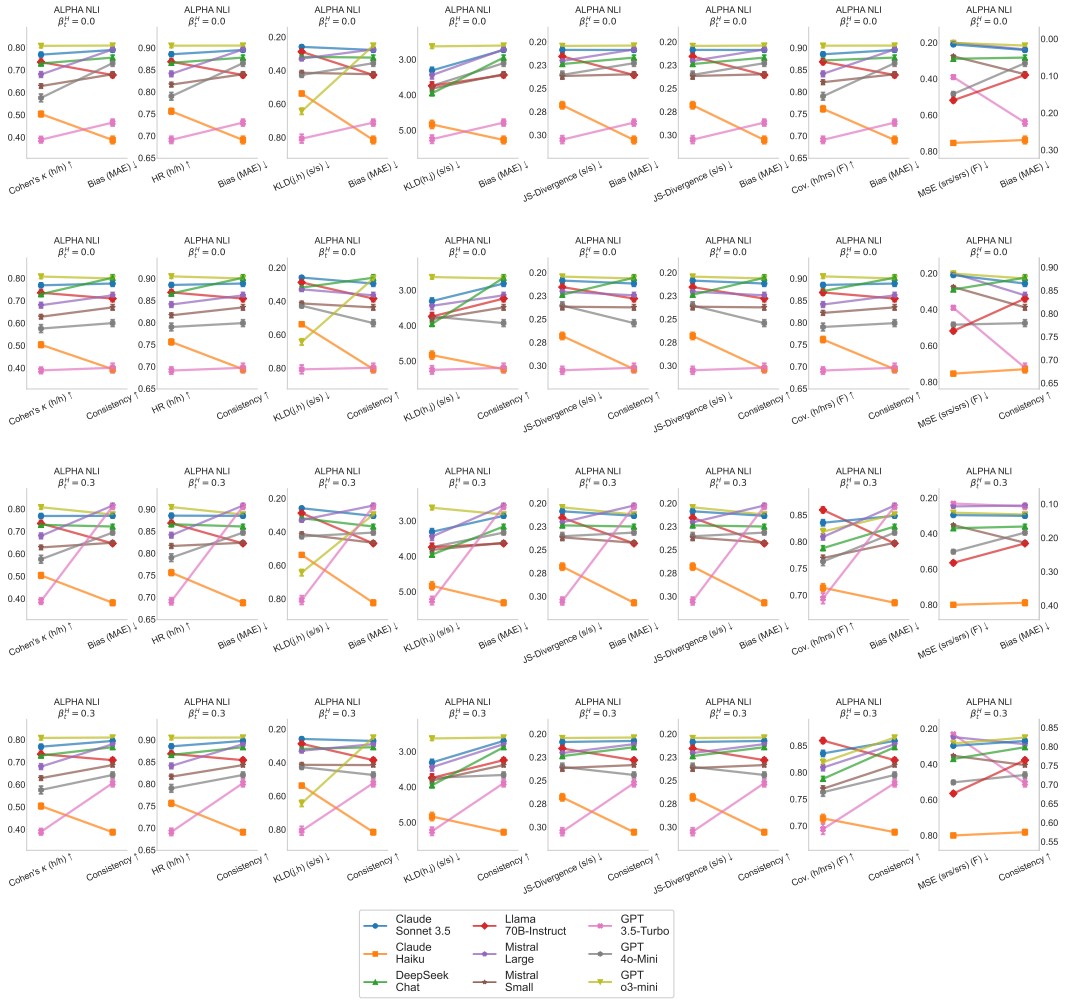

Figure 12: Ranking of models determined by human–judge agreement metrics versus downstream metrics for the $\alpha$-NLI rating task. Top two rows: $\beta_t^H = 0$, Bottom two rows: $\beta_t^H = 0.3$.

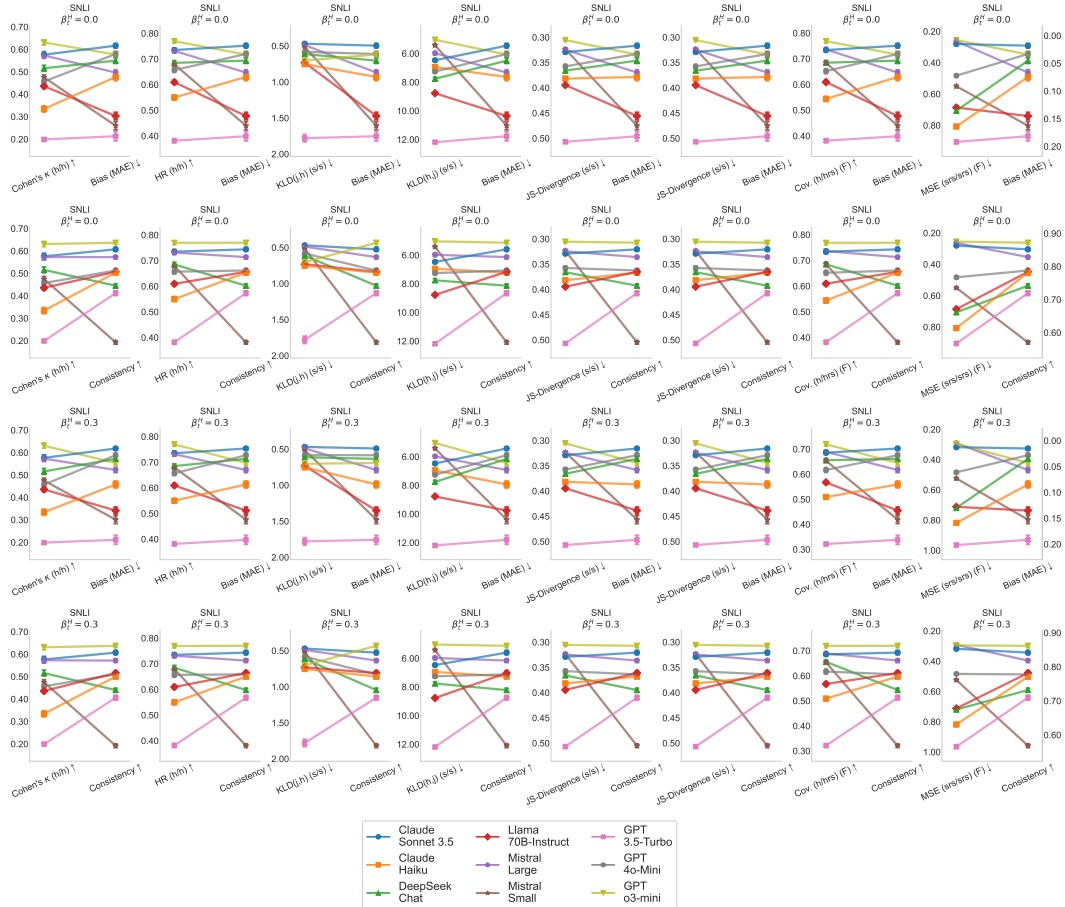

Figure 13: Ranking of models determined by human–judge agreement metrics versus downstream metrics for the SNLI rating task. Top two rows: $\beta_t^H = 0$, Bottom two rows: $\beta_t^H = 0.3$.

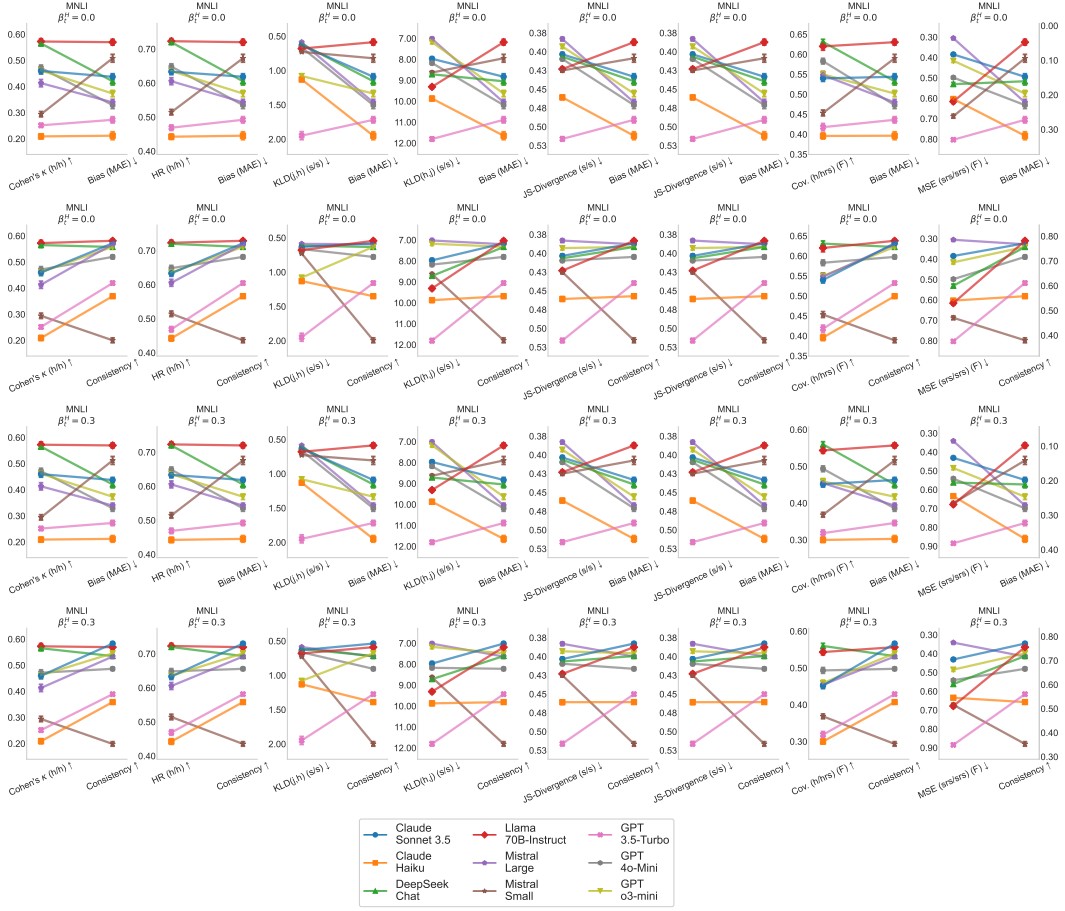

Figure 14: Ranking of models determined by human–judge agreement metrics versus downstream metrics for the MNLI rating task. Top two rows: $\beta_t^H = 0$, Bottom two rows: $\beta_t^H = 0.3$.

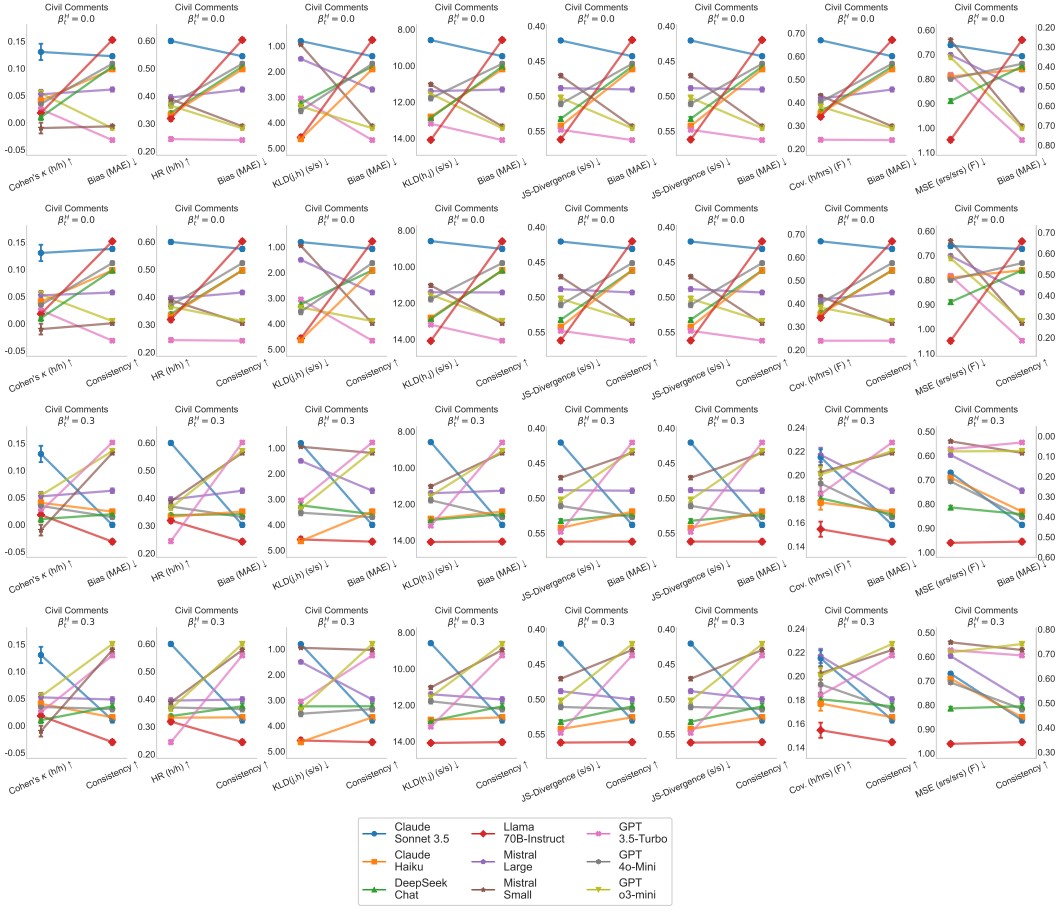

Figure 15: Ranking of models determined by human–judge agreement metrics versus downstream metrics for the Civil Comments rating task. Top two rows: $\beta_t^H = 0$, Bottom two rows: $\beta_t^H = 0.3$.

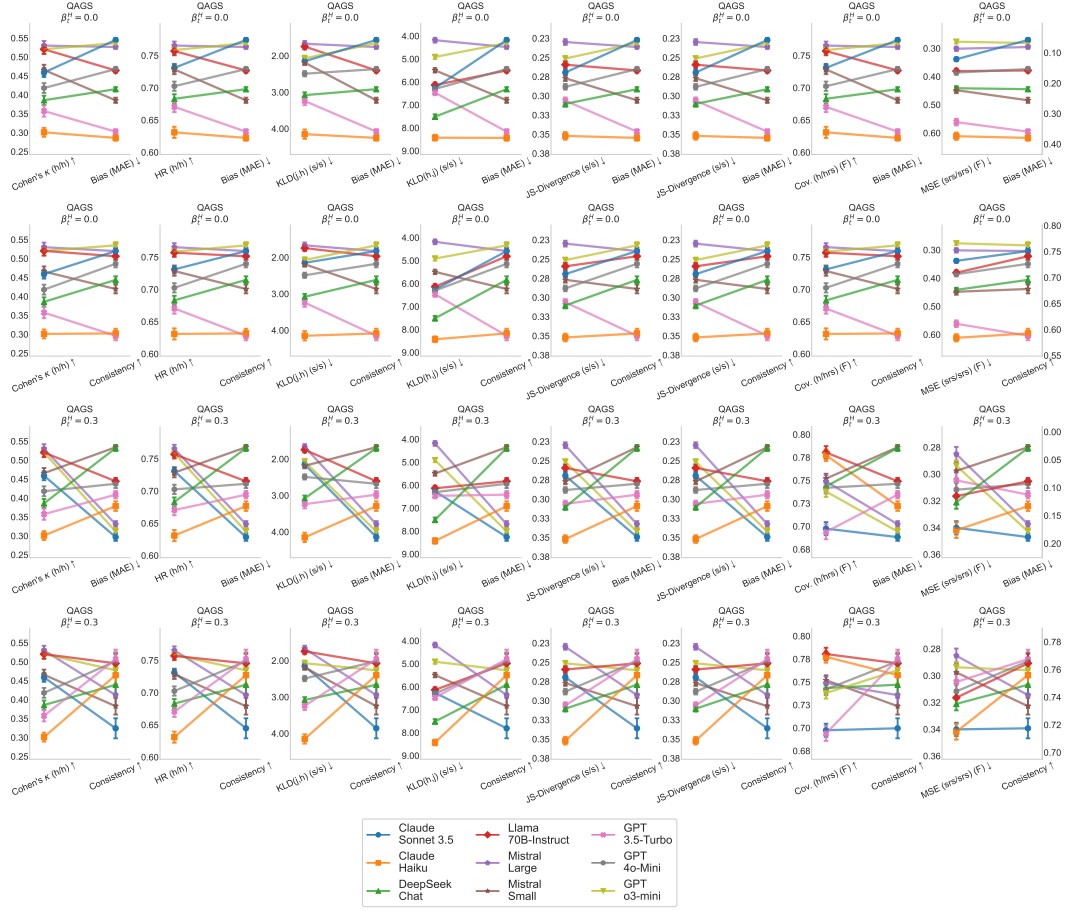

Figure 16: Ranking of models determined by human–judge agreement metrics versus downstream metrics for the QAGS rating task. Top two rows: $\beta_t^H = 0$, Bottom two rows: $\beta_t^H = 0.3$.

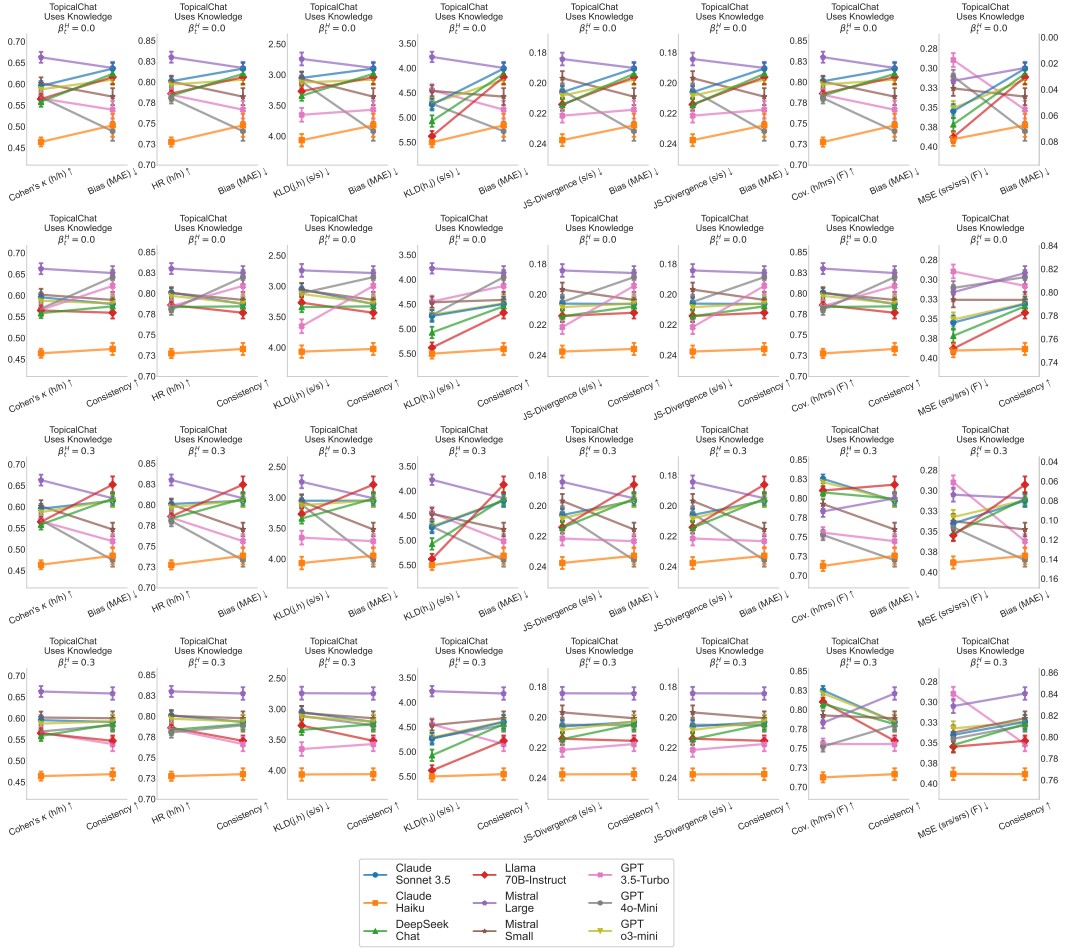

Figure 17: Ranking of models determined by human–judge agreement metrics versus downstream metrics for the TopicalChat "Uses Knowledge" rating task. Top two rows: $\beta_t^H = 0$, Bottom two rows: $\beta_t^H = 0.3$.

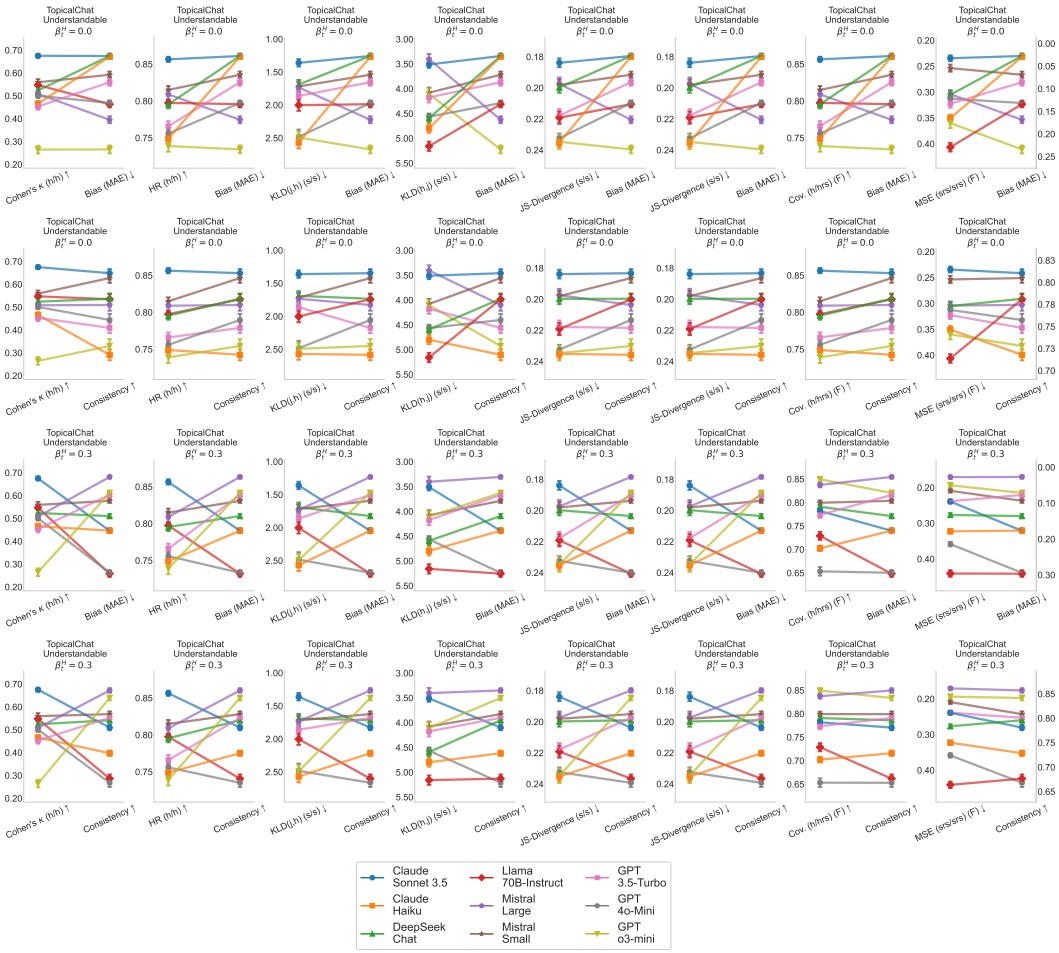

Figure 18: Ranking of models determined by human–judge agreement metrics versus downstream metrics for the TopicalChat "Understandable" rating task. Top two rows: $\beta_t^H = 0$, Bottom two rows: $\beta_t^H = 0.3$.

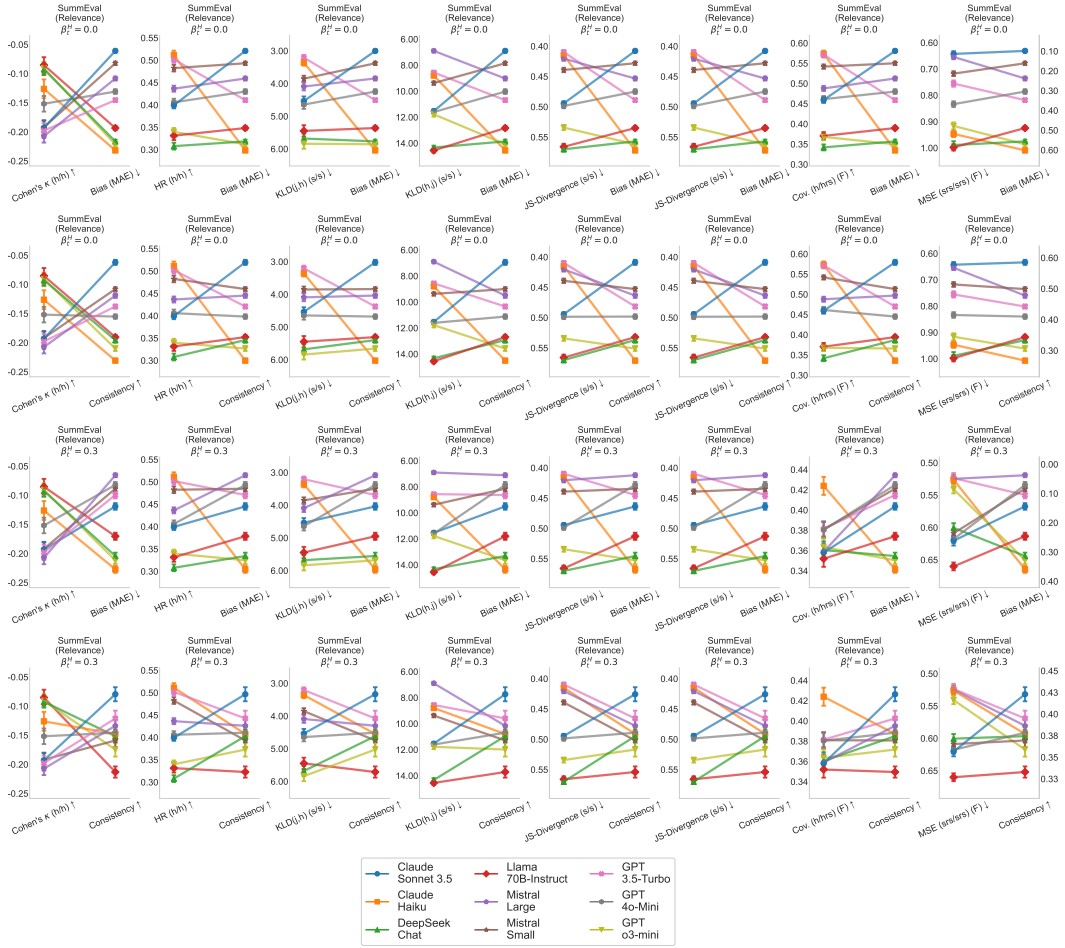

Figure 19: Ranking of models determined by human–judge agreement metrics versus downstream metrics for the SummEval "relevance" rating task. Top two rows: $\beta_t^H = 0$, Bottom two rows: $\beta_t^H = 0.3$.

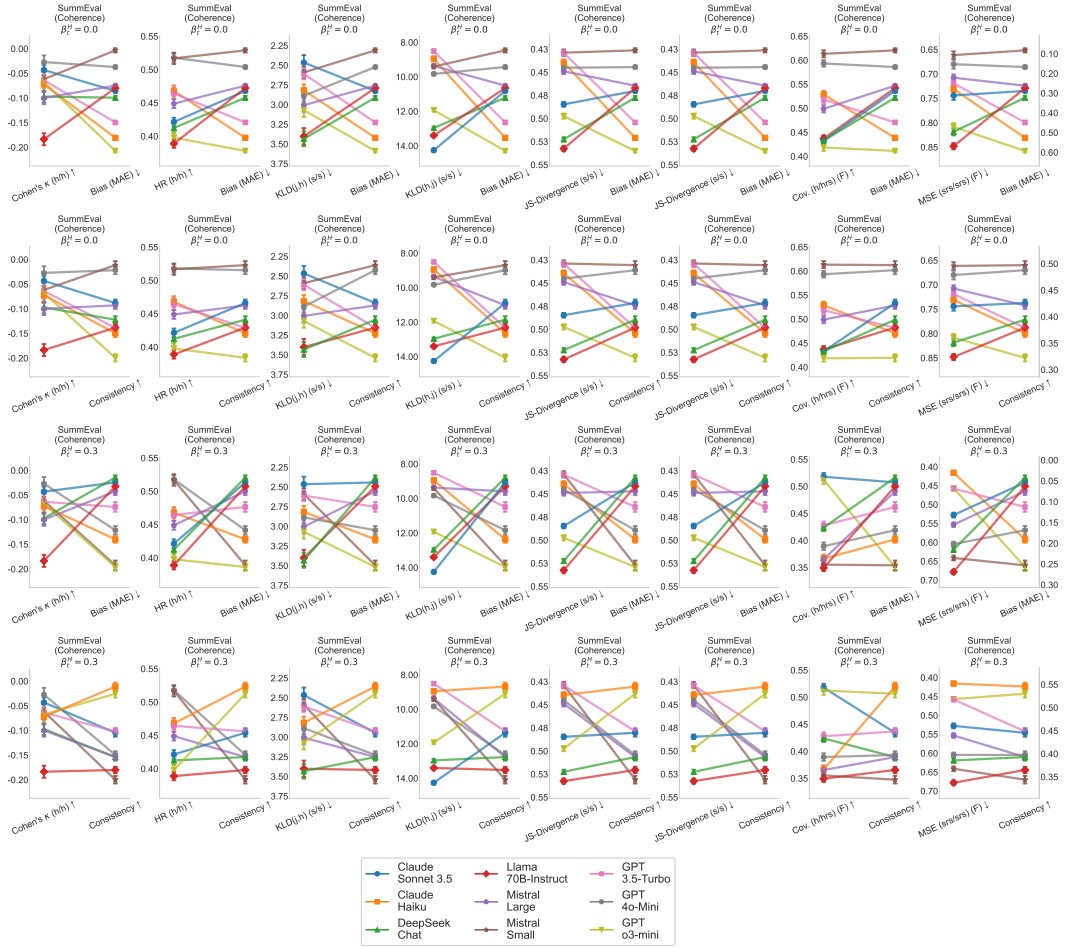

Figure 20: Ranking of models determined by human–judge agreement metrics versus downstream metrics for the SummEval "coherence" rating task. Top two rows: $\beta_t^H = 0$, Bottom two rows: $\beta_t^H = 0.3$.

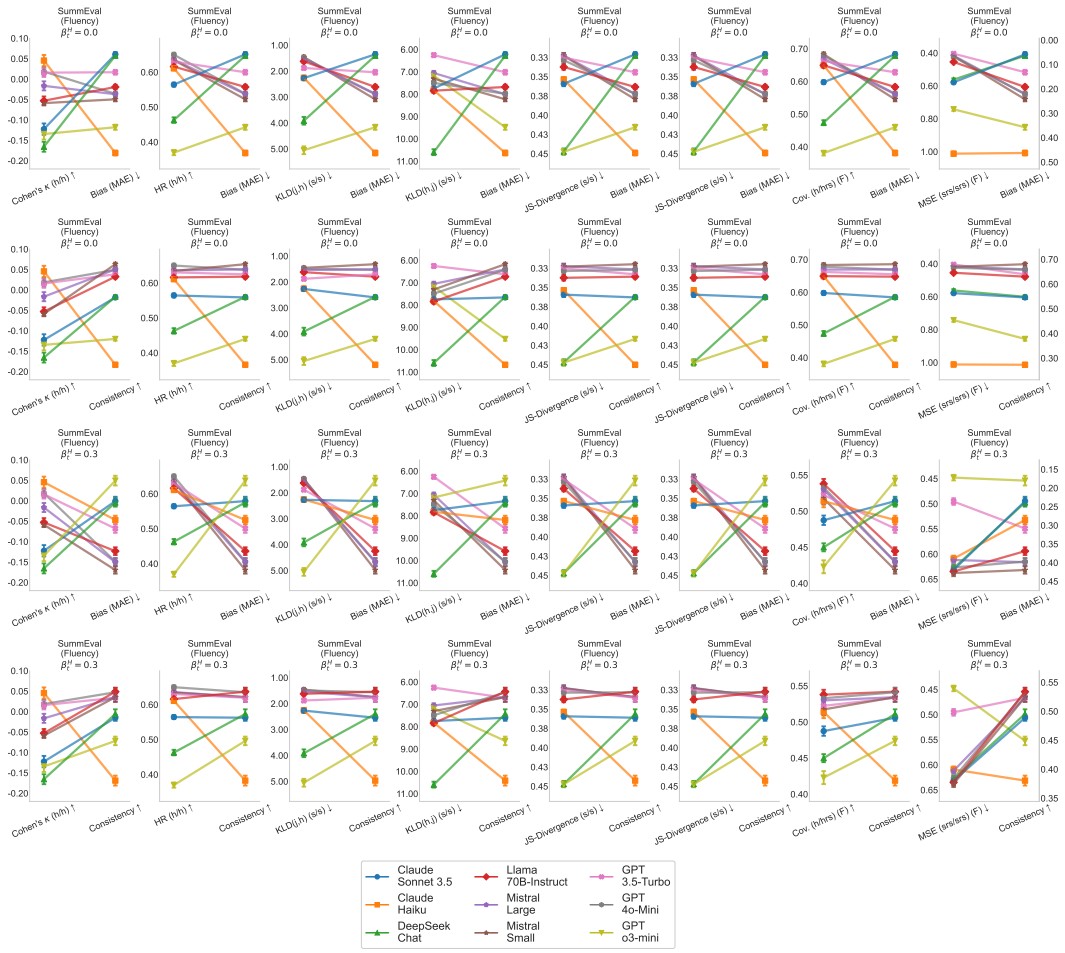

Figure 21: Ranking of models determined by human–judge agreement metrics versus downstream metrics for the SummEval "fluency" rating task. Top two rows: $\beta_t^H = 0$, Bottom two rows: $\beta_t^H = 0.3$.

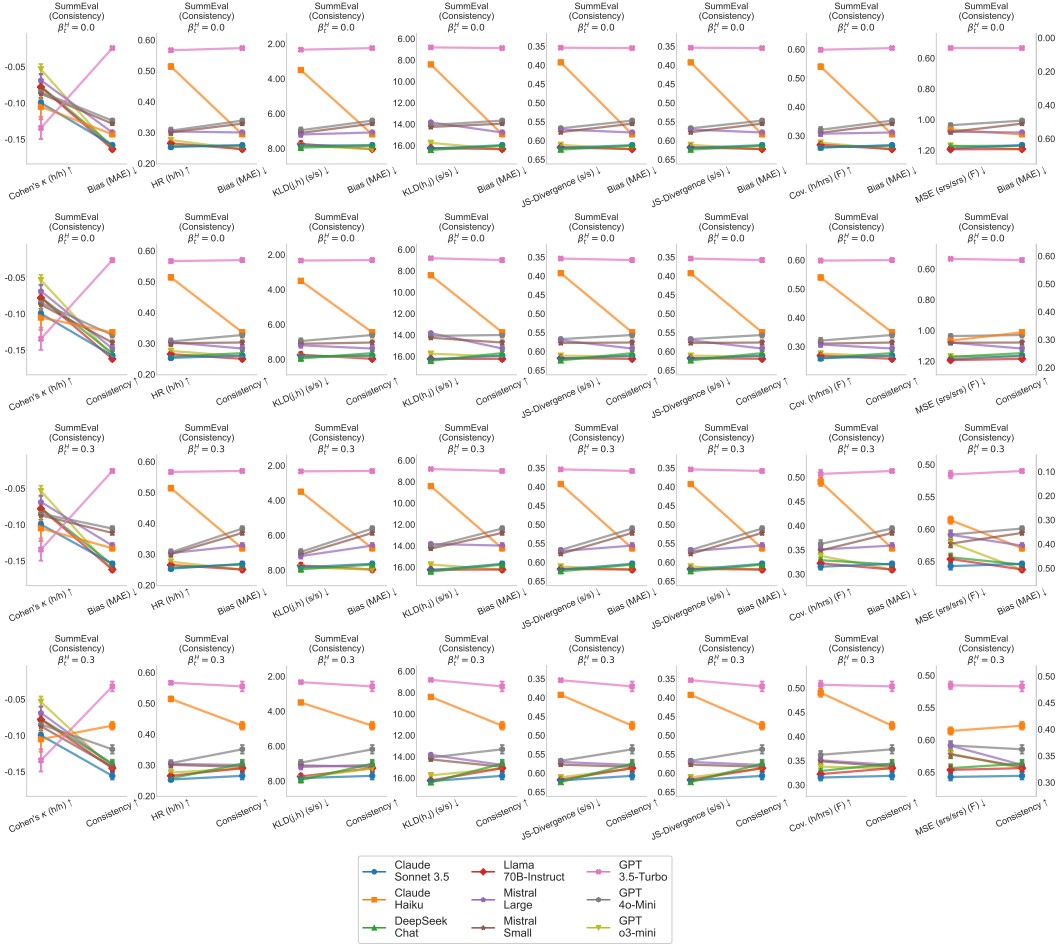

Figure 22: Ranking of models determined by human–judge agreement metrics versus downstream metrics for the SummEval "consistency" rating task. Top two rows: $\beta_t^H = 0$, Bottom two rows: $\beta_t^H = 0.3$.

## D.1 Rating Task Prompts

---

**Response Set (RS) Rating Instructions - Civil Comments**

Rate the toxicity of the following comment:
"{comment_text}"

Select ALL options that could reasonably apply:
  A. Very Toxic - Very hateful, aggressive, or disrespectful; very likely to make you leave a discussion
  B. Toxic - Rude, disrespectful, or unreasonable; somewhat likely to make you leave a discussion
  C. Not Toxic - Not rude or disrespectful; unlikely to make you leave a discussion
**RESPONSE FORMAT:**
  • Provide only the sequence of letters (e.g., "ABC" or "A")
  • Use only letters A, B, and/or C
  • No spaces, punctuation, or other characters
  • Do not provide the same letter more than once (e.g., "AA" or "BB", or "B_" are invalid responses)
  • No explanations

---

**Forced Choice (FC) Rating Instructions - Civil Comments**

Rate the toxicity of the following comment:
"{comment_text}"

Select the ONE option that best applies:
  A. Very Toxic - Very hateful, aggressive, or disrespectful; very likely to make you leave a discussion
  B. Toxic - Rude, disrespectful, or unreasonable; somewhat likely to make you leave a discussion
  C. Not Toxic - Not rude or disrespectful; unlikely to make you leave a discussion
**RESPONSE FORMAT:**
  • Provide only a single letter: A, B, or C
  • The letter must be the first character in your response
  • No spaces, punctuation, or other characters
  • No explanations

---

Figure 23: Rating instructions for Civil Comments "toxicity" annotation task.

> **Response Set (RS) Rating Instructions - SNLI**
>
> Given a context, a statement can be:
> - Definitely correct (Entailment); or
> - Definitely incorrect (Contradiction); or
> - Neither (Neutral).
>
> Your goal is to choose the correct categories for a given pair of context and statement. An automatic detector will estimate your annotation accuracy on this task. If your estimated accuracy is too low, you might be disqualified.
>
> **EXAMPLES:**
> Context: A guitarist is playing in a band.
> Statement: Some people are performing.
> Answer: The statement is definitely correct.
> Now provide a response to the following example:
> Context: "{context}"
> Statement: "{statement}"
> Select ALL options that could reasonably apply:
> - A. Entailment - Definitely correct
> - B. Neither - Neutral
> - C. Contradiction - Definitely incorrect
>
> **RESPONSE FORMAT:**
> - Provide only the sequence of letters (e.g., "ABC" or "A")
> - Use only letters A, B, and/or C
> - No spaces, punctuation, or other characters
> - Do not provide the same letter more than once (e.g., "AA" or "BB", or "B_" are invalid responses)
> - No explanations

> **Forced Choice (FC) Rating Instructions - SNLI**
>
> Given a context, a statement can be either:
> - Definitely correct (Entailment); or
> - Definitely incorrect (Contradiction); or
> - Neither (Neutral).
>
> Your goal is to choose the correct category for a given pair of context and statement. An automatic detector will estimate your annotation accuracy on this task. If your estimated accuracy is too low, you might be disqualified. If you feel uncertain about some examples, just choose the best category you believe the statement should be in.
>
> **EXAMPLES:**
> Context: A guitarist is playing in a band.
> Statement: Some people are performing.
> Answer: The statement is definitely correct.
> Now provide a response to the following example:
> Context: "{context}"
> Statement: "{statement}"
> Select ONE option that best applies:
> - A. Entailment - Definitely correct
> - B. Neither - Neutral
> - C. Contradiction - Definitely incorrect
>
> **RESPONSE FORMAT:**
> - Provide only a single letter: A, B, or C
> - The letter must be the first character in your response
> - No spaces, punctuation, or other characters
> - No explanation

Figure 24: Rating instructions for SNLI natural language inference task.

Given a context, a statement can be:
- Definitely correct (Entailment); or
- Definitely incorrect (Contradiction); or
- Neither (Neutral).

Your goal is to choose the correct categories for a given pair of context and statement. An automatic detector will estimate your annotation accuracy on this task. If your estimated accuracy is too low, you might be disqualified.

**EXAMPLES:**

Context: A guitarist is playing in a band.

Statement: Some people are performing.

Answer: The statement is definitely correct.

Now provide a response to the following example:

Context: "{context}"

Statement: "{statement}"

Select ALL options that could reasonably apply:
- A. Entailment - Definitely correct
- B. Neither - Neutral
- C. Contradiction - Definitely incorrect

**RESPONSE FORMAT:**
- Provide only the sequence of letters (e.g., "ABC" or "A")
- Use only letters A, B, and/or C
- No spaces, punctuation, or other characters
- Do not provide the same letter more than once (e.g., "AA" or "BB", or "B_" are invalid responses)
- No explanations

Given a context, a statement can be either:
- Definitely correct (Entailment); or
- Definitely incorrect (Contradiction); or
- Neither (Neutral).

Your goal is to choose the correct category for a given pair of context and statement. An automatic detector will estimate your annotation accuracy on this task. If your estimated accuracy is too low, you might be disqualified. If you feel uncertain about some examples, just choose the best category you believe the statement should be in.

**EXAMPLES:**

Context: A guitarist is playing in a band.

Statement: Some people are performing.

Answer: The statement is definitely correct.

Now provide a response to the following example:

Context: "{context}"

Statement: "{statement}"

Select ONE option that best applies:
- A. Entailment - Definitely correct
- B. Neither - Neutral
- C. Contradiction - Definitely incorrect

**RESPONSE FORMAT:**
- Provide only a single letter: A, B, or C
- The letter must be the first character in your response
- No spaces, punctuation, or other characters
- No explanation

Figure 25: Rating instructions for MNLI natural language inference task.

Given two observations (O-Beginning and O-Ending), and two hypotheses (H1 and H2), your goal is to choose the hypotheses that are likely to cause O-Beginning to turn into O-Ending. An automatic detector will estimate your annotation accuracy on this task. If your estimated accuracy is too low, you might be disqualified.

**EXAMPLES:**

O-Beginning: Jenny cleaned her house and went to work, leaving the window just a crack open.

H1: A thief broke into the house by pulling open the window.

H2: Her husband went home and close the window.

O-Ending: When Jenny returned home she saw that her house was a mess.

Answer: H1.

Now provide a response to the following example:

O-Beginning: "{o_beginning}"

H1: "{H1}"

H2: "{H2}"

O-Ending: "{o_ending}"

Select ALL options that could reasonably apply:

    A. H1

    B. H2

**RESPONSE FORMAT:**

- Provide only the sequence of letters corresponding to response options (e.g., "AB" or "A")
- Use only letters A or B
- No spaces, punctuation, or other characters
- Do not provide the same letter more than once (e.g., "AA" or "BB", or "B_" are invalid responses)
- No explanations

---

**Forced Choice (FC) Rating Instructions - AlphaNLI**

Given two observations (O-Beginning and O-Ending), and two hypotheses (H1 and H2), your goal is to choose one of the hypotheses that is more likely to cause O-Beginning to turn into O-Ending. An automatic detector will estimate your annotation accuracy on this task. If your estimated accuracy is too low, you might be disqualified. If you feel uncertain about some examples, just choose the best category you believe the statement should be in.

**EXAMPLES:**

O-Beginning: Jenny cleaned her house and went to work, leaving the window just a crack open.

H1: A thief broke into the house by pulling open the window.

H2: Her husband went home and close the window.

O-Ending: When Jenny returned home she saw that her house was a mess.

Answer: H1.

Now provide a response to the following example:

O-Beginning: "{o_beginning}"

H1: "{H1}"

H2: "{H2}"

O-Ending: "{o_ending}"

Select ONE option that best applies:

    A. H1

    B. H2

**RESPONSE FORMAT:**

- Provide only a single letter: A or B
- The letter must be the first character in your response
- No spaces, punctuation, or other characters
- No explanation

Figure 26: Rating instructions for AlphaNLI abductive reasoning task.

**Response Set (RS) Rating Instructions - SummEval Relevance**

You will be given one summary written for a news article. Your task is to rate the summary on one metric. Please make sure you read and understand these instructions carefully.
**Evaluation Criteria:**
Relevance - selection of important content from the source. The summary should include only important information from the source document. Penalize summaries which contain redundancies and excess information.
**Evaluation Steps:**

1. Read the summary and the source document carefully.

2. Compare the summary to the source document and identify the main points of the article.

3. Assess how well the summary covers the main points of the article, and how much irrelevant or redundant information it contains.

4. Select ALL options that reasonably apply, based on different plausible interpretations of the rating criteria.

Now provide a response to the following example:
Article: "{article}"
Summary: "{summary}"
Select ALL options that could reasonably apply:

A. Relevant - The summary captures the main points effectively with minimal redundancy

B. Not Relevant - The summary misses key points or contains excessive irrelevant information

**RESPONSE FORMAT: [omitted for brevity]**

---

**Forced Choice (FC) Rating Instructions - SummEval Relevance**

You will be given one summary written for a news article. Your task is to rate the summary on one metric. Please make sure you read and understand these instructions carefully.
**Evaluation Criteria:**
Relevance - selection of important content from the source. The summary should include only important information from the source document. Penalize summaries which contain redundancies and excess information.
**Evaluation Steps:**

1. Read the summary and the source document carefully.

2. Compare the summary to the source document and identify the main points of the article.

3. Assess how well the summary covers the main points of the article, and how much irrelevant or redundant information it contains.

4. Select ONE option that best applies.

Now provide a response to the following example:
Article: "{article}"
Summary: "{summary}"
Select ONE option that best applies:

A. Relevant - The summary captures the main points effectively with minimal redundancy

B. Not Relevant - The summary misses key points or contains excessive irrelevant information

**RESPONSE FORMAT: [omitted for brevity]**

Figure 27: Rating instructions for SummEval "relevance" rating task.

**Response Set (RS) Rating Instructions - SummEval Coherence**

You will be given one summary written for a news article. Your task is to rate the summary on one metric. Please make sure you read and understand these instructions carefully.
**Evaluation Criteria:**
Coherence - the collective quality of all sentences. We align this dimension with the DUC quality question of structure and coherence whereby the summary should be well-structured and well-organized. The summary should not just be a heap of related information, but should build from sentence to a coherent body of information about a topic.
**Evaluation Steps:**

1. Read the news article carefully and identify the main topic and key points.

2. Read the summary and compare it to the news article. Check if the summary covers the main topic and key points of the news article, and if it presents them in a clear and logical order.

3. Select ALL options that reasonably apply, based on different plausible interpretations of the rating criteria.

Now provide a response to the following example:
Article: "{article}"
Summary: "{summary}"
Select ALL options that could reasonably apply:
    A. Coherent
    B. Incoherent
**RESPONSE FORMAT: [omitted for brevity]**

---

**Forced Choice (FC) Rating Instructions - SummEval Coherence**

You will be given one summary written for a news article. Your task is to rate the summary on one metric. Please make sure you read and understand these instructions carefully.
**Evaluation Criteria:**
Coherence - the collective quality of all sentences. We align this dimension with the DUC quality question of structure and coherence whereby the summary should be well-structured and well-organized. The summary should not just be a heap of related information, but should build from sentence to a coherent body of information about a topic.
**Evaluation Steps:**

1. Read the news article carefully and identify the main topic and key points.

2. Read the summary and compare it to the news article. Check if the summary covers the main topic and key points of the news article, and if it presents them in a clear and logical order.

3. Select ONE option that best applies.

Now provide a response to the following example:
Article: "{article}"
Summary: "{summary}"
Select ONE option that best applies:
    A. Coherent
    B. Incoherent
**RESPONSE FORMAT: [omitted for brevity]**

Figure 28: Rating instructions for SummEval "coherence" rating task.

**Response Set (RS) Rating Instructions - SummEval Consistency**

You will be given one summary written for a news article. Your task is to rate the summary on one metric. Please make sure you read and understand these instructions carefully.
**Evaluation Criteria:**
Consistency - the factual alignment between the summary and the summarized source. A factually consistent summary contains only statements that are entailed by the source document. Penalize summaries that contain hallucinated facts.
**Evaluation Steps:**

1. Read the news article carefully and identify the main facts and details it presents.

2. Read the summary and compare it to the article. Check if the summary contains any factual errors that are not supported by the article.

3. Select ALL options that reasonably apply, based on different plausible interpretations of the rating criteria.

Now provide a response to the following example:
Article: "{article}"
Summary: "{summary}"
Select ALL options that could reasonably apply:
   A. Consistent
   B. Inconsistent
**RESPONSE FORMAT:**
   • Provide only the sequence of letters (e.g., "AB" or "A")
   • Use only letters A or B
   • No spaces, punctuation, or other characters
   • Do not provide the same letter more than once (e.g., "AA" or "BB", or "B_" are invalid responses)
   • No explanations

---

**Forced Choice (FC) Rating Instructions - SummEval Consistency**

You will be given one summary written for a news article. Your task is to rate the summary on one metric. Please make sure you read and understand these instructions carefully.
**Evaluation Criteria:**
Consistency - the factual alignment between the summary and the summarized source. A factually consistent summary contains only statements that are entailed by the source document. Penalize summaries that contain hallucinated facts.
**Evaluation Steps:**

1. Read the news article carefully and identify the main facts and details it presents.

2. Read the summary and compare it to the article. Check if the summary contains any factual errors that are not supported by the article.

3. Select ONE option that best applies.

Now provide a response to the following example:
Article: "{article}"
Summary: "{summary}"
Select ONE option that best applies:
   A. Consistent
   B. Inconsistent
**RESPONSE FORMAT:**
   • Provide only a single letter: A or B
   • The letter must be the first character in your response
   • No spaces, punctuation, or other characters
   • No explanation

Figure 29: Rating instructions for SummEval "consistency" rating task.

**Response Set (RS) Rating Instructions - SummEval Fluency**

You will be given one summary written for a news article. Your task is to rate the summary on one metric. Please make sure you read and understand these instructions carefully.
**Evaluation Criteria:**
Fluency - the quality of the summary in terms of grammar, spelling, punctuation, word choice, and sentence structure.
**Evaluation Steps:**

1. Read the summary carefully.

2. Assess the grammar, spelling, punctuation, word choice, and sentence structure.

3. Select ALL options that reasonably apply, based on different plausible interpretations of the rating criteria.

Now provide a response to the following example:
Article: "{article}"
Summary: "{summary}"
Select ALL options that could reasonably apply:
   A. Fluent - The summary has good grammar, appropriate word choice, and flows naturally
   B. Not Fluent - The summary has errors that affect readability or sound unnatural
**RESPONSE FORMAT:**
   • Provide only the sequence of letters (e.g., "AB" or "A")
   • Use only letters A or B
   • No spaces, punctuation, or other characters
   • Do not provide the same letter more than once (e.g., "AA" or "BB", or "B_" are invalid responses)
   • No explanations

---

**Forced Choice (FC) Rating Instructions - SummEval Fluency**

You will be given one summary written for a news article. Your task is to rate the summary on one metric. Please make sure you read and understand these instructions carefully.
**Evaluation Criteria:**
Fluency - the quality of the summary in terms of grammar, spelling, punctuation, word choice, and sentence structure.
**Evaluation Steps:**

1. Read the summary carefully.

2. Assess the grammar, spelling, punctuation, word choice, and sentence structure.

3. Select ONE option that best applies.

Now provide a response to the following example:
Article: "{article}"
Summary: "{summary}"
Select ONE option that best applies:
   A. Fluent - The summary has good grammar, appropriate word choice, and flows naturally
   B. Not Fluent - The summary has errors that affect readability or sound unnatural
**RESPONSE FORMAT:**
   • Provide only a single letter: A or B
   • The letter must be the first character in your response
   • No spaces, punctuation, or other characters
   • No explanation

Figure 30: Rating instructions for SummEval "fluency" rating task.

## Response Set (RS) Rating Instructions - QAGS

In this task, you will read an article and a sentence.
The task is to determine if the sentence is factually correct given the contents of the article. Many sentences contain portions of text copied directly from the article. Be careful as some sentences may be combinations of two different parts of the article, resulting in sentences that overall aren't supported by the article. Some article sentences may seem out of place (for example, "Scroll down for video"). If the sentence is a copy of an article sentence, including one of these sentences, you should still treat it as factually supported. Otherwise, if the sentence doesn't make sense, you should mark it as not supported. Also note that the article may be cut off at the end.
Now provide a response to the following example:
Article: "{article}"
Sentence: "{sentence}"
Is the sentence supported by the article? Select ALL options that could reasonably apply:
- A. Supported - The sentence is factually correct given the contents of the article
- B. Not Supported - The sentence contains facts not supported by the article

**RESPONSE FORMAT:**
- Provide only the sequence of letters (e.g., "AB" or "A")
- Use only letters A or B
- No spaces, punctuation, or other characters
- Do not provide the same letter more than once (e.g., "AA" or "BB", or "B_" are invalid responses)
- No explanations

## Forced Choice (FC) Rating Instructions - QAGS

In this task, you will read an article and a sentence.
The task is to determine if the sentence is factually correct given the contents of the article. Many sentences contain portions of text copied directly from the article. Be careful as some sentences may be combinations of two different parts of the article, resulting in sentences that overall aren't supported by the article. Some article sentences may seem out of place (for example, "Scroll down for video"). If the sentence is a copy of an article sentence, including one of these sentences, you should still treat it as factually supported. Otherwise, if the sentence doesn't make sense, you should mark it as not supported. Also note that the article may be cut off at the end.
Now provide a response to the following example:
Article: "{article}"
Sentence: "{sentence}"
Is the sentence supported by the article? Select ONE option that best applies:
- A. Supported - The sentence is factually correct given the contents of the article
- B. Not Supported - The sentence contains facts not supported by the article

**RESPONSE FORMAT:**
- Provide only a single letter: A or B
- The letter must be the first character in your response
- No spaces, punctuation, or other characters
- No explanation

Figure 31: Rating instructions for QAGS "factual consistency" rating task.

## Response Set (RS) Rating Instructions - Topical Chat Uses Knowledge

Given a conversation and an interesting fact, your task is to rate how well the response uses the provided fact. Please make sure read and understand these instructions carefully.

**Evaluation Criteria:**

Given the interesting fact that the response is conditioned on, how well does the response use the fact?

**Evaluation Steps:**

1. Read the conversation context, fact, and response carefully.

2. Assess whether the response incorporates or references the provided fact.

3. Select ALL options that reasonably apply, based on different plausible interpretations of the rating criteria.

Now provide a response to the following example:

Fact: "{fact}"

Context: "{context}"

Response: "{response}"

Select ALL options that could reasonably apply:

    A. Uses Knowledge - The response clearly uses or references the fact

    B. Doesn't Use Knowledge - The response does not mention or refer to the fact at all

**RESPONSE FORMAT:**

- Provide only the sequence of letters (e.g., "AB" or "A")
- Use only letters A or B
- No spaces, punctuation, or other characters
- Do not provide the same letter more than once (e.g., "AA" or "BB", or "B_" are invalid responses)
- No explanations

## Forced Choice (FC) Rating Instructions - Topical Chat Uses Knowledge

Given a conversation and an interesting fact, your task is to rate how well the response uses the provided fact. Please make sure you read and understand these instructions carefully.

**Evaluation Criteria:**

Given the interesting fact that the response is conditioned on, how well does the response use the fact?

**Evaluation Steps:**

1. Read the conversation context, fact, and response carefully.

2. Assess whether the response incorporates or references the provided fact.

3. Select ONE option that best applies.

Now provide a response to the following example:

Fact: "{fact}"

Context: "{context}"

Response: "{response}"

Select ONE option that best applies:

    A. Uses Knowledge - The response clearly uses or references the fact

    B. Doesn't Use Knowledge - The response does not mention or refer to the fact at all

**RESPONSE FORMAT:**

- Provide only a single letter: A or B
- The letter must be the first character in your response
- No spaces, punctuation, or other characters
- No explanation

Figure 32: Rating instructions for Topical Chat "uses knowledge" rating task.

**Response Set (RS) Rating Instructions - Topical Chat Understandable**

Given a conversation and a response, your task is to rate whether the response is understandable in the context of the conversation. Please make sure you read and understand these instructions carefully.

**Evaluation Criteria:**

Is the response understandable in the context of the history? (Not if it's on topic, but for example if it uses pronouns they should make sense)

**Evaluation Steps:**

1. Read the conversation context, fact, and response carefully.

2. Assess whether you can understand what the response is trying to communicate.

3. Select ALL options that reasonably apply, based on different plausible interpretations of the rating criteria.

Now provide a response to the following example:

Fact: "{fact}"

Context: "{context}"

Response: "{response}"

Select ALL options that could reasonably apply:

    A. Understandable - You know what the person is trying to say

    B. Not Understandable - The response is difficult to understand

**RESPONSE FORMAT:**

- Provide only the sequence of letters (e.g., "AB" or "A")
- Use only letters A or B
- No spaces, punctuation, or other characters
- Do not provide the same letter more than once (e.g., "AA" or "BB", or "B_" are invalid responses)
- No explanations

---

**Forced Choice (FC) Rating Instructions - Topical Chat Understandable**

Given a conversation and a response, your task is to rate whether the response is understandable in the context of the conversation. Please make sure you read and understand these instructions carefully.

**Evaluation Criteria:**

Is the response understandable in the context of the history? (Not if it's on topic, but for example if it uses pronouns they should make sense)

**Evaluation Steps:**

1. Read the conversation context, fact, and response carefully.

2. Assess whether you can understand what the response is trying to communicate.

3. Select ONE option that best applies.

Now provide a response to the following example:

Fact: "{fact}"

Context: "{context}"

Response: "{response}"

Select ONE option that best applies:

    A. Understandable - You know what the person is trying to say

    B. Not Understandable - The response is difficult to understand

**RESPONSE FORMAT:**

- Provide only a single letter: A or B
- The letter must be the first character in your response
- No spaces, punctuation, or other characters
- No explanation

Figure 33: Rating instructions for Topical Chat "understandable" rating task.

# E    Synthetic Data Experiments: Setup Details and Results

In the main text, we claim that differences in how humans and LLMs resolve indeterminacy in forced-choice rating tasks heavily bias LLM-as-a-judge validations. In this section, we provide evidence for this claim via synthetic experiments. These experiments examine how different operationalizations of human–judge agreement respond to forced choice selection effects. These experiments further characterize the joint effects of forced choice selection and rater error (see § C for the full model).

## E.1    Experiment Design.

*Human Rating Distribution*. For each item, we sample the response set distribution $\boldsymbol{\theta}_i^{*,H} \sim \mathrm{Dir}(\mathbf{1}_{|\mathcal{Q}|})$. We let $\epsilon$ denote the probability that a rater selects an observed response set that differs from their stable response set. We construct the error matrix $\mathbf{E}_i^H$ such that diagonal entries denote the probability of no rating error $(1 - \epsilon)$ and off-diagonal entries denote the probability of rating error $(\epsilon)$. We use a skew parameter $\eta$ to control how errors are distributed across response sets. We let $k = 0$ denote the index of the option used to categorize items (e.g., as "toxic"). The skew parameter controls whether errors systematically favor $(\eta > 0)$ or disfavor $(\eta < 0)$ response sets containing option $k = 0$.

We model $\mathbf{F}_i^H$ by sampling an exponential decay function $\mathbf{F}_{i,k,v}^H \propto \exp(-\gamma^H \cdot r_k)$. Here, $r_k$ denotes the rank (low to high) of the $k$th option in the $v$th response set and $\gamma^H$ controls the extent to which forced choice ratings are biased towards low-index options (e.g., including the index used to categorize options $k = 0$). We compute the forced choice distribution via $\mathbf{O}_i^H = \mathbf{F}_i^H(\mathbf{E}_i^H(\boldsymbol{\theta}_i^{*,H}))$.

*Judge Rating Distribution*. We model judge systems by sampling an ensemble of distributions with varying similarity to the human rating distribution. We control the deviation of the $z$th judge's rating distribution via $\sigma^{J_z} \sim \mathcal{U}(\sigma_{\min}, \sigma_{\max})$. We then sample the $z$th judges' response set distribution by applying $\boldsymbol{\theta}_i^{*,J_z} = \Pi_\Delta\{\boldsymbol{\theta}_i^* + \boldsymbol{\epsilon}_i^{J_z}\}$, where $\boldsymbol{\epsilon}_i^{J_z} \sim \mathcal{N}(0, (\sigma^{J_z})^2\mathbf{I})$ and $\Pi_\Delta$ projects onto the probability simplex. We sample $\mathbf{F}_i^{J_z}$ following the same procedure used to sample the human rating distribution and. Because judge systems are not affected by rater error, we directly compute $\mathbf{O}_i^{J_z} = \mathbf{F}_i^{J_z}(\boldsymbol{\theta}_i^{*,J_z})$.

### E.1.1    Measuring Asymmetry in Human and Judge System Responses to Indeterminacy

To measure the similarity between how humans and the judge system respond to indeterminate items, we measure whether a rater tends to systematically favor or disfavor the option used to categorize each item – e.g., as "toxic", "factually inconsistent", or "irrelevant" – given its inclusion in a response set. This metric, which we refer to as *forced choice selection effects*, is measured via:

$$\Gamma = \frac{1}{|\mathcal{Q}_+|} \sum_{\mathcal{S} \in \mathcal{Q}} \frac{1}{|\mathcal{S}|} P(O = o_k \mid o_k \in \mathcal{S}),$$

where $\mathcal{S} \in \mathcal{Q}$ is a response set, and $\mathcal{Q}_+ \in \mathcal{Q}$ denotes all response sets containing the option at index $k = 0$. This metric is interpreted similarly to an odds ratio, and has the interpretation:

- $\Gamma = 1$ describes a setting where a rater resolves indeterminacy by selecting a random option.
- $\Gamma > 1$ describes a setting where a rater tends to select the positive option — e.g., "toxic", "factually inconsistent" — when they identify multiple options as correct for an item.
- $\Gamma < 1$ describes a setting where a rater tends to select the negative option — e.g., "not toxic", "factually consistent" — when they identify multiple options as correct for an item.

We let $\Gamma^H$ and $\Gamma^{J_z}$ denote human and judge system forced choice selection effects, respectively. We then leverage this metric to construct two conditions:

- **Symmetric selection effects:** $\mathrm{sign}(\Gamma^H - 1) = \mathrm{sign}(\Gamma^{J_z} - 1)$. Symmetric selection effects occur when, on average, humans and the judge system resolve indeterminacy in the same way – i.e., by favoring or disfavoring the option used to categorize each item.
- **Asymmetric selection effects:** $\mathrm{sign}(\Gamma^H - 1) \neq \mathrm{sign}(\Gamma^{J_z} - 1)$. Asymmetric selection effects occur when humans and the judge system resolve indeterminacy differently — i.e., by selecting different forced choice options given that they identify multiple as correct.

**Experimental Parameters.** We construct sampling parameters for humans and the judge system $(\gamma)$ such that $\Gamma \approx \{0.5, 1, 2\}$. We run all experiments with 50 judge systems. We use 100 items

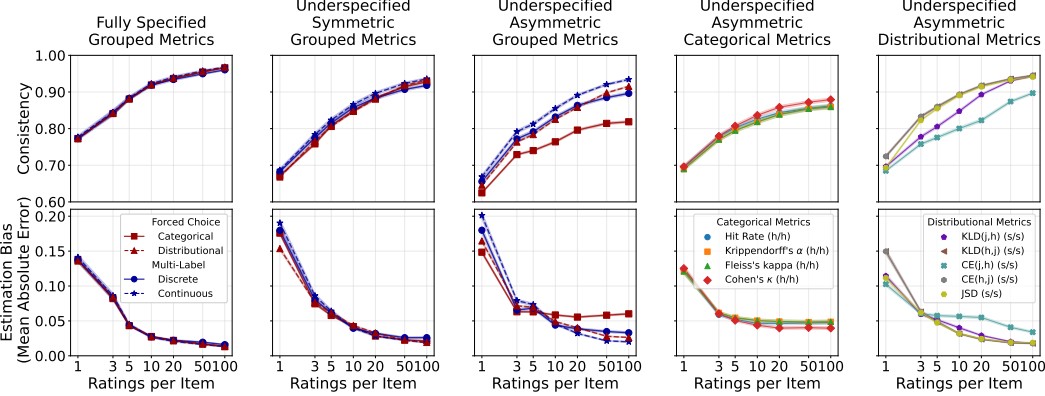

Figure 34: Downstream performance of judge systems selected using different human–judge agreement metrics. The X-axis shows ratings-per-item used to estimate the human rating distribution. Results show that categorical metrics perform poorly under asymmetric FC selection effects (col 3-4).

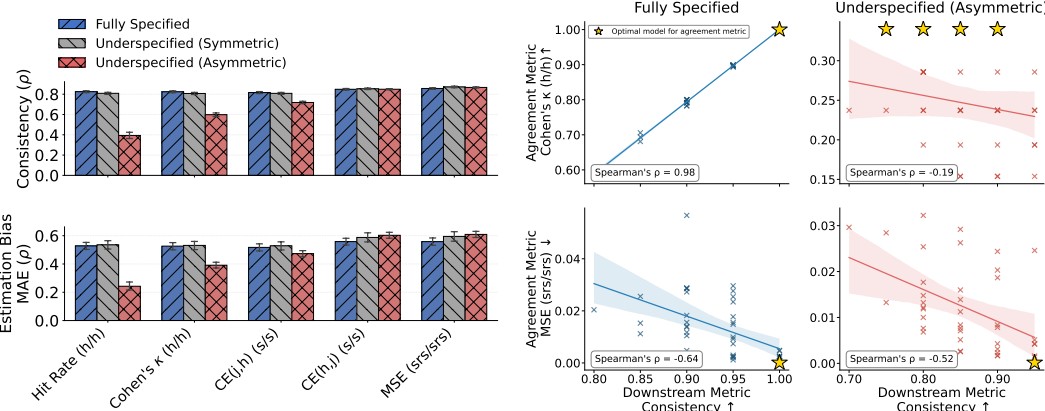

Figure 35: Correlation ($\rho$) between judge rankings from downstream vs. human–judge agreement metrics. Categorical metrics show low correlation under asymmetric FC selection effects.

Figure 36: Saturation effects with categorical metrics under asymmetric forced choice selection effects. Fully specified rating tasks (left column) and MSE (srs/srs) (bottom row) recover the optimal judge system (shown via a star). Categorical metrics recover four equally performant judge systems under asymmetric selection effects (top right).

in all experiments and select option and response set configurations satisfying: $2 \leq |\mathcal{O}| \leq 10$, $2 \leq |\mathcal{Q}| \leq 30$, $|\mathcal{O}| \leq |\mathcal{Q}|$. We let $\sigma_{\min} = 0.02$ and $\sigma_{\max} = .4$ when sampling judge systems.

### E.2 Results.

**Finding 1: When human raters resolve indeterminacy differently from judge systems, agreement metrics measured against forced choice ratings yield sub-optimal selections.** As shown in the right three columns of Figure 34, categorical human–judge agreement metrics select sub-optimal judge systems when (1) rating tasks are underspecified and (2) selection effects are asymmetric. In contrast, we observe that all agreement metrics perform similarly when the rating task is fully specified (Fig. 34, col 1) or underspecified with symmetric selection effects (Fig. 34, col 2). Figure 35 corroborates these findings by indicating weak Spearman correlation between categorical human–judge agreement metrics and downstream performance metrics under asymmetric selection effects. Figure 36 illustrates the mechanism driving instability of categorical metrics. When a rating task is underspecified, categorical agreement metrics do not yield a unique "optimal" judge system with

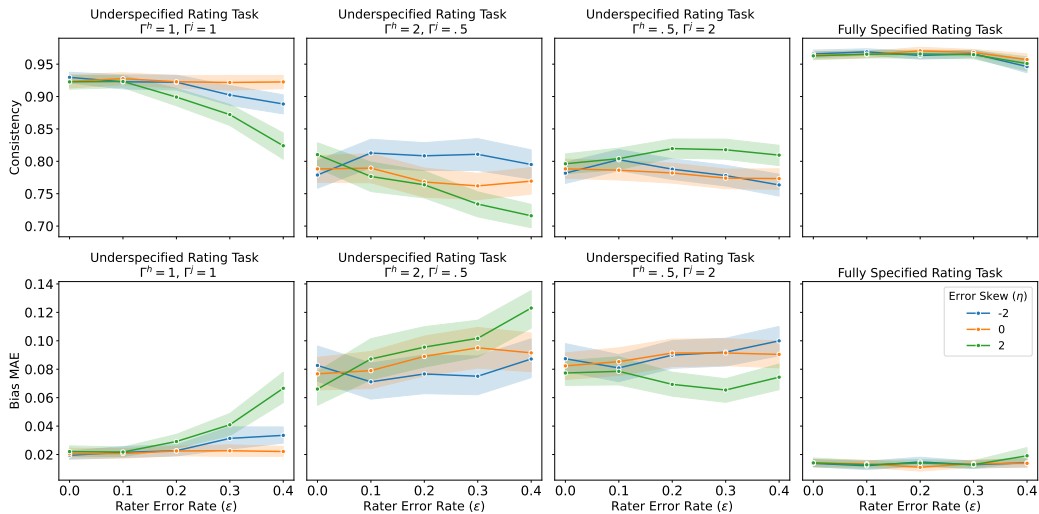

Figure 37: Joint effects of forced choice selection and rater error on the reliability of HR (h/h). Y-axis shows downstream performance of judge systems selected via HR (h/h) with 100 ratings-per-item. We parameterize rater error magnitude via $\epsilon$, which controls the probability that observed and stable response sets differ (§ C). Positive ($\eta = 2$) and negative ($\eta = -2$) skew indicate that errors tend to favor and disfavor option $k = 0$, respectively. Additive instability occurs under positive selection effects and positive skew (left two columns, green). Results averaged over $\tau = [.3, .5, .7]$.

respect to the downstream metric (upper right). This ceiling effect enables selecting a judge system that is optimal for the agreement metric but suboptimal for the downstream metric. In contrast, other configurations shown in Figure 36 all yield a single optimal judge system.

**Finding 2: Fully specified rating tasks enable more performant judge system selections and more effective use of limited annotation budgets.** Across all analyses, we find that fully specifying rating tasks improve judge system selections (Fig. 34, 35, 36). Figure 34 also shows resource benefits associated with fully specifying rating tasks. Judge systems selected with one rating per item on a fully specified task (Fig. 34, left) match the performance of those selected with *three* ratings per item on an underspecified task (Fig. 34, center) – a 66% reduction in per-item requirements (See § G).

**Finding 3: Rater error can further weaken the stability of categorical agreement metrics, but appears less of a concern than forced choice selection effects in our specific setting.** Given the significant impacts of rater error documented in prior work (e.g., [Klie et al., 2023, Plank, 2022, Gordon et al., 2021]), we conduct additional experiments characterizing the effect of rater error on judge system rankings. Figure 37 illustrates that the effects of rater error on the reliability of HR (h/h) are greatest when forced choice selection and rater error are additive (Fig. 37, column 1-2, green). However, we also observe that the rank correlation between error-free and error-corrupted judge system rankings remains high for MSE (srs/srs) and JSD (s/s) across error settings (Fig. 38). Lemma C.8 (Appendix C) pinpoints the mechanism. Because rater error affects human ratings but *not* judge system ratings, there is not an opportunity for asymmetries to arise between humans and the judge system, as there are with forced choice selection. Thus, the impact of rater error on the *comparative ranking* of judge systems remains limited. However, given the synthetic nature of our experiments, further work is needed to characterize the effect of rater error on judge system selection.

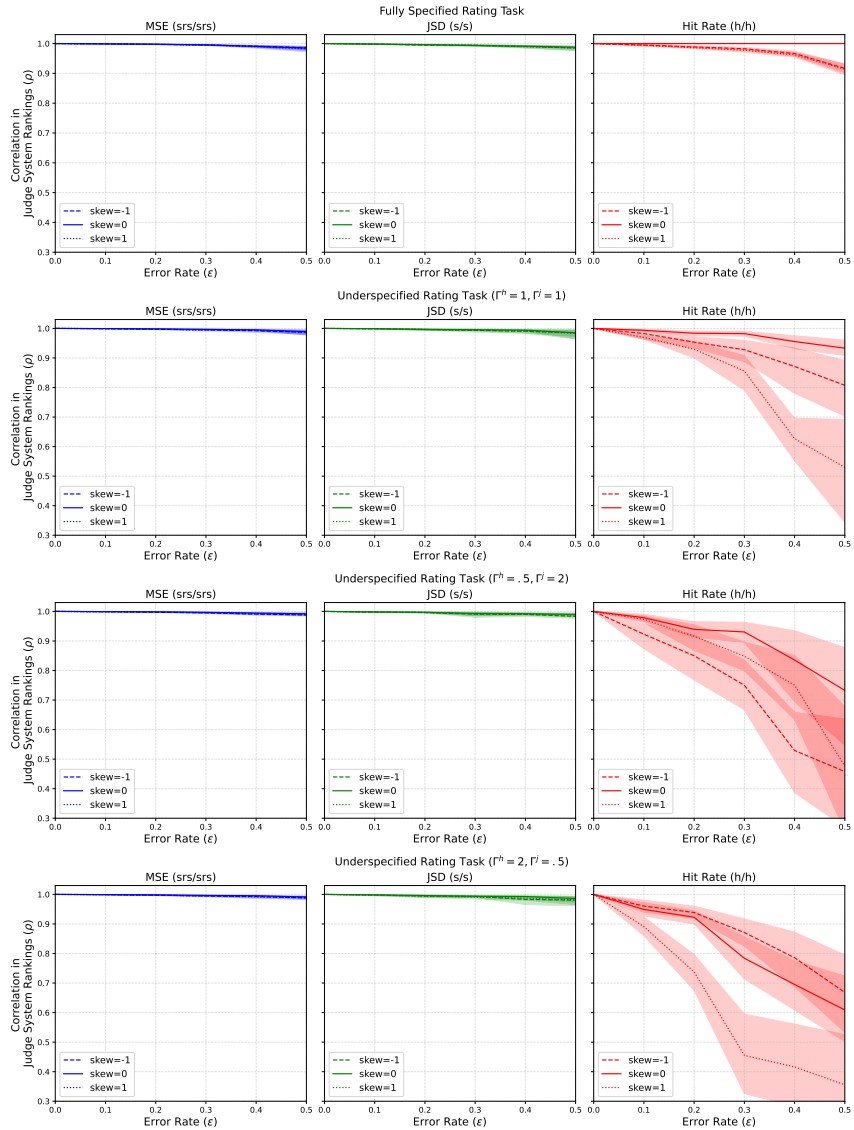

Figure 38: Spearman correlation ($\rho$) between judge system ($N = 50$) rankings obtained via error-corrupted versus error-free human rating distributions. We isolate rater error effects by using population rating distributions, eliminating finite sample estimation error as a confounder. We compute error-corrupted forced choice distribution via $\mathbf{O}_i = \mathbf{F}_i(\mathbf{E}_i\boldsymbol{\theta}_i^*)$ and error-corrupted multi-label vector via $\boldsymbol{\Omega}_i = \boldsymbol{\Lambda}(\mathbf{E}_i\boldsymbol{\theta}_i^*)$. We compute uncorrupted forced choice distribution via $\mathbf{O}_i^* = \mathbf{F}_i(\boldsymbol{\theta}_i^*)$ and uncorrupted multi-label vector via $\boldsymbol{\Omega}_i^* = \boldsymbol{\Lambda}(\boldsymbol{\theta}_i^*)$. We observe that leveraging JSD (s/s) and MSE (srs/srs) as a human-judge agreement yields a consistent ranking of judge systems across error-corrupted and error-free settings, even under a large magnitude of error ($\epsilon$) and non-zero skew ($\eta$). In contrast, using Hit Rate (h/h) to rank judge systems yields rank inconsistencies when (1) the rating task is underspecified, (2) $\epsilon > 0$, and (3) skew ($\eta$) $\neq 0$. This provides additional evidence for the relative insensitivity of judge selection procedures to rater error (as compared to forced choice selection effects), particularly when adopting non-categorical human-judge agreement metrics.

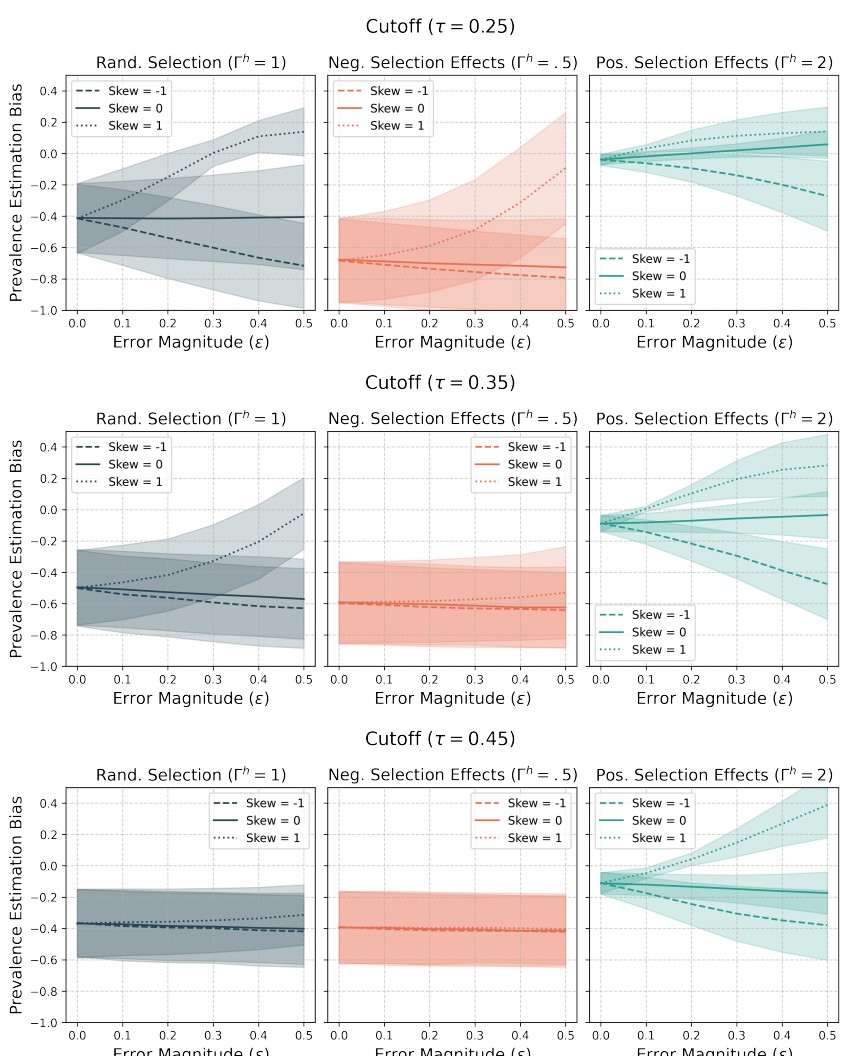

Figure 39: Bias in prevalence estimates obtained from thresholding the forced choice distribution $\mathbf{O}_i = \mathbf{F}_i(\mathbf{E}_i\boldsymbol{\theta}_i^*)$ versus multi-label vector $\boldsymbol{\Omega}_i^* = \boldsymbol{\Lambda}\boldsymbol{\theta}_i^*$. This experiment eliminates finite sample error as a confounder by directly using population rating vectors. Rater error has the largest affect on prevalence estimates when it is correlated with the option ($k = 0$) used to determine item-level categorizations (i.e., has non-zero skew $\eta \neq 0$). Even large magnitudes of error ($\epsilon \geq .4$) has a limited affect on prevalence estimates when $\eta = 0$. Overall, rater error has a significant affect when using human ratings to directly compute $\mathcal{G}_{\text{target}}$ prevalence estimates (shown in this figure). The affect of rater error on judge system selection depends on the specific relationship between rater error and judge system rating distributions (shown in Figures 34, 37, 38, Lemma C.8).

# F  Examples of Monotonicity Violations Among Pairs of Performance Metrics

**Example 1: Hit Rate (Forced Choice) and KL-Divergence (Forced Choice).**

Let $\mathbf{p} = (a_{\text{hard}}^H, a_{\text{hard}}^J, \text{Hit Rate})$ and let $\mathbf{p}_* = (a_{\text{soft}}^H, a_{\text{soft}}^J, \text{KL-Divergence})$. Consider forced choice distributions recovered from human ratings and the judge systems $Z$ and $W$, respectively: $\mathbf{O}^H = a_{\text{soft}}^H(\mathbb{P}^H)$, $\mathbf{O}^{J,Z} = a_{\text{soft}}^J(\mathbb{P}^{J,Z})$, $\mathbf{O}^{J,W} = a_{\text{soft}}^H(\mathbb{P}^{J,W})$, where we omit $i$ from all terms for brevity.

Suppose these distributions are defined over three options $\mathcal{O} = \{o_1, o_2, o_3\}$ and for an item $i$:

Human: $\mathbf{O}^H = (0.6, 0.3, 0.1)$, Judge $Z$: $\mathbf{O}^{J,Z} = (0.8, 0.1, 0.1)$, Judge $W$: $\mathbf{O}^{J,W} = (0.5, 0.4, 0.1)$

Under $\mathbf{p}$, we have:

$$\text{HR}(\mathbf{O}^{J,Z}, \mathbf{O}^H) = 1.0 > \text{HR}(\mathbf{O}^{J,W}, \mathbf{O}^H) = 0.0 \implies \mathcal{G}_{judge}^Z \succ_{\mathbf{p}} \mathcal{G}_{judge}^W.$$

But under $\mathbf{p}_*$, we have:

$$\text{KL}(\mathbf{O}^H \| \mathbf{O}^{J,Z}) \approx 0.15 > \text{KL}(\mathbf{O}^H \| \mathbf{O}^{J,W}) \approx 0.02 \implies \mathcal{G}_{judge}^W \succ_{\mathbf{p}} \mathcal{G}_{judge}^Z.$$

Thus, we have identified a pair of conditional rating distributions $\mathbb{P}^{J,Z}, \mathbb{P}^{J,W}$ and a corresponding human rating distribution $\mathbb{P}^H$ where $\mathbf{p}_*$ is not a monotone transformation of $\mathbf{p}$, so rank consistency between $\mathbf{p}$ and $\mathbf{p}_*$ cannot hold.

---

**Example 2: KL-Divergence (Forced Choice) and MSE (Multi-Label).** Let $\mathbf{p} = (a_{\text{soft}}^H, a_{\text{soft}}^J, \text{KL-divergence})$ and let $\mathbf{p}_* = (a_{\text{srs}}^H, a_{\text{srs}}^J, \text{MSE})$. Let $\mathcal{O} = \{o_1, o_2\}$ and $\mathcal{Q} = \{\{o_1\}, \{o_2\}, \{o_1, o_2\}\}$. Suppose that humans have no rater error (i.e., $\mathbf{E}^H$ and $\mathbf{E}^J$ are both the identity). Let $\mathbb{P}_i^H$ satisfy the decomposition:

$$\mathbf{O}^H = (.4, .6)^\top, \quad \mathbf{F}^H = \begin{bmatrix} 1 & 0 & 0 \\ 0 & 1 & 1 \end{bmatrix}, \quad \boldsymbol{\theta}^{*,h} = (0.4, 0.5, 0.1)^\top, \quad \boldsymbol{\Omega}^H = (.5, .6)^\top.$$

Let $\mathbb{P}_i^{J,Z}$ denote the conditional rating distribution of the judge system satisfying:

$$\mathbf{O}^{J,Z} = (.4, .6)^\top, \quad \mathbf{F}^{J,Z} = \begin{bmatrix} 1 & 0 & 1 \\ 0 & 1 & 0 \end{bmatrix}, \quad \boldsymbol{\theta}^{*,J,Z} = (0.0, 0.6, 0.4)^\top, \quad \boldsymbol{\Omega}^{J,Z} = (.4, .6)^\top.$$

Let $\mathbb{P}_i^{J,W}$ denote the conditional rating distribution of the judge system satisfying:

$$\mathbf{O}^{J,W} = (.5, .5)^\top, \quad \mathbf{F}^{*,J,W} = \begin{bmatrix} 1 & 0 & 1 \\ 0 & 1 & 0 \end{bmatrix}, \quad \boldsymbol{\theta}^{*,J,W} = (0.4, 0.5, 0.1)^\top, \quad \boldsymbol{\Omega}^{J,W} = (.5, .6)^\top.$$

Under $\mathbf{p}$, we have:

$$\text{KL}(\mathbf{O}^H | \mathbf{O}^{J,Z}) = 0 < \text{KL}(\mathbf{O}^H | \mathbf{O}^{J,W}) \approx 0.02 \implies \mathcal{G}_{judge}^Z \succ_{\mathbf{p}} \mathcal{G}_{judge}^W.$$

But under $\mathbf{p}_*$

$$\text{MSE}(\boldsymbol{\Omega}^{J,Z}, \boldsymbol{\Omega}^H) = 0.01 > \text{MSE}(\boldsymbol{\Omega}^{J,W}, \boldsymbol{\Omega}^H) = 0.00 \implies \mathcal{G}_{judge}^W \succ_{\mathbf{p}_*} \mathcal{G}_{judge}^Z.$$

yielding a violation of monotonicity. Thus, we have identified a pair of conditional rating distributions $\mathbb{P}_i^{J,Z}, \mathbb{P}_i^{J,W}$ and a corresponding human rating distribution $\mathbb{P}_i^H$ where $\mathbf{p}_*$ is not a monotone transformation of $\mathbf{p}$, so rank consistency between $\mathbf{p}$ and $\mathbf{p}_*$ cannot hold.

# G Practical Implementation Considerations

We now describe how our framework relates to several practical dimensions that are often of interest during the design of LLM-as-a-judge meta-evaluations.

- **Computational Overhead:** On a per-item (i.e., target system output) basis, our framework imposes *no additional computational overhead* over status quo meta-evaluation practices that leverage (1) forced choice elicitation with (2) hard aggregation and and (3) categorical human–judge agreement metrics. In particular, all operationalizations of human–judge agreement (Table 1) have a constant computational overhead. Thus, adopting our proposed MSE metric recovered from soft response set aggregation imposes no further overhead. Additionally, prompting a judge system via response set elicitation requires the same number of inference calls and output tokens per-call as the status quo forced choice elicitation approach. In the case in which practitioners prompt with both forced choice and response set elicitation, our framework will result in a $2x$ increase in system calls. However, we note that this is not strictly speaking necessary when directly leveraging our proposed multi-label human–judge agreement metrics, which do not make use of forced choice ratings.
- **Cognitive Overhead:** In some cases, prompting human raters with response set elicitation as opposed to forced choice elicitation may increase the cognitive overhead on a per-item basis [Dhar and Simonson, 2003]. For rating tasks with a simple structure (e.g., Yes/No, Win/Tie/Loose), response set elicitation may *decrease* the overall cognitive burden of the rating process by reducing the deliberation required to select a single option when multiple could be viewed as correct. However, in settings with many options, response set elicitation may require a more lengthy process of per-item deliberation — e.g., by requiring raters to carefully and independently assess each option. While our experiments make the benefits of response set elicitation clear (e.g., Fig. 5), future work should explore tradeoffs between the cognitive cost and information gain associated with alternative elicitation regimes.
- **Rating Resource Requirements:** The two points above speak to the *per-item cost* involved with collecting ratings. However, our results also illustrate that the number of ratings-per-item are an important design consideration of LLM-as-a-judge meta-evaluation (Fig. 34). Evaluations with very few ratings per item (e.g., 1-3) fail to capture sufficient information about indeterminacy, resulting in poor judge system selection regardless of the human–judge agreement metric used. This need for multiple ratings is inherent to any approach that aims to capture the distribution of human interpretations, as opposed to a limitation specific to our framework. Importantly, at a fixed rating budget (i.e., ratings-per-item) our proposed approaches—fully specified rating tasks and multi-label agreement metrics—yields to more performant selections of judge systems than the status quo approach that relies on hard aggregation of forced choice ratings with categorical agreement metrics (Fig. 34).
- **Rating Scale Construction:** As noted in § 5, our framework is specifically designed for closed form rating tasks with a discrete set of options. As such, is not designed for open form feedback or continuous rating scales. In general, structuring rating task scales with fewer options enables more efficient per-item estimation of the forced choice distribution, response set distribution, and forced choice translation matrix. This is because tasks with fewer options also have a coarser discretization of the forced choice and response set probability space. This introduces a tradeoff between the discretization of the rating scale and the ratings-per-item required for estimation. Exploring the design of rating scales and elicitation techniques under indeterminacy is an interesting avenue for future research.

## G.1 Performing Response Set Reconstruction with Pre-collected Forced-Choice Ratings.

Practitioners who wish to validate judge systems using a dataset with pre-collected forced-choice ratings have two options.

**Option 1: Perform a sensitivity analysis.** When *no* response set ratings can be collected, practitioners can compare judge systems while systematically varying the relationship between the *known* forced choice distribution and the *unknown* response set distribution:

1. Estimate the $i$'th item's forced choice distribution $\hat{\mathbf{O}}_i$ using forced choice ratings. Each entry in this vector denotes the empirical probability of a rater endorsing the $k$'th forced choice option for the $i$'th item.

2. Obtain a set of reconstructed (or simply hypothesized) forced choice translation matrices $\hat{\mathbf{F}}$. For "reconstruction", one can consider a range of plausible values of the sensitivity parameter $\beta^H \in [0, 1]$. Tables 4-6 (Appendix D) provide examples of how to construct $\hat{\mathbf{F}}$ from $\beta^H$, where we assume that $\hat{\mathbf{F}}$ is fixed across items.

3. Use the set of reconstructed or hypothesized $\hat{\mathbf{F}}$ to obtain estimates of the response set distribution: $\hat{\boldsymbol{\theta}}_i = \hat{\mathbf{F}}\hat{\boldsymbol{O}}_i$.

The estimated response set distribution can then be used to compute multi-label agreement metrics (see Appendix D). To perform this sensitivity analysis, practitioners then check whether the optimal judge system remains constant across different $\hat{\mathbf{F}}$.

**Option 2: Estimate the response set distribution.** When a small additional auxiliary corpus (e.g., 20-100 items) of paired forced choice and response set ratings can be collected, practitioners can directly estimate the response set distribution without requiring a sensitivity parameter. This approach involves the following steps:

1. Estimate the forced choice translation matrix $\hat{\mathbf{F}}$ using the auxiliary corpus. Each entry of $\hat{\mathbf{F}}$ denotes the empirical probability of a rater endorsing the $v$'th response set given that they selected the $k$'th forced choice response. As above, we assume $\hat{\mathbf{F}}$ is fixed across items.

2. Estimate the forced choice distribution $\hat{\mathbf{O}}_i$ for each item.

3. Use $\hat{\mathbf{F}}$ to estimate the response set distribution: $\hat{\boldsymbol{\theta}}_i = \hat{\mathbf{F}}\hat{\mathbf{O}}_i$.

As with option 1 above, the estimated response set distribution can then be used to compute multi-label agreement metrics.

### G.2 Threshold Selection for Downstream Task Metrics.

The threshold parameter $\tau$ used for downstream evaluation tasks is a *policy determination* on the part of the evaluation designer. A small value of $\tau$ (e.g., .1-.3) denotes that an item should be categorized as positive (e.g., for "toxicity") if a small proportion of human raters identify a "positive" interpretation in the text. A large value of $\tau$ (e.g., .8-.9) denotes that many raters need to identify on a "positive" interpretation before an item is categorized as positive. Safety-critical applications (e.g., content moderation, identifying physical safety threats) may benefit from lower thresholds to minimize false negatives; applications requiring high precision may prefer higher thresholds. The interpretation of $\tau$ is analogous to the probability cutoff in binary classifiers: selecting very low or very high values is likely to introduce high agreement rates due to most samples being labeled as negative or positive, respectively. Therefore, when $\tau$ is near 0 or 1, metrics such as the false positive rate or false negative rate may be more informative than accuracy-based measures (e.g., hit-rate). Analogously to computing the AUROC in the binary classification setting, we recommend evaluating over a range of plausible cutoffs $\tau$ to determine the effect of the parameter on judge system selection.

**Societal Impacts.** In this paper, we introduce a framework for LLM-as-a-judge validation in under rating task indeterminacy. Although our arguments have potential societal consequences, especially if the practices we advocate for see adoption and thus change current GenAI evaluation practice, there are no consequences we feel the need to highlight that are specific to this work rather than applicable to any work aiming to improve upon current evaluation practices. The validation practices described in this paper are not an endorsement for the adoption of a judge system in any particular setting.

