# OpenReview forum: "Validating LLM-as-a-Judge Systems under Rating Indeterminacy"
_NeurIPS.cc/2025/Conference — NeurIPS 2025 poster_

### Official Review · Reviewer_yWpq · 2025-06-18

**Clarity:** 2
**Significance:** 3
**Originality:** 3
**Rating:** 4
**Confidence:** 2

**Summary:**

This paper is about evaluation under multi-choice options, where there is no single correct answer that may be adequate for a particular instance. The paper calls this indeterminacy. The paper then goes on to show that the problem can have consequences on which system is judged to be the best.

**Questions:**

Q1 - Would your setup be applicable for text generation tasks rather than discrete annotations?

Q2 - Why do you call it interdeminate? Isn't multi-label annotation and/or multi-choice a common scenario with well-defined evaluation metrics?

Q3 - what often happens is also that no single gold label is chosen from the annotations of each annotator but one considers the distributions of labels over annotators. Isn't that similar to your setup?

**Ethical Concerns:**

["NO or VERY MINOR ethics concerns only"]

**Final Justification:**

The authors clarified some of my concerns and their response helped me to better understand the paper.

**Limitations:**

yes

**Paper Formatting Concerns:**

All ok - except for missing introduction section label.

**Quality:**

3

**Strengths And Weaknesses:**

**Strengths**

The paper is well-written and contains both mathematical insights and experimental results. The setting itself is interesting, i.e., ambiguity in classification tasks rather than forced-choice options.

**Weaknesses**

I must admit that I did not fully follow the paper, but this is mostly likely attributable to me, rather than the paper. On the one hand, I wonder why multi-choice annotations should cause a problem; any labeling task where multiple options could be adequate can be obtained by simply considering the power set and then do standard annotation and evaluation on the power set. It seems I must have surely misunderstood a lot about the framework.

A minor point is that the paper does not have a section 1. I assume this is because the authors wanted to save space, but it is in any case quite unusual.

---

> ### Author Rebuttal · Authors · 2025-07-25
>
> We thank the reviewer for taking the time to deeply engage with our work. We are encouraged that the reviewer finds our problem setting *"interesting"* and the paper to be *"well-written."*
>
> ## Framework Overview
>
> To address the reviewer's concern that they may have misunderstood the framework, we provide a brief overview, grounded in the reviewer’s comments. We note that reviewer yWpq has *already* identified key insights about ambiguous classification tasks: our paper synthesizes these insights into a comprehensive framework for validating judge systems when multiple rating options can be "correct."
>
> Our framework connects two ways of encoding ratings for closed form rating tasks:
> - **Forced choice.** Forced choice ratings are produced by instructing a rater to select a single "correct" option. We call the distribution over forced choice ratings assigned by each annotator the *forced choice distribution*. The distributional (e.g., KL-Divergence) and categorical (e.g., Krippendorff’s $\alpha$) human-judge agreement metrics currently used to validate LLM-as-a-judge systems (e.g., [1,4,5]) can be computed from this forced choice distribution.
> - **Response set.** Suppose that we instead instruct a rater to select *all* options that could be interpreted as “correct”. As reviewer yWpq aptly notes, uncertainty over these option sets can be encoded via a second *response set distribution*: the distribution encoding the probability of each rater endorsing a combination of options in the power set. As reviewer yWpq suggests, the multi-label metrics recovered from response set ratings offer a rich representation of rating uncertainty when more than one option can be "correct."
>
> Surprisingly, however, **multi-label metrics are ***not*** currently used to validate LLM-as-a-judge systems**. Instead, the "state-of-the-art" in both the LLM-as-a-judge setting (i.e., [1,4-5] cited by reviewer BT77) and the traditional model evaluation setting (e.g., [2,3] ) is distributional metrics (e.g., JS-Divergence, KL-Divergence) defined over *forced choice* ratings. In our paper we show empirically via a comprehensive evaluation over 11 rating tasks that measuring human--judge agreement against these status quo metrics yields highly sub-optimal judge system selections (e.g., performing as much as 30% worse, Fig 1).
>
> Of course, the insight that multi-label metrics are preferable under indeterminacy is conceptually quite simple once stated—but to emphasize, this point, however intuitive, is overlooked in existing literature and generally not integrated into practice. We not only make this point but also make the more fundamental contribution of introducing a unifying theoretical model that connects multi-label metrics to their distributional (e.g., KL-Divergence) and categorical (e.g., Cohen's $\kappa$) counterparts. In particular, **we leverage a model for the human rating process (Fig. 3) to establish a novel unifying representation of forced choice and response set operationalizations of human rating uncertainty**, and thus renders them amenable to direct comparison (see Table 2).
>
> This theoretical model also buys us several practical benefits. For instance, we can:
> 1. Establish the necessary conditions for any two metrics defined against any two rating distributions to yield a consistent ranking of judge systems. We show that these conditions are quite strong in practice: even different types of multi-label agreement metrics (e.g., discrete "coverage" metrics versus continuous mean squared error) produce significant performance discrepancies (see Fig 4: Multi-Label Discrete vs Continuous).
>
> 2. Pinpoint the mechanism that causes distributional agreement metrics (e.g., "state-of-the-art" JS-Divergence [1]) to be unreliable. In particular, we identify asymmetries in how LLMs and humans pick a single "correct" forced choice option from their response set as the factor that introduces performance degradation in forced choice agreement metrics.
>
> In sum, reviewer yWpq correctly identifies human rating uncertainty over response sets (i.e., encoded as a subset of the powerset) as a powerful representation of human rating variation under indeterminacy. We build this insight into a theoretically rigorous and empirically validated framework with clear practical recommendations for its adoption going forward.
>
> ## Reviewer Questions
>
> We now address specific questions raised by reviewer yWpq.
>
> > Q1 - Would your setup be applicable for text generation tasks rather than discrete annotations?"
>
> Our setup is applicable to LLM-as-a-judge or human evaluations of how well target models perform on text generation tasks, provided that the LLM-as-a-judge and human evaluations themselves take values on a discrete scale (e.g., such as by scoring generated cover letters on a 1-5 quality scale).  This is the setting that frequently arises when assessing target models’ summarization abilities  (e.g., SummEval) and open-ended model responses (e.g., MT-Bench [12], PRISM [6]) because a discrete rating is ultimately used to quantify and compare output quality. However, our setup is *not directly applicable* to evaluating open-ended generations when a discrete scale is not used. That is, if the LLM-as-a-judge provides an open-ended free-text assessment of a target model’s summary, this is not something our framework handles. We highlight this limitation in Section 4 (line 320).
>
> > Q2 - Why do you call it interdeminate? Isn't multi-label annotation and/or multi-choice a common scenario with well-defined evaluation metrics?
>
> We use the term “indeterminate” to refer to rating tasks where more than one option can be viewed as "correct."  This is distinct from the question of whether multi-label annotation is sought. Indeed, a key issue we identify with existing practice is that forced choice (select just one option) elicitation schemes are often used in indeterminate settings (where a rater may view multiple options as correct).  As mentioned in the review, using a multi-label annotation workflow is much more natural in such settings; but this isn’t always done. Our work provides a robust theoretical and empirical justification for precisely *why* this approach should be adopted when a forced choice rating task is indeterminate.
>
> We get the sense from this and other comments that reviewer yWpq is perhaps surprised that current practice does *not* already apply the response set elicitation approach with corresponding multi-label metrics for which we advocate in this paper.  That so many widely used evaluation datasets for indeterminate tasks were collected under forced choice elicitation was also surprising to us, which is what led us to write this paper.
>
>
> > Q3 - what often happens is also that no single gold label is chosen from the annotations of each annotator but one considers the distributions of labels over annotators. Isn't that similar to your setup?
>
> Absolutely. Modeling a distribution of forced choice or response set labels is core to our probabilistic model for rating variation. Our theoretical framework establishes precisely *how* one should model these distributions of labels over annotations to produce valid selections of judge systems.  In particular, in our work we distinguish between looking at the distribution of *forced choice* responses and *response set* responses.  We show that distributional metrics such as JS-Divergence that consider rating distributions from forced choice elicitation schemes often do lead to misranking. Instead, we argue that the relevant distribution is the one that comes from response set (multi-label) annotation.
>
>
> ## Presentation
>
> > A minor point is that the paper does not have a section 1. I assume this is because the authors wanted to save space, but it is in any case quite unusual.
>
> Thank you very much for letting us know. This missing section header is an editorial oversight that we missed during the final deadline push. We have corrected this in the draft; we also note that the paper remains within the original page limit after other minor formatting adjustments.
>
> ## References:
> [1] Elangovan et al., "Beyond correlation: The Impact of Human Uncertainty in Measuring the Effectiveness of Automatic Evaluation and LLM-as-a-Judge." ICLR, 2025.
>
> [2] Tommaso Fornaciari, Alexandra Uma, Silviu Paun, Barbara Plank, Dirk Hovy, Massimo Poesio, et al. Beyond black & white: Leveraging annotator disagreement via soft-label multi-task learning. NACCL, 2021.
>
> [3] Katherine M Collins, Umang Bhatt, and Adrian Weller. Eliciting and learning with soft labels from every annotator. HCOMP, 2022.
>
> [4] Jaehun Jung, Faeze Brahman, and Yejin Choi. Trust or escalate: Llm judges with provable guarantees for human agreement. ICLR, 2024.
>
> [5] Lianmin Zheng, Wei-Lin Chiang, Ying Sheng, Siyuan Zhuang, Zhanghao Wu, Yonghao Zhuang, Zi Lin, Zhuohan Li, Dacheng Li, Eric Xing, et al. Judging llm-as-a-judge with mt-bench and chatbot arena. NeurIPS, 2023.
>
> [6] Hannah Rose Kirk et al. The PRISM Alignment Dataset: What Participatory, Representative and Individualised Human Feedback Reveals About the Subjective and Multicultural Alignment of Large Language Models. NeurIPS 2024.

---

> ### Comment · Reviewer_yWpq · 2025-08-04
>
> Thanks for your explanations.I am not an expert for this paper, and agree with the authors that the multi-label annotation is intuitive and should be adopted. If indeed a lot of researchers don't do that, then I am willing to increase my score, keeping the low confidence.

---

### Official Review · Reviewer_9S5u · 2025-06-30

**Clarity:** 4
**Significance:** 4
**Originality:** 3
**Rating:** 5
**Confidence:** 4

**Summary:**

This paper addresses a critical limitation in current LLM-as-a-judge validation practices: how to properly evaluate judge systems when rating tasks are indeterminate (i.e., multiple answers may be "correct"). The authors introduce a comprehensive framework distinguishing between forced-choice and response-set elicitation schemes, and demonstrate theoretically and empirically that standard validation approaches can severely misestimate judge performance. Through experiments across 11 rating tasks and 8 commercial LLMs, they show that judges selected using conventional metrics can perform up to 30% worse than those selected using their proposed alternatives. The paper provides concrete recommendations including fully-specified rating tasks, multi-label response elicitation, and continuous multi-label agreement metrics.

**Questions:**

1. Resampling Justification: Line 244 mentions resampling LLMs 10 times per item. Could the authors provide stronger justification for this specific number and analyse sensitivity to this choice?
2. Threshold Selection: How sensitive are the proposed multi-label metrics to the threshold parameter τ used in downstream tasks? Guidelines for selecting appropriate thresholds would be valuable.
3. Cross-Domain Validation: How well does the framework generalise to domains not represented in the current evaluation (e.g., creative writing assessment, scientific accuracy)?

**Ethical Concerns:**

["NO or VERY MINOR ethics concerns only"]

**Final Justification:**

The rebuttal effectively resolved several key technical concerns, particularly around methodology and practical implementation. The authors' systematic sensitivity analysis justifying their resampling choice (N=10) and the robust theoretical backing for error tolerance (Figure 37, Lemma C.8) strengthen confidence in their approach. Their explanation that discrete-option scales are standard practice in LLM-as-a-judge benchmarks, coupled with evidence of biases in continuous scales, provides reasonable justification for scope limitations. The finding that "noisy" response-set ratings still provide robust feedback significantly enhances the practical viability of their framework.

The fundamental issue of generalisability to domains beyond their eleven-task evaluation (such as creative writing or complex reasoning) wasn't fully addressed, and the homogeneity assumptions about human rating processes still pose practical concerns for diverse annotator populations. Despite these limitations, I assign high weight to the excellent technical contributions and theoretical rigour that address a fundamental problem in AI evaluation. The empirical demonstration of 30% performance degradation using standard methods is practically significant, and the actionable recommendations are well-supported. The contributions outweigh the limitations, justifying acceptance whilst acknowledging important areas for future work, particularly empirical validation across more diverse domains and annotator populations.

**Limitations:**

Yes

**Paper Formatting Concerns:**

The paper follows NeurIPS formatting guidelines appropriately.

**Quality:**

3

**Strengths And Weaknesses:**

## Strengths
- Quality: The theoretical framework is rigorous and well-grounded in established psychological research on forced-choice selection effects. The empirical evaluation is comprehensive, spanning multiple domains (toxicity, factuality, relevance) and models. The authors provide formal identifiability results (Theorem 2.1) and necessary conditions for rank consistency (Theorem C.7), establishing clear mathematical foundations for their claims.
- Clarity: The paper is exceptionally well-written with clear exposition of complex concepts. Figure 1 effectively illustrates the core problem, whilst the framework diagram (Figure 2) provides an excellent conceptual overview. The distinction between forced-choice and response-set elicitation is clearly explained and consistently maintained throughout.
- Significance: This work addresses a fundamental problem in AI evaluation that has practical implications for high-stakes applications like content moderation and safety assessment. The finding that standard approaches can select suboptimal judges by substantial margins (30% performance degradation) is practically significant. The framework provides actionable guidance for improving evaluation practices.
- Originality: The application of forced-choice selection effects to LLM-as-a-judge validation appears novel. The theoretical framework connecting different elicitation schemes, aggregation methods, and agreement metrics is original and provides new insights into why current practices fail under indeterminacy.
## Weaknesses
- Limited Scope: The framework only applies to discrete-option rating tasks, excluding continuous scales and open-form feedback, which limits its applicability to some evaluation scenarios.
- Implementation Complexity: Whilst the authors claim no additional computational overhead, collecting response-set ratings may increase cognitive burden for human annotators, particularly for tasks with many options. The practical implementation challenges deserve more thorough consideration.
- Generalisability Concerns: The experiments focus primarily on established benchmark datasets. More investigation of how the framework performs across diverse domains and rating task designs would strengthen the claims.
- Methodological Limitations: The synthetic experiments make strong homogeneity assumptions about the human rating process (noted in limitations), which may not hold in practice across diverse annotator populations.

---

> ### Author Rebuttal · Authors · 2025-07-30
>
> We thank the reviewer for providing a thorough assessment of our work. We appreciate that the reviewer finds our framework *“rigorous and well-grounded”* , *”exceptionally well-written”*, and that it *“addresses a fundamental problem in AI evaluation.”* We would like to engage with several of the key points raised by the reviewer:
>
> &nbsp;
>
> ## Weakness 1: Limited scope
>
> >  The framework only applies to discrete-option rating tasks, excluding continuous scales and open-form feedback, which limits its applicability to some evaluation scenarios.
>
> Thank you for raising this important point on framework scope. While this indeed does restrict the scope of our framework (see limitations Section 4), discrete-option scales are becoming a standard in LLM-as-a-judge rating tasks. In line with standard LLM-as-a-judge benchmarks (e.g., [1,2]), we preclude open-form feedback because a quantitative performance assessment is ultimately required to produce numeric results used to measure performance and compare systems. Existing benchmarks (e.g., JudgeBench [1]) *do* support continuous scales. However, prior work has identified significant biases that arise with continuous scales [3] (e.g., whereby LLMs demonstrate a strong preference for the number "42"). This has led to recommendations that LLM-as-a-judge task designers adopt *discrete-option rating tasks* [4]. In sum, while we acknowledge this as a limitation of our framework, we believe our framework remains applicable in a broad range of practical scenarios.
>
> &nbsp;
>
> ## Weakness 2: Implementation complexity
>
> > Collecting response-set ratings may increase cognitive burden for human annotators, particularly for tasks with many options. The practical implementation challenges deserve more thorough consideration.
>
> Thank you for drawing attention to this important practical question. In Appendix G, **we provide detailed discussion of practical implementation details associated with our framework.** As we discuss in the *Cognitive Overhead* subsection (lines 1347-56), for rating tasks with a simple structure (e.g., Yes/No, Win/Tie/Loose), response set elicitation may decrease the overall cognitive burden of the rating process by reducing the deliberation required to select a single option when multiple could be viewed as correct. However, in settings with many options, response set elicitation may require a more lengthy process of per-item deliberation — e.g., by requiring raters to carefully and independently assess each option.
>
> Fortunately, even "noisy" (i.e., error-corrupted) response set ratings collected from human raters who quickly complete the rating task *may* serve as a robust feedback signal for judge validation. In particular, Figure 37 reports findings from additional synthetic data experiments that examine the robustness of judge system rankings to error in human response set ratings. We observe that multi-label metrics --- i.e., MSE (srs/srs) --- recover robust judge system rankings even when using error-corrupted response set ratings. This indicates that human raters can be instructed to *quickly* complete response set ratings when cognitive overhead is a significant concern. While this empirical finding may seem counterintuitive at first, Lemma C.8 (Appendix C) makes the mechanism clear: because rater error only affects the *human* rating process, relative rankings of *judge systems* remain robust to rater error (over response sets).
>
> We acknowledge that the analysis above remains speculative: in line with the reviewer comment, further work is needed to understand the cognitive load imposed by response set ratings across a range of evaluation tasks.
>
> &nbsp;
> ## Weakness 3: Generalisability to new tasks
>
> > The experiments focus primarily on established benchmark datasets. More investigation of how the framework performs across diverse domains and rating task designs would strengthen the claims.
>
> We agree that further investigation on how the framework generalizes to diverse domains and rating task designs would strengthen evidence for its practical applicability. However, in its current form, our work does evaluate on *eleven diverse rating tasks* spanning concepts such as "toxicity", "entailment", "relevance", "coherence", "factuality", "fluency", "uses knowledge", and "understandable" (Table 3). For comparison, our most closely related work --- Elangovan et al. [5] --- *also evaluates on eleven rating tasks.* Therefore, while there is certainly an opportunity to expand our evaluation further, we believe the current scope sufficiently demonstrates its generalizability across settings.
>
> We plan to release a Colab Notebook alongside our public code, and hope that this will spur productive adoption in a range of practical scenarios.
>
> &nbsp;
> ## Weakness 4: Homogeneity assumptions on the human rating process
>
> > Methodological Limitations: The synthetic experiments make strong homogeneity assumptions about the human rating process (noted in limitations), which may not hold in practice across diverse annotator populations.
>
> As we also note in Section 4 (Limitations), we do place homogeny assumptions on the human rating process, which may be violated in some cases. In plain English, our homogeny assumption holds that different human raters have a similar forced choice translation matrices $F$ governing the translation of response set to forced choice ratings. This assumption can be tested by instructing multiple human raters to provide a small corpus of paired forced choice and response set ratings. If we observe a (statistically significant) difference in translation probabilities across raters for the same items, this provides evidence that the assumption may be violated violated.
>
> When this homogeny assumption *is* violated, approaches that attempt to use the forced choice translation matrix to reconstruct the response set distribution may perform unreliably (see our response to BT77, Question 3). This is because there is no longer a fixed translation matrix that can "reverse" forced choice ratings collected from different human raters. In this setting, it becomes even more critical to collect multi-label (response set) ratings that have not been corrupted by the forced choice translation process.
>
> &nbsp;
> ## Questions
>
> > Resampling Justification: Line 244 mentions resampling LLMs 10 times per item. Could the authors provide stronger justification for this specific number and analyse sensitivity to this choice?
>
> We thank the reviewer for their careful attention to our experimental setup. When sampling judge system responses, our goal is to estimate the forced choice distribution $\hat{O}^J_i$ and the response set distribution $\hat{\theta}^J_i$ from a finite sample of paired forced choice and response set ratings. Since we use empirical response frequencies to estimate these probability vectors (see Appendix D), our estimators are *consistent*: i.e.,  they produce estimates that converge to the true population parameters as sample size increases. Our selected sample size (N=10) operates in the middle of a range that we have already **established via a systematic sensitivity analysis.**
>
> In particular, we conduct synthetic experiments that systematically assess the robustness of our findings to the number of samples used to estimate *human* rating distributions $\hat{O}^H_i$ and $\hat{\theta}^H_i$ (see Figure 33, Appendix D). This experiment assumes that we know the true response set and forced choice distributions for each judge system and estimates $\hat{O}^H_i$, $\hat{\theta}^H_i$ using an experimentally manipulated number of human ratings-per-item. However, because arguments to human--judge agreement metrics are associative, we can also interpret these results as varying the number of samples used to estimate *judge* rating distribution assuming that the *human* rating distribution is known. Findings demonstrate that multi-label agreement metrics perform best across  a varying number of ratings-per-item used to estimate rating distributions (i.e., 1,3,5,10,20,50,100). We also observe that the benefits of multi-label metrics improve as the number of ratings-per-item increases.
>
>
> > Threshold Selection: How sensitive are the proposed multi-label metrics to the threshold parameter τ used in downstream tasks? Guidelines for selecting appropriate thresholds would be valuable.
>
> The threshold parameter $\tau$ used for downstream tasks is a *policy determination* on the part of the evaluation designer. A small value of $\tau$ (e.g., .1-.3) denotes that an item should be categorized as positive (e.g., for "toxicity") if a small proportion of human raters identify a "positive" interpretation in the text. A large value of $\tau$ (e.g., .8-.9) denotes that many raters need to agree on a "positive" interpretation before an item is categorized as positive. We agree that guidance on selection of $\tau$ is important: we will add this discussion to Appendix G (Practical Guidelines).
>
> Further, we systematically vary $\tau$ as part of our analysis. E.g., Figure 4 varies $\tau \in \\{.1,.2, .3,.4,.5,.6,.7,.8,.9 \\}$.
>
> > Cross-Domain Validation: How well does the framework generalise to domains not represented in the current evaluation (e.g., creative writing assessment, scientific accuracy)?
>
> We thank the reviewer for this question. Please see our comments on Weakness 3 above for a detailed response.
>
> &nbsp;
> ## References
>
> [1] Zheng et al, Judging Llm-as-a-Judge with Mt-bench and Chatbot Arena, NeurIPS, 2023.
>
> [2] Tan et al, JudgeBench: A Benchmark for Evaluating LLM-based Judges, ICLR, 2025.
>
> [3] Gu et al, A Survey on LLM-as-a-Judge, 2502.01534, 2025
>
> [4] Husain et al., Frequently Asked Questions (And Answers) About AI Evals, 2025.
>
> [5] Elangovan et al., Beyond correlation: The Impact of Human Uncertainty in Measuring the Effectiveness of Automatic Evaluation and LLM-as-a-Judge. ICLR, 2025.

---

> > ### Author Response · Authors · 2025-08-05
> >
> > Thank you for your detailed feedback on our work. We hope that our rebuttal has addressed the key concerns raised in your review. We would be happy to discuss any follow-up questions you might have during the time remaining in the discussion period.

---

> > ### Comment · Reviewer_9S5u · 2025-08-08
> >
> > The authors provide a comprehensive and well-structured rebuttal that effectively addresses most of the concerns raised. Their response demonstrates strong engagement with the feedback and provides valuable additional context, particularly regarding the practical implementation considerations and the robustness of their approach to rater error (Figure 37, Lemma C.8). The clarification about discrete-option scales being standard practice in LLM-as-a-judge benchmarks, along with references to existing biases in continuous scales, provides good justification for their scope limitations. The response about homogeneity assumptions is technically sound but doesn't fully mitigate concerns about real-world applicability across diverse annotator populations.
> >
> > Overall, the rebuttal strengthens confidence in the work's technical contributions whilst highlighting that some practical limitations remain important considerations for future research. The promise to release a Colab notebook and the systematic sensitivity analysis for the resampling choice (N=10) are particularly valuable additions that enhance reproducibility and practical adoption.

---

### Official Review · Reviewer_BT77 · 2025-07-01

**Clarity:** 3
**Significance:** 3
**Originality:** 1
**Rating:** 4
**Confidence:** 4

**Summary:**

This paper examines the limitations of current LLM-as-a-judge validation practices, especially under tasks with ambiguous or subjective criteria. The authors show that standard forced-choice evaluation schemes can significantly misrepresent the true agreement between human raters and LLM judges. They introduce a probabilistic framework that models rating task indeterminacy and propose multi-label, response-set-based evaluation as a more accurate alternative. Extensive experiments across diverse tasks and models demonstrate that their approach leads to more reliable and fair judge system selection.

**Questions:**

1. Does using distribution-based evaluation signals lead to better results in RLHF or DPO training compared to forced-choice labels?

2. What are the main differences between this work and [1]? Please clarify the novelty over prior work.

3. For practitioners working with pre-existing datasets that used forced-choice ratings, what are the concrete steps to retroactively estimate the response set distribution without collecting additional human annotations?

[1] Elangovan et al., "Beyond correlation: The Impact of Human Uncertainty in Measuring the Effectiveness of Automatic Evaluation and LLM-as-a-Judge." ICLR. 2025

**Ethical Concerns:**

["NO or VERY MINOR ethics concerns only"]

**Final Justification:**

The authors' detailed response is greatly appreciated. The original concerns are addressed. However, I think it might be better if there were some additional results regarding practical usage of this evaluation framework during rebuttal phase. As a result, I decide to keep my original score.

**Limitations:**

yes

**Paper Formatting Concerns:**

1. There is no section heading for the introduction.

2. The citation style does not match that of NeurIPS.

**Quality:**

3

**Strengths And Weaknesses:**

## **Strengths**

1. The paper provides a solid theoretical framework that emphasizes evaluating human and LLM judgments as distributions rather than single labels, which is a significant and valuable perspective.

2. The authors conducted extensive experiments across multiple real-world tasks and a variety of commercial LLMs, demonstrating the empirical validity of their approach.

## **Weaknesses**

1. While the proposed evaluation framework is shown to improve judge system selection, the paper does not provide direct experimental evidence that these improvements translate to better LLM training or downstream task performance (e.g., through RLHF, DPO, or similar methods).

2. The problem formulation and the proposed solution approach appear to be similar to those in [1]. It would be helpful if the authors could clearly articulate the differences between their work and this prior research.

[1] Elangovan et al., "Beyond correlation: The Impact of Human Uncertainty in Measuring the Effectiveness of Automatic Evaluation and LLM-as-a-Judge." ICLR. 2025

---

> ### Author Rebuttal · Authors · 2025-07-30
>
> We thank the reviewer for their thoughtful feedback on our work. We are encouraged that the reviewer finds we provide a *“solid theoretical framework”* with a *“significant and valuable perspective.”*  We also appreciate that the reviewer finds our empirical validation *"extensive."* The reviewer also raises several important concerns, which we carefully address below:
>
> &nbsp;
>
> ## Weakness 1: Limited evidence for improvements in downstream task performance
>
> > While the proposed evaluation framework is shown to improve judge system selection, the paper does not provide direct experimental evidence that these improvements translate to better LLM training or downstream task performance (e.g., through RLHF, DPO, or similar methods).
>
> **Overview.** We thank the reviewer for this excellent suggestion: we agree that showing our framework improves performance on downstream *training* tasks (e.g., RLHF, DPO) is a fruitful direction for future work. Our experiments *do not* address this question directly. However, they *do* provide evidence that our framework leads to significant improvements on **downstream *evaluation* tasks** performed by judge systems. These downstream evaluation tasks are central to key policy decisions, such as whether target system outputs fall within an acceptable safety range for system deployment.
>
> **Example.** In *content filtering* tasks, a rater (human or LLM) decides whether to allow or suppress each target system output -- e.g., based on whether it contains "toxicity", "physical safety threats", or "factual errors." Judge systems are increasingly used for content filtering tasks when generating refusals [1]. We show that status quo validation approaches lead to the selection of judge systems with as much as *30% worse content filtering performance* than those selected using our framework (Figure 1).
>
> **Practical Impact.** In the example above, judge systems selected via status quo approaches provide a false positive or false negative "toxicity" flag 30% more often than those selected with our framework. This can lead to unsafe outputs being incorrectly labeled as "safe" by a judge system or a decision to deploy a target system with poor safety characteristics. In sum, we provide evidence that our framework has a significant downstream impact, albeit in an *evaluation* as opposed to a *training* context.
>
> &nbsp;
>
> ## Weakness 2: Differentiation from Elangovan et al.
> > The problem formulation and the proposed solution approach appear to be similar to those in [2]. It would be helpful if the authors could clearly articulate the differences between their work and this prior research.
>
> **Overview.** **Our approach leveraging response set elicitation with multi-label agreement metrics is more robust to indeterminacy than Elangovan et al. [2]'s.** As the reviewer correctly notes, both Elangovan et al. [2] and our work study a setting where judge systems are validated for "ambiguous" tasks. However, our approaches diverge in how they account for human rating uncertainty throughout the validation process. These differences lead to a significant performance improvement with our proposed approach.
>
> **Conceptual Advance.**  Elangovan et al. [2] represent human rating uncertainty via soft labels over *forced choice ratings*.  However, we show that soft labels are insufficient when each rater identifies more than one "correct" response. This is because soft labels do not account for all options included in a single rater's response set (i.e., containing multiple "correct" responses). Formally, theorem 3.1 establishes that judge systems can be ranked differently when measuring agreement via multi-label (response set) versus distributional (forced choice) metrics used by Elangovan et al. [2].
>
> **Empirical Benefits.** Our use of response set ratings with multi-label metrics translates to a *significant performance improvement over Elangovan et al. [2].* In particular, Elangovan et al. [2] measure Jensen-Shannon Divergence (JSD) between judge system and human labels. **This is reflected by our JS-Divergence (s/s) baseline in Figure 4.** On average, JS-Divergence (s/s) performs 2-12 percentage points worse than our multi-label MSE metric (Figure 4, top right panel). The magnitude of improvement over Elangovan et al. [2] depends on the degree of indeterminacy in the human rating process (Figure 4, $\beta^H$). We will add this discussion above to our related work section to improve clarity for future readers.
>
> &nbsp;
>
> ## Questions
>
> > Does using distribution-based evaluation signals lead to better results in RLHF or DPO training compared to forced-choice labels?
>
> Our framework is designed to support better judge system selection in indeterminate rating tasks.  The question of how these improved judge systems perform as a feedback signal for *training* remains an open area for future work. For instance, a nascent line of work explores the use of synthetic "AI feedback" (e.g., obtained from an LLM-as-a-judge applying a rubric) to guide the training process (e.g., via DPO) [3,4]. However, while prior work has identified flaws affecting LLM-as-a-judge systems (e.g., [5,6]), this work does not examine the effect of judge system selection on the use of the judge as an "AI feedback” signal.
>
> Since judge system quality directly impacts the quality of the feedback signal, improved judge selection should in principle translate to improvements in post-training performance. This suggests that training pipelines using judge systems selected via our multi-label approach *may* in turn provide more effective synthetic "AI feedback." However, in line with experimental setups of prior work [4,5], we do not directly examine the connection with post-training as part of our evaluation. Thus, we leave this interesting line of questions to future work.
>
> > What are the main differences between this work and [2]? Please clarify the novelty over prior work.
>
> See discussion of Weakness 2 above.
>
> > For practitioners working with pre-existing datasets that used forced-choice ratings, what are the concrete steps to retroactively estimate the response set distribution without collecting additional human annotations?
>
>
> Practitioners who wish to validate judge systems via pre-existing datasets with forced choice ratings have two options.
>
> **Option 1: Perform a sensitivity analysis.**  When *no* response set ratings can be collected, practitioners can compare judge systems while systematically varying the relationship between the *known* forced choice distribution and the *unknown* response set distribution. To perform this sensitivity analysis, practitioners can:
>
> 1. Estimate the $i$'th item's forced choice distribution $\hat{O}_{i}$ using forced choice ratings. Each entry in this vector denotes the empirical probability of a rater endorsing the $k$'th forced choice option for the $i$'th item.
> 2. Obtain a set of reconstructed (or simply hypothesized) forced choice translation matrices $\{\hat{F}\}$.  For “reconstruction”, one can consider a range of plausible values of the sensitivity parameter $\beta^H \in [0,1]$. Tables 4-6 (Appendix D) provide examples of how to construct $\hat{F}$ from $\beta^H$, where we assume that $\hat{F}$ is fixed across items.
> 3. Use the set of reconstructed or hypothesized $\{\hat{F}\}$ to obtain estimates of  the response set distribution: $\hat{\theta}_{i} = \hat{F} \hat{O}_i$.
>
> The estimated response set distribution can then be used to compute multi-label agreement metrics (see Appendix D, lines 1235-1241). To perform this sensitivity analysis, practitioners then check whether the optimal judge system remains constant across different  $\{\hat{F}\}$.
>
> **Option 2: Estimate the response set distribution.** When a small additional auxiliary corpus (e.g., 20-100 items) of paired forced choice and response set ratings can be collected, practitioners can directly estimate the response set distribution without requiring a sensitivity parameter. This approach involves the following steps:
>
> 1. Estimate the forced choice translation matrix $\hat{F}$ using the auxiliary corpus. Each entry of $\hat{F}$ denotes the empirical probability of a rater endorsing the $v$'th response set given that they selected the $k$'th forced choice response. As above, we assume $\hat{F}$ is fixed across items.
> 2. Estimate the forced choice distribution $\hat{O}_i$ for each item.
> 3. Use $\hat{F}$ to estimate the response set distribution: $\hat{\theta}_{i} = \hat{F} \hat{O}_i$.
>
> As with option 1 above, the estimated response set distribution can then be used to compute multi-label agreement metrics. We will add these important practical details to our practical implementation guidelines (Appendix G) and provide walk-through instructions in our public code release.
>
>
> &nbsp;
> ## References
> [1] Cui et al., OR-Bench: An Over-Refusal Benchmark for Large Language Models, ICML, 2025.
>
> [2] Elangovan et al., Beyond correlation: The Impact of Human Uncertainty in Measuring the Effectiveness of Automatic Evaluation and LLM-as-a-Judge. ICLR. 2025.
>
> [3] Viswanathan et al., Checklists Are Better Than Reward Models For Aligning Language Models, ArXiv, arXiv/2507.18624. 2025.
>
> [4] Tunstall et al., Zephyr: Direct distillation of lm alignment. ArXiv, abs/2310.16944, 2023.
>
> [5] Zu et al., Investigating Non-Transitivity in LLM-as-a-Judge, ICML, 2025.
>
> [6] Shi et al., Judging the Judges: A Systematic Study of Position Bias in LLM-as-a-Judge, ArXiv, abs/2406.07791. 2024.

---

> > ### Comment · Reviewer_BT77 · 2025-08-05
> >
> > The authors' detailed response is greatly appreciated. The original concerns are addressed.

---

### Official Review · Reviewer_VqiR · 2025-07-03

**Clarity:** 2
**Significance:** 3
**Originality:** 3
**Rating:** 5
**Confidence:** 3

**Summary:**

The paper discusses indeterminate rating tasks for LLM based Judges  and the failure of  the cases where a particular choices are forced in vanilla inference setting  without any reasoning.  The work proposes training-free multi-label metrics based solution to  mitigate the issue of indeterminacy. It is evaluated against 11 rating tasks including  diverse open-weight and close models demonstrating superior performance over previous methods.

**Questions:**

1. Dataset size and sampling details: The paper does not specify the size of the evaluation dataset for each of the 11 tasks. It is unclear whether all samples from the source dataset were used or whether sub-sampling was performed, which is critical for assessing the robustness and generalizability of the results.

2. Citation and dataset clarification: Line 225 references "JudgeBench" without citing it, and the paper does not explicitly identify the five datasets used from JudgeBench. This omission undermines transparency and reproducibility, as it leaves readers without clarity on the source of the evaluation data.

3. Human Evaluation: Is there any human evaluation  performed to ensure the indeterminacy of the tasks? Adding qualitative examples for each evaluation tasks in Appendix could be useful.

**Ethical Concerns:**

["NO or VERY MINOR ethics concerns only"]

**Final Justification:**

The authors have resolved most of my concerns related to reasoning and experimental settings (sampling and no indeterminacy setups). New reasoning model has been added and the proposed method works effectively.

I am unsure about this comment " “No explanations” in the output does not inhibit reasoning traces" regarding the prompting technique and considering "multi-label" without taking the nature order of the options into account explicitly. Hence, I will keep my low confidence.

**Limitations:**

Yes

**Quality:**

3

**Strengths And Weaknesses:**

### Strengths

1. The work introduces a probabilistic theoretical framework for indeterminate rating scenarios, providing a rigorous foundation for modeling uncertainty in ambiguous judge evaluation settings.

2. The proposed method demonstrates significant improvements over existing baselines, achieving up to 30% performance gains in key metrics through its approach to probabilistic reasoning.



### Weaknesses

1. The NO chain-of-thoughts/reasoning ("No explanations" in prompt)  is highly impractical for currently deployed LLMs for a hard task like judgement. Existing benchmarks, such as RewardBench[1], allows the model to reason upto 1024 tokens before generating the final judgement. Hence,  the conclusions from this rigid assumption is questionable due to the way modern LLMs are trained.

2. With the similar motivation of point 1, it is unclear how "No explanations" work for Reasoning models, such as o3/r1 models, as they are  explicitly trained  to generated long CoTs before the final answer.

3. The Independence assumption are broken when the labels are not explicitly specified  to be different, such as for  the prompt in Figure 22, the LLM may  think that  all '"very toxic" queries are also always  "toxic" resulting in returning "AB" instead of only "A" or "B".  Thus, it is unclear how the  proposed method work in such cases.

4.  It is unclear how  the LLMs are sampled, like the temperature, limited the significance of  the reported  result. If non-greedy sampling is used, how many times each query is sent to the model. If only one sample is generated  per query, how does the results, such as Figure 6,  change  for a new run?  If more than one, how  is the result aggregated?

5. The paper only evaluates on possible indeterminate settings, so it is unclear how well the framework performs where indeterminacy is minimal, such as RewardBench[1].

References:

[1] Lambert, Nathan, et al. "Rewardbench: Evaluating reward models for language modeling." arXiv preprint arXiv:2403.13787 (2024).

---

> ### Author Rebuttal · Authors · 2025-07-30
>
> We thank the reviewer for taking the time to provide a detailed review of our paper. We appreciate that the reviewer finds our work provides a *“rigorous foundation for modeling uncertainty in ambiguous judge evaluation settings”* and *“demonstrates significant improvements over existing baselines.”* The reviewer also raises several concerns, which we address below:
>
> ## Weakness 1: Incompatibility with reasoning models.
>
> >The NO chain-of-thoughts/reasoning ("No explanations" in prompt) is highly impractical [...]. Existing benchmarks, such as RewardBench[1], allows the model to reason up to 1024 tokens before generating the final judgement. Hence, the conclusions from this rigid assumption is questionable [...]
>
> **Our framework is fully compatible with judge systems that leverage reasoning and CoT.** As a reminder, the main point of our paper is to provide a framework informing how to evaluate LLM-as-a-judge systems. We place no restrictions on *which* specific judge systems are being evaluated.  The results presented in the paper show that under task indeterminacy, standard approaches to evaluating LLM-as-a-judge system performance lead to incorrect conclusions and fail to identify the truly best performing model. When we started running our experiments, reasoning models were not yet widely available.  We agree with the reviewer that now that reasoning models are broadly available and show high performance, it is worth including a reasoning model in our experiments.
>
> **Technical Note:** The instruction in our rating prompts to include “No explanations” in the output does not inhibit reasoning traces (e.g., Figure 22 referenced in the review). Rather, as with standard benchmarks (e.g., MTBench[1], JudgeBench [2]), this instruction is there to simplify model responses and make it easier to extract the answer. Commercial models (e.g., o1/3, DeepSeek R1) provide reasoning traces in dedicated output fields—the “no explanations” instruction does not suppress reasoning, but does produce simpler outputs in the dedicated output fields. For models where reasoning/CoT are returned in the same field, we recommend excluding that instruction and instead performing output parsing to extract the answer from the unified reasoning + output field.
>
> **Additional reasoning model experiment: o3-mini.** To demonstrate compatibility with reasoning models, we re-ran experiments with o3-mini.  As suggested by the reviewer, we set the max token length to 1024 tokens. We preview a summary of our findings on the TopicalChat Understandable task (Figure 6; columns 3-4) below. We found that it is (1) indeed “in truth” the *second best* performing judge model; (2) misranked as *among the worst* performing models under standard evaluation approaches; (3) *correctly identified by our proposed alternative* as being the *second best* performing judge model.  We will also update all plots in the paper to include data from o3-mini.
>
>
> | Rank              |Ground Truth (Bias MAE)  |  Baseline (Hit-Rate) | Ours (Multi-Label MSE)
> | :---------------- | :------: | :----: |  :----: |
> | #1        |   Mistral Large   | Claude Sonnet 3.5 | Mistral Large |
> | #2        |    **GPT o3-Mini**   | Llama 70B-Instruct | **GPT o3-Mini** |
> | #3        |   GPT-3.5 Turbo   | Mistral Small | Mistral Small |
> | #4        |   Mistral Small   | Mistral Large | Claude Sonnet 3.5 |
> | #5        |   DeepSeek Chat   | DeepSeek Chat | GPT-3.5 Turbo |
> | #6        |   Claude Haiku   | GPT-3.5 Turbo | DeepSeek Chat |
> | #7        |   Claude Sonnet 3.5   | Claude Haiku | Claude Haiku |
> | #8        |   GPT-4o Mini   | GPT-4o Mini | GPT-4o Mini |
> | #9        |   Llama 70B-Instruct   | **GPT o3-Mini** | Llama 70B-Instruct |
>
>
> In sum, these additional experiments including a reasoning model further strengthen our core findings: standard approaches to LLM-as-a-judge performance evaluation give misleading results under indeterminacy, both for reasoning and non-reasoning models.  Our framework allows us to clearly understand these misleading results, and offers an alternative that demonstrably improves upon the standard approach.
>
>
>
> &nbsp;
> ## Weakness 2: Violations of rating option independence
>
> > The Independence assumption are broken when the labels are not explicitly specified to be different… The LLM may think that all '"very toxic" queries are also always "toxic" resulting in returning "AB" instead of only "A" or "B" (Figure 22).
>
> We do not make any independence assumptions that would preclude selecting AB = {“Very Toxic”, “Toxic”} under response set elicitation.  Indeed, the core issue that motivates our work is that many *existing* rating task designs make this unreasonable assumption. I.e., many *existing* rating tasks are structured as forced choice selection tasks requiring that just A xor B is selected, whereas raters actually want to select AB.  Our work shows that forced choice elicitation leads to erroneous conclusions about judge system performance, and we *advocate* for rating task designs that enable AB to be selected. This is clear to see in our response set rating prompt (Figure 22; top), which instructs a model to select all options that could reasonably apply. For instance, in the reviewer's example, a judge system might select both "Very Toxic" and "Toxic.” In contrast, a forced choice rating prompt (Figure 22; bottom) instructs the judge to select a single correct option (e.g., ``Toxic’’). Our results show that LLM-as-a-judge validation is confounded when an LLM includes multiple options in a response set (e.g., "Toxic’", "Very Toxic") but is forced to select a single option as a forced choice response.
>
> &nbsp;
> ## Weakness 3: Documentation of sampling procedure.
>
> > It is unclear how the LLMs are sampled, like the temperature [...]. If non-greedy sampling is used, how many times each query is sent to the model? [...]
>
> Thank you for noting this oversight on our part!  We thought we had included configuration/decoding details in our appendix but we see that indeed we had not.  We will certainly provide complete details in a camera-ready.
>
> **Temperature.** In line with the setup of our most closely related work Elangovan et al [4], we set the temperature of each model to a default value of 1.0.
>
> **Sampling.** We use non-greedy sampling with N=10 samples per item to estimate the empirical distribution over rating task options and response sets. E.g., $\hat{Y}^J = [0.3, 0.2, 0.5]$ corresponds to 3/10 LLM calls returning "Very Toxic", 2/10 calls returning "Toxic", and 5/10 calls returning "Not Toxic." We then (1) aggregate this empirical distribution into a rating task response (e.g., a hard categorical output reflecting the most frequent option) and (2) compute human--judge agreement metrics.
>
> &nbsp;
> ## Weakness 4: Evaluation limited to indeterminate settings
>
> Thank you for raising this important conceptual point. When presenting a method designed to perform better in a more complex setting (e.g., under indeterminacy), it is important to understand whether the method underperforms baselines if applied in the standard setting (here, little-to-no indeterminacy).  Fortunately, our existing experiments show that our recommended method continues to outperform alternatives across varying levels of indeterminacy in the human rating task.
>
> We examine the effects of indeterminacy using a parameter $\beta^H$, which encodes the probability that a human rater endorses multiple items in their response set given that they selected a single forced choice response.  When $\beta^H= 0$, we’re effectively assuming there’s no indeterminacy in the human rating task:  i.e., that human raters give the same response under forced choice as multi-label (response set) elicitation. **Multi-label metrics perform at least as well as baselines when $\beta^H= 0$** (Figure 4).
>
> We also agree that adding rating tasks where indeterminacy is expected to be minimal (e.g., from RewardBench [3]) would provide further evidence for our framework's robustness.
>
> &nbsp;
> ## Questions
> > The paper does not specify the size of the evaluation dataset for each of the 11 tasks. It is unclear whether all samples from the source dataset were used or whether sub-sampling was performed [...]
>
> Please see Appendix D (Item Sampling, lines 1227-1230) for these full details. In short, we randomly sub-sample 200 items for each rating task. These 200 item subsets are of similar size to those included in RewardBench [3] (Table 1). We will release the datasets used for our evaluation for reproducibility.
>
> >  Line 225 references "JudgeBench" without citing it, and the paper does not explicitly identify the five datasets used from JudgeBench. [...]
>
> Thank you for raising this point. In addition to the existing JudgeBench citation on line 320, we have added a second citation at line 225. The specific datasets from JudgeBench are listed on line 225: SNLI, MNLI, α-NLI, SummEval, and QAGS.
>
>
> > Is there any human evaluation performed to ensure the indeterminacy of the tasks? Adding qualitative examples [...] could be useful.
>
> We appreciate this excellent suggestion. We will add qualitative examples from all tasks to Appendix D. Prior work has also extensively studied indeterminacy in several of our rating tasks (e.g., NLI [5] and toxicity [6]). We will add this discussion to our related work.
>
> &nbsp;
> ## References
> [1] Zheng et al, Judging Llm-as-a-Judge with Mt-bench and Chatbot Arena, NeurIPS, 2023.
>
> [2] Tan et al, JudgeBench: A Benchmark for Evaluating LLM-based Judges, ICLR, 2025.
>
> [3] Lambert et al, Rewardbench: Evaluating Reward Models for Language Modeling, arXiv:2403.13787 (2024).
>
> [4] Elangovan et al, Beyond correlation: The Impact of Human Uncertainty in Measuring the Effectiveness of Automatic Evaluation and LLM-as-a-Judge, ICLR, 2025.
>
> [5] Zhang et al, Identifying Inherent Disagreement in Natural Language Inference, NAACL, 2021.
>
> [6] Goyal et al, Is Your Toxicity My Toxicity, CHI, 2022.

---

> > ### Author Response · Authors · 2025-08-05
> >
> > Thank you for your detailed feedback on our work. We hope that our rebuttal has addressed the key concerns raised in your review. We would be happy to discuss any follow-up questions you might have during the time remaining in the discussion period.

---

> > ### Comment · Reviewer_VqiR · 2025-08-08
> >
> > I thank the authors for resolving most of my concerns related to reasoning and experimental settings. I am unsure about this comment " “No explanations” in the output does not inhibit reasoning traces" regarding the prompting technique and considering "multi-label" without taking the nature order of the options into account explicitly. Thus, I will raise my score to 5, with a lower confidence.

---

### Decision · Program_Chairs · 2025-09-17

**Decision:**

Accept (poster)

**Comment:**

# Summary
This paper addresses a critical limitation in current LLM-as-a-judge validation practices: how to properly evaluate judge systems when rating tasks are indeterminate (i.e., multiple answers may be "correct"). The authors introduce a comprehensive framework distinguishing between forced-choice and response-set elicitation schemes, and demonstrate theoretically and empirically that standard validation approaches can severely misestimate judge performance. Through experiments across 11 rating tasks and 8 commercial LLMs, they show that judges selected using conventional metrics can perform up to 30% worse than those selected using their proposed alternatives. The paper provides concrete recommendations including fully-specified rating tasks, multi-label response elicitation, and continuous multi-label agreement metrics.

# Strengths
* The paper provides a solid theoretical framework that emphasizes evaluating human and LLM judgments as distributions rather than single labels, which is a significant and valuable perspective.
* The application of forced-choice selection effects to LLM-as-a-judge validation appears novel.
* The authors conducted extensive experiments across multiple real-world tasks and a variety of commercial LLMs, demonstrating the empirical validity of their approach.
* The paper is exceptionally well-written with clear exposition of complex concepts.

# Overall
The paper introduces a novel perspective on evaluating performance via distributions instead of single labels. All concerns and questions have been addressed by the authors during the author response period.